Cancellous bone and theropod dinosaur locomotion. Part I—an examination of cancellous bone architecture in the hindlimb bones of theropods

Bishop Peter J. 1 2 3 4 pbishop@rvc.ac.uk
http://orcid.org/0000-0003-4157-8434 Hocknull Scott A. 1 2 5
http://orcid.org/0000-0001-8174-3890 Clemente Christofer J. 6 7
http://orcid.org/0000-0002-6767-7038 Hutchinson John R. 8
http://orcid.org/0000-0002-6930-2002 Farke Andrew A. 9
http://orcid.org/0000-0003-1295-6395 Beck Belinda R. 2 10
Barrett Rod S. 2 3
http://orcid.org/0000-0002-0824-9682 Lloyd David G. 2 3
1 Geosciences Program, Queensland Museum , Brisbane, QLD , Australia
2 School of Allied Health Sciences, Griffith University , Gold Coast, QLD , Australia
3 Gold Coast Orthopaedic Research, Engineering and Education Alliance, Menzies Health Institute Queensland , Gold Coast, QLD , Australia
4 Current affiliation: Structure and Motion Laboratory, Department of Comparative Biomedical Sciences, Royal Veterinary College , Hatfield, Hertfordshire , UK
5 School of Biosciences, University of Melbourne , Melbourne, VIC , Australia
6 School of Science and Engineering, University of the Sunshine Coast , Maroochydore, QLD , Australia
7 School of Biological Sciences, University of Queensland , Brisbane, QLD , Australia
8 Structure and Motion Laboratory, Department of Comparative Biomedical Sciences, Royal Veterinary College , Hatfield, Hertfordshire , UK
9 Raymond M. Alf Museum of Paleontology at The Webb Schools , Claremont, CA , USA
10 Exercise and Human Performance, Menzies Health Institute Queensland , Gold Coast, QLD , Australia
Wedel Mathew
Electronic publication date: 2018 Oct 31
Publication date: 2018
Volume: 6
Electronic Location ID: e5778
Received 2018 Jan 8; Accepted 2018 Sep 18
Copyright: © 2018 Bishop et al.
Copyright year: 2018
Copyright holder: Bishop et al.
License: This is an open access article distributed under the terms of the Creative Commons Attribution License, which permits unrestricted use, distribution, reproduction and adaptation in any medium and for any purpose provided that it is properly attributed. For attribution, the original author(s), title, publication source (PeerJ) and either DOI or URL of the article must be cited.
License URL: https://creativecommons.org/licenses/by/4.0/

Keywords: Cancellous bone, Theropod, Bird, Locomotion, Biomechanics

Funding: An Australian Government Research Training Program Scholarship The Paleontological Society An International Society of Biomechanics Matching Dissertation Grant An Australian Research Council DECRA Fellowship DE120101503 The donation of CT scan time and technical assistance by Queensland X-ray This study was financially supported by an Australian Government Research Training Program Scholarship (to Peter Bishop), the Paleontological Society (Robert J. Stanton & James R. Dodd Award, to Peter Bishop), an International Society of Biomechanics Matching Dissertation Grant (to Peter Bishop), an Australian Research Council DECRA Fellowship (DE120101503, to Christofer Clemente) and the donation of CT scan time and technical assistance by Queensland X-ray (to Scott Hocknull). The funders had no role in study design, data collection and analysis, decision to publish, or preparation of the manuscript.

==============================
This paper is the first of a three-part series that investigates the architecture of cancellous (‘spongy’) bone in the main hindlimb bones of theropod dinosaurs, and uses cancellous bone architectural patterns to infer locomotor biomechanics in extinct non-avian species. Cancellous bone is widely known to be highly sensitive to its mechanical environment, and has previously been used to infer locomotor biomechanics in extinct tetrapod vertebrates, especially primates. Despite great promise, cancellous bone architecture has remained little utilized for investigating locomotion in many other extinct vertebrate groups, such as dinosaurs. Documentation and quantification of architectural patterns across a whole bone, and across multiple bones, can provide much information on cancellous bone architectural patterns and variation across species. Additionally, this also lends itself to analysis of the musculoskeletal biomechanical factors involved in a direct, mechanistic fashion.

On this premise, computed tomographic and image analysis techniques were used to describe and analyse the three-dimensional architecture of cancellous bone in the main hindlimb bones of theropod dinosaurs for the first time. A comprehensive survey across many extant and extinct species is produced, identifying several patterns of similarity and contrast between groups. For instance, more stemward non-avian theropods (e.g. ceratosaurs and tyrannosaurids) exhibit cancellous bone architectures more comparable to that present in humans, whereas species more closely related to birds (e.g. paravians) exhibit architectural patterns bearing greater similarity to those of extant birds. Many of the observed patterns may be linked to particular aspects of locomotor biomechanics, such as the degree of hip or knee flexion during stance and gait. A further important observation is the abundance of markedly oblique trabeculae in the diaphyses of the femur and tibia of birds, which in large species produces spiralling patterns along the endosteal surface. Not only do these observations provide new insight into theropod anatomy and behaviour, they also provide the foundation for mechanistic testing of locomotor hypotheses via musculoskeletal biomechanical modelling.

Introduction

Background

Perhaps more than any other group of extinct vertebrates (except hominin primates), dinosaurs have been the subject of extensive research into a wide variety of aspects concerning their palaeobiology. One such aspect is their manner of locomotion, which has often been the topic of much debate. Locomotion has played an important role in arguments surrounding dinosaur physiology, behaviour and palaeoecology (Alexander, 1989; Bakker, 1980, 1986; Bell & Snively, 2008; Horner & Lessem, 1993; Molnar & Farlow, 1990; Ostrom, 1969; Paul, 1988, 2008; Pontzer, Allen & Hutchinson, 2009; Thomas & Farlow, 1997; Thulborn, 1984). Movement has also been important to understanding dinosaur evolution. For example, approximately three-quarters of the features that distinguish dinosaurs from other animals relate to their erect (parasagittal), ancestrally bipedal posture and locomotion (Brusatte, 2012; Novas, 1996). Furthermore, much of dinosaur evolution was accompanied by major changes in locomotor morphology, and by inference, behaviour (Carrano, 2000, 2005; Gatesy, 2002; Gatesy & Middleton, 1997; Hutchinson & Allen, 2009; Maidment et al., 2014; Middleton & Gatesy, 2000; Novas, 1996).

The only primary evidence of dinosaur locomotion available to palaeontologists is the fossils left behind, either body fossils (bones) or trace fossils (footprints and trackways). Fossil footprints and trackways are the most direct line of evidence of locomotion in extinct dinosaurs (Farlow et al., 2012; Gatesy et al., 1999; Gillette & Lockley, 1989; Lockley, 1991; Thulborn, 1990). However, footprints and trackways do not provide direct insight into the movement or coordination of individual limb segments except for the distal limb; moreover, they cannot be definitively assigned to a particular trackmaker (Hutchinson & Gatesy, 2006; Lockley, 1991; Thulborn, 1990). In contrast, the associated bones of the animal’s skeleton can be positively assigned to a given species, and if preserved well can provide insight into parts of the animal that never touched the substrate.

Owing to their often large size and comparatively detailed body fossil record, many investigations have examined how osteology may relate to locomotor behaviour in extinct dinosaurs. These studies have typically focused on externally visible features, such as bone shapes or proportions (Carrano, 1998, 2001, 2005; Christiansen, 1999; Coombs, 1978; Gatesy, 1991b; Gatesy & Middleton, 1997; Maidment & Barrett, 2014; Maidment et al., 2012), joint range of motion (Mallison, 2010a, 2010b; Paul, 1998) or geometrical relationships between inferred muscle lines of action and joints (Bates, Benson & Falkingham, 2012; Carrano, 2000; Hutchinson et al., 2005, 2008; Maidment et al., 2014; Russell, 1972). The insight such studies can provide are usually only general, often having little bearing for understanding the posture or gait of any one species, and moreover carry the caveat of unknowns of soft tissue influences, which may be substantial (Bonnan et al., 2010; Hutchinson & Gatesy, 2006; Tsai & Holliday, 2015). Additionally, these studies may only be able to clarify the range of potential locomotor behaviours used by extinct dinosaurs, rather than the reconstruct the behaviors actually used.

A further line of osteological evidence that has been frequently investigated is the cross-sectional geometry of the mid-shaft of limb bones (Alexander, 1985, 1989, 1991; Christiansen, 1997, 1998; Cubo et al., 2015; Fariña, Vizcaíno & Blanco, 1997; Farke & Alicea, 2009; Farlow, Smith & Robinson, 1995; Heinrich, Ruff & Weishampel, 1993; Lovejoy et al., 2002; Mazzetta, Fariña & Vizcaíno, 1998; Wilson & Carrano, 1999). The implicit assumption of such enquiry is that the manner in which cortical bone is distributed around a diaphyseal cross-section is related to the magnitude and direction of bending and torsional stresses it experiences (Biewener, 1992; Brassey et al., 2013; Wainwright et al., 1976). Therefore, the geometry of a limb bone’s cross-section at midshaft may provide insight into whole-bone loading mechanics, and by extension, locomotor behaviour. However, a growing body of experimental evidence indicates that there is no simple correlation between cortical bone morphology and aspects of bone loading, such as bending direction (Bertram & Biewener, 1988; Biewener & Taylor, 1986; Butcher et al., 2008; Demes, 2007; Demes et al., 1998, 2001; Lieberman, Polk & Demes, 2004; Main & Biewener, 2004; Pearson & Lieberman, 2004; Thomason, 1995; Wallace et al., 2014). Without a strong comparative framework derived from suitable extant species (if they exist), inferences drawn solely from mid-shaft cortical bone morphology should be viewed with caution (see also Farke & Alicea, 2009).

Cancellous bone in brief

One aspect of osteology that has remained understudied by dinosaur palaeontologists is the three-dimensional (3D) architecture of cancellous (‘spongy’) bone, the other main type of bone tissue found in limb bones. Cancellous bone is found throughout the vertebrate skeleton, including in the ends of long bones, vertebrae, throughout short bones (e.g. those of the wrist and ankle) and between the opposing cortices of many flat bones, such as those of the skull (Carter & Beaupré, 2001; Currey, 2002; Martin, Burr & Sharkey, 1998). This work will only consider cancellous bone in the endochondral bones of the appendicular skeleton. Furthermore, it will not consider medullary bone, the loosely packed bone that is periodically formed in birds (Dacke et al., 1993) and at least some dinosaurs (Hübner, 2012; Lee & Werning, 2008; Schweitzer, Wittmeyer & Horner, 2005), despite its superficial similarity to cancellous bone. Medullary bone is rapidly laid down to act as a calcium reservoir for the production of eggshells before they are laid, and consequently its tissue is not as mechanically competent as that of other, permanent bone tissues: its primary function is metabolic, rather than mechanical (Currey, 2002).

The macroscopic architecture of cancellous bone is characterized by a complex, 3D lattice-like array of interlinking bony struts called trabeculae, from the Latin trabecula, meaning ‘small beam’ (Fig. 1). The shape of individual trabeculae may be rod-like, plate-like or some variant in between (Singh, 1978). Despite being not as mechanically competent as cortical bone, cancellous bone forms a key component of the skeleton; in humans, it comprises some 70% of the whole skeleton by volume (Huiskes, 2000).

Figure 1 Cancellous bone occurrence and macrostructure, as illustrated here with the femur of a cow (Bos tauros), sectioned in the coronal plane.

(A) Cancellous bone occurs in the proximal and distal ends of the bone (as indicated by the braces), underlying the thin cortical bone capping the epiphyses and apophyses, as well as the metaphyses. (B) A close-up view of the cancellous bone reveals the high porosity of the tissue, giving it a spongy appearance.

The highly complex macrostructure of cancellous bone gives it an exceptionally high ratio of surface area to volume, which makes it a useful reservoir for calcium homeostasis (Clarke, 2008; Swartz, Parker & Huo, 1998). More importantly, this high surface area also leads to a rate of remodelling that is an order of magnitude greater than that of cortical bone; in humans, some 25% by volume is remodelled per year, compared to 2–3% for cortical bone (Clarke, 2008; Huiskes et al., 2000; Lane, Riley & Wirganowicz, 1996; Parfitt, 1983). This rapid remodelling of cancellous bone allows it to adapt to changes in its mechanical environment more quickly than cortical bone. There is an every-growing body of empirical evidence, derived from both experimental and comparative studies, demonstrating how cancellous bone is highly sensitive and well adapted to its mechanical environment. Moreover, when this mechanical environment changes, cancellous bone is able to adapt its architecture in an accurate and predictable fashion. Much of this work has been recently reviewed in detail by Kivell (2016), and will not be discussed further here.

The fabric of cancellous bone (and why cancellous bone shows directionality)

A salient observation of previous studies is that the orientation of trabeculae (i.e. the fabric of the cancellous bone architecture) is a fundamental component of how cancellous bone is adapted to its environment. Indeed, fabric anisotropy is one of the most important parameters in determining the mechanical behaviour of cancellous bone, second only to bone volume fraction, a measure of porosity (Cowin, 1997; Goldstein, Goulet & McCubbrey, 1993; Kabel et al., 1999; Maquer et al., 2015; Mittra, Rubin & Qin, 2005; Odgaard et al., 1997; Turner, 1992; Turner et al., 1990; Ulrich et al., 1999). Furthermore, the principal material directions1 in cancellous bone are very closely aligned with the principal fabric directions of its architecture (Fig. 2); that is, the principal axes of the mechanical compliance matrix and fabric tensors are closely aligned (Odgaard et al., 1997; Turner et al., 1990)2. Moreover, the degree of fabric anisotropy relates closely with the degree of anisotropy of the mechanical properties: the relative magnitudes of eigenvalues of the fabric tensor closely match that of their respective compliance matrix eigenvalues (Odgaard et al., 1997)3.

Figure 2 Cancellous bone fabric as represented by its principal architectural directions.

(A) A cube of cancellous bone of side length 5.33 mm, from the proximal femur of a freshwater crocodile, Crocodylus johnstoni, with the principal directions of the bone’s architecture superimposed. As an orthotropic material, cancellous bone fabric is completely described by three principal directions. (B) The fabric ellipsoid representation for this cube of cancellous bone is derived from the vectors that describe the principal architectural directions. The ellipsoid’s major, semimajor and minor axes are given by the primary (u1), secondary (u2) and tertiary (u3) directions of the cancellous bone architecture, which correspond to the eigenvectors of the fabric tensor. The relative lengths of each axis depend on the relative magnitudes of the principal directions, which correspond to the eigenvalues of the fabric tensor. The degree of anisotropy (DA) describes the extent to which the trabeculae are aligned within a sample, and is given as the relative magnitude of the primary and tertiary eigenvalues (i.e. DA = e1/e3); in this instance DA = 1.44. The cancellous bone geometry was derived via micro-computed X-ray tomographic scanning (Siemens Inveon, 80 kV, 500 µA, 900 ms exposure, 53.3 µm isotropic resolution) and 3D visualization (Mimics 17.0, Materialise NV, Belgium). The material directions were calculated using the mean intercept length method as implemented in the software Quant3D 2.3 (see Ketcham & Ryan, 2004).

In a comparative context, many previous studies have also demonstrated that differences in cancellous bone fabric direction are indicative of differences in locomotor behaviour, presumably because different behaviours (e.g. joint kinematics) engender different loading regimes and directions thereof (Amson et al., 2017; Barak et al., 2013, Biewener et al., 1996; Carter & Beaupré, 2001; Goldstein et al., 1991; Kamibayashi et al., 1995; Matarazzo, 2015; Podsiadlo et al., 2008; Radin et al., 1982; Ryan & Ketcham, 2005; Van der Meulen et al., 2006). When the loading regimes change, cancellous bone fabric direction also changes, in a highly predictable fashion (Barak, Lieberman & Hublin, 2011; Polk, Blumenfeld & Ahluwalia, 2008; Pontzer et al., 2006) (Fig. 3). These observations also appear to apply across species as well as within species, as demonstrated by work on several species of primate (Barak et al., 2013; Ryan & Ketcham, 2005).

Figure 3 Cancellous bone fabric direction can change in response to experimentally induced changes in mechanical loading.

(A–C) The study of Goldstein et al. (1991). In the distal femur of normal dogs (A), the principal directions of cancellous bone fabric (arrows) vary throughout the bone. After 38 weeks following surgical implantation of load cells (B, arrows indicate direction of applied principal compressive stress), the principal directions of the cancellous fabric were greatly altered, and were reoriented to align with the compressive stress applied by the load cells (C). (D–F) The study of Pontzer et al. (2006). Subjecting guineafowl to running on inclined treadmills caused them to move with a more flexed knee posture compared to running on the level (the angle θ is reduced). The postural change resulted in an altered relative orientation of the joint force that the distal femur experienced (E, F, red arrow), which after 45 days was found to produce a changed orientation of peak trabecular density (dotted arrow).

A strong correspondence between the directionality of cancellous bone and in vivo mechanical loading was first suggested more than 150 years ago (Von Meyer, 1867; Ward, 1838). This became widely publicised by Wolff (1892) as the trajectorial theory, which was proposed as an overarching paradigm that related cancellous bone architecture to its mechanical environment. In its modern formulation (Cowin, 2001), the trajectorial theory can be stated thus: at remodelling equilibrium (Cowin, 1986), the principal material directions of a given volume of cancellous bone are aligned with principal stress trajectories4, but only at spatial scales at which the cancellous bone can be treated as a continuous material (Fig. 4). The continuum scale is the scale at which the mechanical behaviour of a volume of cancellous bone structure can be replaced by a set of material properties that are averaged across the same volume. Only at this scale, or larger, can the averaged architecture and mechanical properties of cancellous bone be legitimately compared with the averaged network of principal stress trajectories (Cowin, Sadegh & Luo, 1992; Martin, Burr & Sharkey, 1998; Oxnard & Yang, 1981; Tsubota, Adachi & Tomita, 2002; Tsubota et al., 2009). The spatial scale at which the continuum concept can be invoked for cancellous bone has been suggested to be at least three to five times trabecular spacing (Cowin, 2001; Cowin, Sadegh & Luo, 1992; Harrigan et al., 1988).

Figure 4 Trabeculae tend to align themselves with the orientation of principal stresses resulting from in vivo loading.

(A) Coronal micro-computed tomographic section through a human proximal femur, illustrating the architecture of cancellous bone. Image provided courtesy of SCANCO Medical AG. (B) A typical loading regime experienced by the proximal femur during locomotion, here the single-legged stance phase of walking (after Rudman, Aspden & Meakin, 2006). This consists of the joint reaction force applied by the acetabulum (JRF), the force of the adductor muscles pulling on the trochanter (add) and the small forces applied by the capsular ligaments (c). (C) Principal stress trajectories resulting from the loading regime in (B), as calculated by a two-dimensional finite element analysis (after Rudman, Aspden & Meakin, 2006). Note the striking correspondence of the main tracts of trabeculae in (A) and the principal stress trajectories in (C).

The trajectorial theory of cancellous bone architecture has received strong support from many experimental (Biewener et al., 1996; Lanyon, 1974; Su et al., 1999) and theoretical studies (Beaupré, Orr & Carter, 1990; Carter, Orr & Fyhrie, 1989; Currey, 2002; Gefen & Seliktar, 2004; Giddings et al., 2000; Hayes & Snyder, 1981; Jacobs, 2000; Jacobs et al., 1997; Koch, 1917; Miller, Fuchs & Arcan, 2002; Pauwels, 1980; Rudman, Aspden & Meakin, 2006; Sverdlova, 2011; Vander Sloten & Van der Perre, 1989), which have repeatedly shown striking similarity between cancellous bone fabric and principal stress trajectories generated from physiological loading. However, whilst it aptly describes the phenomenological association between cancellous bone architecture and its mechanical environment, the trajectorial theory does not link the two together via a mechanistic explanation. Such a mechanistic explanation was provided by Fyhrie & Carter (1986), who demonstrated that strain energy density (SED) in a given volume of cancellous bone is minimized when the architecture is anisotropic such that (i) the direction of maximum stiffness is parallel to that of the maximum principal stress, (ii) the direction of minimum stiffness is parallel to that of the minimum principal stress and (iii) the direction of the intermediate stiffness is parallel to that of the intermediate principal stress. Thus, if SED is a stimulus for trabecular remodelling, cancellous bone adaptation at the continuum level can be mechanistically linked to remodelling activites at the cellular level.

More recent computational modelling studies have shown that SED, or a related measure such as strain or stress, is indeed likely an important driver of trabecular remodelling. Common to each is the notion of the ‘mechanostat’ of bone (Christen et al., 2014; Cresswell et al., 2016; Frost, 1987, 2003; Lambers et al., 2013; Schulte et al., 2013): bone remodels through the addition of bone tissue by osteoblasts to areas experiencing high strain (i.e. overloaded areas) and the removal of bone tissue by osteoclasts from areas experiencing low strain (i.e. underloaded areas) (Figs. 5A–5D). By this process, at remodelling equilibrium all parts of the cancellous structure bear the same amount of strain, or more correctly, their SED is the same. By using a uniform SED as a remodelling objective, numerous continuum-level finite element computational models have predicted bulk density distributions and fabric patterns that accurately reflect reality (Beaupré, Orr & Carter, 1990; Carter & Beaupré, 2001; Carter, Orr & Fyhrie, 1989; Coelho et al., 2009; Jacobs et al., 1997; Kowalczyk, 2010; Turner, Anne & Pidaparti, 1997). More impressively, high-resolution simulations of cellular-level remodelling produce models that spontaneously ‘self-trabeculate’ from an initially isotropic configuration (Martin, Burr & Sharkey, 1998). The result of such simulations is a cancellous structure with trabeculae of realistic proportions, and in those models simulating whole bones, life-like whole-bone architectures (Adachi et al., 2001; Boyle & Kim, 2011; Huiskes et al., 2000; Jang & Kim, 2008, 2010a, 2010b; Mullender & Huiskes, 1995; Phillips, 2012; Phillips, Villette & Modenese, 2015; Ruimerman et al., 2005; Smith et al., 1997; Tsubota, Adachi & Tomita, 2002; Tsubota et al., 2009; Wang et al., 2012) (Figs. 5E and 5F). Moreover, these trabeculae, or more correctly, the fabric directions, are aligned with the continuum-level principal stress trajectories, and when the loading regime changes, the model re-adapts to produce a new cancellous architecture, where the trabeculae are aligned with the new continuum-level principal stress trajectories (Adachi et al., 2001; Huiskes et al., 2000; Mullender & Huiskes, 1995; Ruimerman et al., 2005; Wang et al., 2012). Hence, the trajectorial theory, which is a global pattern observable on the scale of whole bones, may be considered emergent from the local actions of cells.

Figure 5 Cancellous bone remodelling at the cellular level can bring about changes in the entire architecture at the whole-bone level.

(A–D) Schematic illustration of the mechanostat of cancellous bone. Given an initial architecture in (A), a change in the loading regime will lead to some parts becoming overloaded (high stress, dotted) and others becoming underloaded (low stress, horizontal hatching) in (B). Surface remodelling by osteoblasts and osteoclasts (C) acts to deposit additional bone material in those overloaded areas (dark grey) and remove bone material from those underloaded areas (light grey); arrows show direction in which local bone surface moves. This continues ad infinitum until all bone tissue is neither too highly strained nor too little strained. (E, F) Illustrates the application of the mechanostat principal on the level of the whole bone, via computational modelling (adapted from Jang & Kim, 2008, 2010a). In this example of the human proximal femur, with loads simulating both the joint reaction force and forces from the abductor muscles, the initially isotropic architecture (E) undergoes remodelling until equilibrium is reached. The resulting equilibrium architecture (F) is extremely similar to that observed in the real specimen (cf. Fig. 4A).

A rigid application of the trajectorial theory to cancellous bone would predict that trabeculae are oriented at right angles to each other. However, this is often not the case; in fact, orthogonal intersections seem to be the exception, rather than the rule (Murray, 1936). The reason for this apparent paradox is that most bones experience multiple, often diverse loading regimes. It is this adaptation to multiple different loading regimes, each with different principal stress trajectories, that produces the nonorthogonality observed in the majority of cancellous bone architectures (Ben-Zvi et al., 2017; Heřt, 1994; Pidaparti & Turner, 1997; Skedros & Baucom, 2007). Thus, only in bones that tend to experience a single loading regime would an orthogonal ‘trajectorial structure’ be expected in cancellous bone. One such example is the calcaneum of a number of digitigrade mammals, such as sheep (Lanyon, 1974), mule deer (Skedros & Baucom, 2007; Skedros, Hunt & Bloebaum, 2004; Skedros et al., 2007), horses (Vander Sloten & Van der Perre, 1989), macropod marsupials (Biewener et al., 1996) and cattle (Fig. 6). These bones are loaded in an extremely consistent manner, by the pull on the distal end from the Achilles tendon and superficial digital flexor tendons. A uniform strain energy distribution within a volume of cancellous bone is hence not usually achieved in any single given loading regime (Jang & Kim, 2008; Van Rietbergen et al., 1999, 2003). Rather, it is the time-averaged distribution of SED, resulting from multiple daily loading regimes that a bone experiences, which is uniform and which drives cancellous bone remodelling. This has been demonstrated by numerous computational simulations of the bone remodelling process. Specifically, no one loading regime will lead to replication of all the observed architectural features in a bone; only when multiple loading regimes are considered can all of a bone’s cancellous architecture be accounted for by the trajectorial theory (Beaupré, Orr & Carter, 1990; Boyle & Kim, 2011; Carter & Beaupré, 2001; Carter, Orr & Fyhrie, 1989; Coelho et al., 2009; Jacobs et al., 1997; Jang & Kim, 2008, 2010a, 2010b; Phillips, Villette & Modenese, 2015; Sverdlova, 2011; Tsubota, Adachi & Tomita, 2002; Tsubota et al., 2009; Turner, Anne & Pidaparti, 1997).

Figure 6 Orthogonal arrangements of trabeculae usually reflect a highly consistent loading regime experienced by a bone.

(A) Sagittal section through the calcaneum of a cow, with the pull of the Achilles tendon on the distal end indicated by the arrow. (B) A force applied to the free end of a cantilever beam is comparable to the loading regime experienced by the cow calcaneum during locomotion. The bending of the cantilever beam produces principal stress trajectories that are very similar to the overall arrangement of trabeculae in the calcaneum (solid lines are trajectories of compressive stress; dashed lines are trajectories of tensile stress). Since the calcaneum is only loaded in this fashion, the two systems of trabeculae (one curving from up, one curving down) tend to intersect at right angles.

Non-mechanical influences on cancellous bone architecture

Although the architecture of cancellous bone is clearly influenced by its mechanical environment, it may also be influenced by other factors, such as ontogeny and genetics. Epigenetic influences on cancellous bone mechanobiology may also exist, but exactly what these could be, and how much they interact with genetic influences, remains unknown.

Many studies have demonstrated that the cancellous architecture in a particular region of a bone changes considerably throughout the ontogeny of an individual (Abel & Macho, 2011; Gosman & Ketcham, 2009; Gosman, Stout & Larsen, 2011; Nafei et al., 2000a, 2000b; Raichlen et al., 2015; Ryan & Krovitz, 2006; Tanck et al., 2001; Townsley, 1948; Volpato, 2008; Wolschrijn & Weijs, 2005). Such changes are necessitated by increases in absolute bone size, and it is therefore unsurprising that the most rapid changes occur early in ontogeny, during the growth of an individual (Gosman & Ketcham, 2009; Gosman, Stout & Larsen, 2011; Raichlen et al., 2015; Ryan & Krovitz, 2006; Tanck et al., 2001). The timing of these ontogenetic changes in cancellous bone architecture often reflect the timing of ontogenetic changes in locomotor behaviour, especially the initial commencement of sustained locomotor-induced loading (Gosman & Ketcham, 2009; Gosman, Stout & Larsen, 2011; Raichlen et al., 2015; Ryan & Krovitz, 2006; Tanck et al., 2001; Townsley, 1948; Volpato, 2008; Wolschrijn & Weijs, 2005). Therefore, mechanical factors influence cancellous bone remodelling not only in the adult, but across the entire lifespan of an individual.

Given that the adaptation of cancellous bone to its mechanical environment occurs throughout the life of an individual, an interesting proposition arises if the rate at which bone remodels decreases through ontogeny (Christiansen et al., 2000; Keaveny & Yeh, 2002; Lieberman et al., 2003; Pearson & Lieberman, 2004). That is, the adaptive response of cancellous bone in the adult may not be as proficient as in earlier stages of life. If this occurs, the architecture observed in the adult may reflect, to some degree, the habitual loading experienced during ontogeny, and not just the current habitual loading environment (Carlson et al., 2006; Petterson et al., 2010; Pontzer et al., 2006). This phenomenon of ontogenetic inertia has not been investigated in great detail, but a general consideration may nevertheless be made. The potential for ontogenetic inertia in cancellous bone architecture will depend on at least four variables, namely (i) the absolute rate at which cancellous bone remodels, (ii) the lifespan of the individual, (iii) the absolute increase in bone size through ontogeny and (iv) the degree to which locomotor-induced loading changes through ontogeny.

Ontogenetic inertia will be minimal in species that have a high rate of bone remodelling compared to their lifespan. For example, adult humans remodel about 25% by volume of their cancellous bone per year (Huiskes et al., 2000; Parfitt, 1983). Given the lengthy lifespan of humans, this implies that cancellous bone will be turned over many times during the life of an individual, erasing the ‘signals’ of locomotor-induced loading from earlier stages in life. However, there may be a small, immediate component of ontogenetic inertia. This is because bone (re)modelling can only occur on pre-existing bone surfaces (Carter & Beaupré, 2001; Martin, Burr & Sharkey, 1998; Mullender & Huiskes, 1995), and hence there may be some lag left over between successive ‘bone generations’. Over the lifetime of an individual, however, this will be inconsequential. A great increase in the absolute size of limb bones through ontogeny, as seen in humans, will also result in the complete turning over of cancellous bone many times, reducing the magnitude of ontogenetic inertia. If locomotor behaviour does not change appreciably throughout ontogeny, then ontogenetic changes in locomotor-induced bone loading will be minimal. Consequently, the cancellous bone architecture observed in the adult will reflect the current habitual loading environment, because this environment has remained effectively unaltered for a significant length of time. Such a pattern is also observed in humans, where locomotor behaviour effectively matures by the age of 4 years (Sutherland, 1997), and the cancellous bone architecture in the human proximal femur and tibia is effectively unchanged from about 9 years of age onward (Gosman & Ketcham, 2009; Ryan & Krovitz, 2006). Minimal ontogenetic inertia would also be expected for ostriches, which have a high rate of bone remodelling (Currey, 2003), a sizeable lifespan (Davies, 2002), exhibit great increase in bone size from chick to adult, and which show little ontogenetic change in locomotor behaviour as far as limb posture is concerned (Smith, Jespers & Wilson, 2010). One further consideration is that the magnitude of ontogenetic inertia may also depend on if the bone has experienced relatively ‘novel’ mechanical loading conditions in its recent past. In such a situation more rapid remodelling may occur in response to these novel loading conditions (Robling, Castillo & Turner, 2006; Robling & Turner, 2009), serving to ‘erase’ older ontogenetic signal and thereby decreasing the magnitude of ontogenetic inertia.

As regards genetic influences on cancellous bone architecture, these influences probably depend on the scale at which the topic is approached. Many studies have investigated the genetic effects on cancellous bone adaptation to mechanical loading, particularly in different strains of mice, and have shown that genetics can indeed modulate cancellous bone mechanobiology (Havill et al., 2010; Judex et al., 2004; Wallace, Judex & Demes, 2015; Wallace et al., 2012). However, the aforementioned investigations concern within-species differences, and concern very specific regions of a given bone. They hence do not illustrate how genetics influences the adaptation of entire bones, across the skeleton and across species that load their bones in different manners.

In terms of the architecture of whole bones, genetic factors strongly influence a bone’s initial development. The basic structure of the whole bone derives from the systematic expression of positional information encoded in the genome (Lanyon, 1996; Lovejoy et al., 2002). Moreover, recent research has indicated that some aspects of the finer-scale architectural features may also be influenced by genetic factors, in addition to mechanical factors. For example, the gross architecture of cancellous bone (such as density distribution) in the adult human ilium appears quite early on during foetal development, well before the onset of locomotor-induced loading, suggestive of genetic influence (Abel & Macho, 2011; Cunningham & Black, 2009a, 2009b). However, such a phenomenon is not observed in the human proximal femur or tibia (Gosman & Ketcham, 2009; Ryan & Krovitz, 2006). The early appearance of an adult-like gross architecture may alternatively result from in utero muscular contractions, producing mechanical stimulation of the developing bones (Abel & Macho, 2011; Cunningham & Black, 2009a, 2009b; Lanyon, 1974). Further insight is provided by a second example, namely the development of the calcaneum in artiodactyl ungulates. In both sheep (Lanyon, 1974) and mule deer (Skedros, Hunt & Bloebaum, 2004), the cancellous bone architecture observed in the adult calcaneum occurs in the foetus, paralleling the situation in the human ilium. However, when Lanyon and Goodship (reported by Skerry, 2000) transected the Achilles tendon of a developing foetal lamb in utero, they found that subsequent prenatal growth resulted in a disorganized architecture in the experimental calcaneum, compared to the contralateral control. This suggests that in some situations at least, prenatal loading can be responsible for the cancellous bone architecture observed in a newborn animal, possibly diminishing the significance of genetic influences.

In general then, it appears that the influences on a bone’s cancellous architecture shift during ontogeny, from a dominant role of genetic influences in early prenatal development (in utero or in ovo), to an increasingly important role for extragenetic stimuli, such as mechanical loading, in later development and postnatal ontogeny (Skedros et al., 2007). That is, a basic, genetically determined template lays out the gross architecture of cancellous bone, which is subsequently built upon and remodelled during postnatal ontogeny in response to mechanical loading.

Genetic influences on cancellous bone architecture may also extend across species. If the genetic control of cancellous bone architecture is strong enough, the potential arises that the cancellous architecture observed in a given bone in a particular species may not entirely reflect the loads experienced by the individual in life, but also the loads experienced by the homologous bone somewhere in the past of the species’ evolutionary history. For example, if a species of primate that engages in quadrupedal locomotion recently evolved from a species which engaged in leaping locomotion, it may inherit some architectural characteristics from its ancestors. That is, whilst it is a quadruped, its cancellous bone architecture may be somewhat ‘leaper-like’ in nature (Ryan & Ketcham, 2005). This phenomenon of phylogenetic inertia (Blomberg & Garland, 2002) has received only limited attention in the context of cancellous bone architecture, but what research has been conducted shows only a weak effect, if any (Ryan & Shaw, 2012, 2013; Scherf, Harvati & Hublin, 2013; Tsegai et al., 2013). This possibly weak effect further suggests that the architecture of cancellous bone observed in the adult is largely, if not entirely, influenced by mechanical stimuli (Skedros et al., 2007).

The possibility of phylogenetic inertia can only exist if the genome itself (in terms of the allele frequencies at the population level) is subject to the influences of the mechanical loading environment a bone experiences in life. That is, the genome that codes for the initial template of cancellous bone architecture is influenced by how the bone is loaded in life. Or, put another way, patterns of bone loading resulting from a particular locomotor behaviour lead to selection on the phenotype, which in turn influences populational allele frequencies over generations, affecting the predetermination of gross architectural features prior to eventual bone use and loading in life (Lanyon, 1974; Ryan & Shaw, 2012, 2013; Saparin et al., 2011; Townsley, 1948). If epigenetic factors are involved, then the adaptation through the genome may possibly be achieved very quickly. It would indeed be advantageous to have some form of a blueprint in place for the gross architecture of cancellous bone, because this starts the bone ‘off on the right foot’ as soon as the animal is born (or hatches). In precocial species, the young have to start locomoting—and often have to keep up with the adults—as soon as they are born (or hatch); with the cancellous bone architecture already somewhat pre-adapted, this would make the bone more structurally efficient from the very first day of postnatal loading (Gorissen et al., 2016).

In light of the above considerations, there appear to be at least three pathways that the relationship between cancellous bone architecture and its mechanical environment can take (Fig. 7): A direct pathway, whereby locomotor-induced bone loading leads to changes in cancellous bone architecture, via adaptive remodelling throughout the lifetime of an individual.

An indirect pathway, whereby patterns of locomotor behaviour lead to selection on the genome (i.e. adaptation over generations), which in turn affects the genetic predetermination of gross architectural features prior to loading.

A direct pathway, whereby prenatal muscular contractions produce bone loading, leading to adaptive remodelling prior to the commencement of postnatal, locomotor-induced loading.

Figure 7 Three different ways in which the architecture of cancellous bone can be influenced by its mechanical environment.

See main text for full discussion. The dashed grey pathway for epigenetics signifies that it currently remains unknown as to if and how epigenetics may influence cancellous bone mechanobiology.

If epigenetic factors are involved in cancellous bone mechanobiology, then a fourth, indirect, pathway would exist. These pathways are not mutually exclusive of each other, and it is likely that they all contribute to the final architecture observed in cancellous bone, to varying degrees. Pathway (i) can explain changes in cancellous bone architecture in response to changes in loading conditions within individuals, either naturally or experimentally. Pathways (i) and (ii) can explain how cancellous bone architecture reflects the loads experienced in post-natal life, and hence why different cancellous bone architectures reflect different locomotor behaviours, such as those between different species. Pathways (ii) and (iii) can explain the presence of gross architectural features that are characteristic and reflective of adult locomotion, yet which are present in neonates before the onset of locomotor-induced loading.

The utility of cancellous bone in understanding locomotion in extinct, non-avian dinosaurs

Cancellous bone architecture clearly has great potential utility for better understanding locomotor biomechanics in extinct tetrapods. Most previous studies that have used cancellous bone to test hypotheses of behaviour have focused on extinct primates (Barak et al., 2013; D’Anastasio et al., 2013; DeSilva & Devlin, 2012; Hébert, Lebrun & Marivaux, 2012; Macchiarelli et al., 1999; Rook et al., 1999; Ryan & Ketcham, 2002a; Scherf, 2008; Skinner et al., 2015; Su & Carlson, 2017; Su, Wallace & Nakatsukasa, 2013), with few studies directed towards other tetrapod groups (Bishop et al., 2015; Moreno, Carrano & Snyder, 2007; Sues, 1978; Thomason, 1985). Yet extinct, non-avian dinosaurs are a group that would be quite suitable for this kind of investigation. Non-avian dinosaurs lived for a very long period of time and in a wide variety of environments, exhibited a diverse array of locomotor morphologies, and their fossils are relatively abundant and often well-preserved. They are also inferred to have had high rates of bone growth and remodelling, comparable to that of extant mammals and birds (Brusatte, 2012; Currey, 2002; Reid, 2012), in contrast to that observed in most extant, sprawling reptiles (Currey, 2002; De Ricqlès, 1976; Enlow, 1969). Especially in the larger species, they also had both lengthy lifespans and a large change in absolute bone size through ontogeny, having hatched from eggs less than 30 cm in diameter (Horner, 2000). They would hence be expected to show minimal ontogenetic inertia in adults.

To the authors’ knowledge, only two studies have examined cancellous bone architecture as it relates to aspects of non-avian dinosaur biomechanics, although technological limitations may be in part responsible for this. In the first (Sues, 1978), ‘trabeculae’ were observed in the enlarged skull domes of pachycephalosaurs, appearing to be oriented appropriately for receiving the forces that might be experienced during head-butting behaviour. However, these ‘trabeculae’ were later interpreted to be an ontogenetic transitory structure in the growth of the skull bones (Goodwin & Horner, 2004). More recently, cancellous bone in the pedal phalanges of various dinosaur species was imaged using clinical-grade X-ray computed tomographic (CT) scanning, and basic phenomenological interpretations were made (Moreno, Carrano & Snyder, 2007). Similarly, little investigation has been undertaken in the way of cancellous bone architecture in extant dinosaurs(birds). In fact, aside from being qualitatively illustrated on several occasions (Cracraft, 1971; Owen, 1866; Thompson, 1942; Townsley, 1948), and experimentally manipulated according to different loading patterns (Pontzer et al., 2006), the 3D macrostructure of cancellous bone in the limb bones of birds is virtually unstudied.

A phenomenological approach has been the dominant theme of most previous studies of cancellous bone architecture in extinct vertebrates. Often, investigation has largely been limited to comparing the architecture of cancellous bone in a given extinct species to that of extant related species (e.g. DeSilva & Devlin, 2012; Macchiarelli et al., 1999; Ryan & Ketcham, 2002a; Scherf, 2008; Skinner et al., 2015; Thomason, 1985), essentially asking the question ‘what extant species is the extinct one closest to?’ That is, similarity (or difference) in cancellous bone architecture implies similarity (or difference) in locomotor behaviour. Even then, comparisons are often limited to discrete sub-regions of the bone of interest, rather than architectural patterns throughout a whole bone. Whilst this approach may serve as a good starting point, it cannot by itself provide insight into questions of whole-bone loading or musculoskeletal mechanics. Furthermore, whilst such an approach may be useful when applied to a group of animals with close extant relatives of similar morphologies (such as primates), it may not be useful when investigating extinct animals that are quite different from any extant animal group (such as non-avian dinosaurs).

A more appropriate way of investigating cancellous bone architecture in any extinct species is through a holistic, whole-bone, biomechanically-informed approach. By considering the architecture of cancellous bone throughout an entire bone, more insight may potentially be gained, compared to focusing on a limited number of specific regions (Georgiou et al., 2018; Gross et al., 2014; Kivell, 2016; Ryan & Test, 2007; Saparin et al., 2011; Scherf, 2008; Skinner et al., 2015; Stephens et al., 2016; Su, Wallace & Nakatsukasa, 2013; Tsegai et al., 2013, 2017). Even more insight may be possible by considering the architectural patterns of multiple bones, rather than just one (Saers et al., 2016; Stephens et al., 2016; Tsegai et al., 2017). Most of the aforementioned whole-bone studies have focused on how scalar variables (e.g. bone volume fraction) vary throughout a bone. However, as the orientation of cancellous bone fabric is quite telling of loading conditions, and strongly reflects the mechanical performance of cancellous bone tissue as a whole, it is also deserving of considerable research attention. Importantly, cancellous bone fabric can also be related to bone loading mechanics across the scale of whole bones, via the trajectorial theory: principal material directions, and hence principal fabric directions, are aligned with continuum-level principal stresses engendered by physiological loading. Thus, when a whole-bone approach is taken, cancellous bone fabric can be linked with whole-limb musculoskeletal biomechanics in a mechanistic fashion, rather than just phenomenologically.

Outline of this study

In the present study, and the two subsequent parts of this series (Bishop et al., 2018a, 2018b), cancellous bone architecture was investigated in one particular sub-group of dinosaurs, the theropods, to demonstrate how the investigation of microstructural characteristics across whole bones has the potential to provide unparalleled insight into questions of posture and loading mechanics. Theropoda include some of the most iconic of extinct animals, as well as the most species-rich group of modern-day terrestrial vertebrates, the birds (Bennett & Owens, 2002; Chiappe & Witmer, 2002; Gauthier, 1986; Holtz, 2012; Naish, 2012; Sereno, 1999; Weishampel, Dodson & Osmólska, 2004). Over their 230 million year history, theropods have spanned an incredible range of body size, from the two gram Mellisuga helenae (bee hummingbird) to the eight tonne Tyrannosaurus rex (Dunning, 2007; Henderson, 1999; Hutchinson et al., 2011), and, despite being exclusively bipedal, have displayed a wide range of locomotor morphologies (Baumel & Witmer, 1993; Carrano, 1998; Gatesy & Middleton, 1997; Middleton & Gatesy, 2000; Paul, 1988). This makes them excellent candidates for research into questions concerning the biomechanics of terrestrial locomotion, as well as the consequences of large body size on locomotor performance. Additionally, studies of theropod locomotion are critical to charting the evolution of locomotor behaviour on the line to modern birds, including the origin of a novel locomotor pattern, avian flight (Allen, Paxton & Hutchinson, 2009; Gatesy, 2002; Heers & Dial, 2012; Hutchinson & Gatesy, 2000).

Terrestrial locomotion in theropods has received considerable attention over the past three decades, and a substantially more detailed picture of non-avian theropod stance and gait, and its evolution, has emerged. It is now well established that on the line to modern birds, many profound anatomical changes occurred in theropods, including significant modifications of pelvic and hindlimb osteology (Carrano, 2000; Gatesy & Middleton, 1997; Hutchinson, 2001a, 2001b), musculature and proportions (Carrano & Hutchinson, 2002; Hutchinson, 2001a, 2001b, 2002), changes to tail length and construction (Gatesy, 1990, 1995, 2002; Pittman et al., 2013), and changes in the position of the whole-body centre of mass (Allen et al., 2013). These are inferred to have influenced limb posture, from more upright (less flexed hips and knees) in most forms to more crouched (more flexed hips and knees) in the more derived forms (Bates, Benson & Falkingham, 2012; Carrano, 1998; Gatesy, 1990, 1991b, 1995; Gatesy, Bäker & Hutchinson, 2009; Grossi et al., 2014; Hutchinson et al., 2005), as well as the muscular strategies of limb support and propulsion (Gatesy, 1990, 1995, 2002; Hutchinson & Gatesy, 2000). In turn, bone loading mechanics is also inferred to have changed markedly through theropod evolution (Carrano, 1998; Farke & Alicea, 2009). All of these changes were set against a backdrop of substantial body size evolution, with sustained miniaturisation occuring along much of the stem lineage (Benson et al., 2017; Lee et al., 2014; Turner et al., 2007), but also with many instances of secondary gigantism (Benson et al., 2017; Carrano, 2006; Lee et al., 2014). Studies of cancellous bone architecture have the potential to provide new and improved insight on many of these changes.

Here, the 3D architecture of cancellous bone was investigated in the principal hindlimb bones of a variety of extinct, non-avian theropod and extant ground-dwelling bird species. Investigation focused mainly on the direction of the cancellous bone fabric and how this varies spatially throughout a given bone. The reasoning for this is threefold: The direction of fabric alignment is one of the more telling aspects of cancellous bone architecture in terms of identifying differences in locomotor behaviour and bone loading (Barak, Lieberman & Hublin, 2011; Barak et al., 2013; Goldstein et al., 1991; Polk, Blumenfeld & Ahluwalia, 2008; Pontzer et al., 2006; Ryan & Ketcham, 2005).

When considered across the whole bone, the 3D pattern of fabric directions can be analysed within the framework of the trajectorial theory. This encompassing approach provides greater power to an analysis, because this facilitates direct, mechanistic comparisons of cancellous bone architecture to whole-bone loading, as will be done in Parts II (Bishop et al., 2018a) and III (Bishop et al., 2018b).

Fabric direction is probably more reliably assessed for fossil specimens, as opposed to other features such as bone volume fraction, trabecular thickness or trabecular spacing. Although these other architectural features can also be useful for interpreting locomotor biomechanics (Kivell, 2016), their investigation requires excellent preservation of the entire fossil and very high resolution imaging, the latter of which is difficult (if not impossible) for large bones. So long as the gross structure is preserved and able to be imaged, fabric direction can be assessed.

The observations made for theropod limb bones were also compared to those for theropod outgroups (crocodilians and lizards), as well as the other extant obligate biped, humans, which have been very well characterised with respect to cancellous bone architecture and locomotor biomechanics.

The research presented here in Part I includes first and foremost a comprehensive assessment of the gross architectural patterns in the hindlimb bones of many different theropod species, which constitutes a completely novel dataset. In addition to laying the foundations for future studies, it will facilitate the identification of major patterns of similarity and difference between species and between groups. This in turn can elucidate how cancellous bone architecture may have evolved in theropods, and provide new and unique insight into theropod locomotor biomechanics. The manner of loading that is associated with cancellous bone architecture, and how this may reflect differences in posture, muscle control or gross loading regimes (e.g. bending- vs. torsion-dominant), will form the subject of Parts II (Bishop et al., 2018a) and III (Bishop et al., 2018b).

Materials and Methods

The sampling and methods used in the present study are outlined in full below. In brief, this study acquired X-ray CT scans of the main bones of the hindlimb of a range of avian and non-avian theropods, as well as extant reptilian outgroup species and humans, and used a variety of image processing and analysis techniques to help characterise the architecture of cancellous bone in these elements. Approximately 1.45 TB of CT scan data was obtained for over 160 bones, representing at least 44 species (Table 1). Owing to various logistical constraints, these bones were scanned using a variety of machines at a variety of resolutions. Coupled with the fossilization of many of the specimens, these varied resolutions required several different image processing protocols to extract the structural data. Likewise, a number of different analytical approaches were used, some more quantitative than others, to identify the predominant architectural patterns present. For some of the quantitative data, statistical analyses were also conducted to test for correlations of certain architectural features with body size. The whole procedure of data processing and analysis, undertaken on two computers with ≥32 GB of memory and a 2.4 GHz processor each, took approximately 6 months to complete.

Table 1 The specimens investigated.

Higher-order taxonomy	Species	Specimen number*	Element	CT scan settings	Study	Image processing protocol	
Machine	Peak tube voltage (kV)	Tube current (mA)	Exposure time (ms)	In-plane pixel resolution (mm)	Slice thickness (mm)	
Mammalia, Hominidae	Homo sapiens	GU S-0013	Femurp	Stratec XCT 3000	61.3	0.203	100,000	0.161	0.1	This study	3	
Mammalia, Hominidae	Homo sapiens	GU S-0013	Femurd	Stratec XCT 3000	61.3	0.203	100,000	0.137	0.1	This study	3	
Mammalia, Hominidae	Homo sapiens	GU S-0013	Tibiap	Stratec XCT 3000	61.3	0.203	100,000	0.137	0.1	This study	3	
Mammalia, Hominidae	Homo sapiens	GU S-0013	Tibiad	Stratec XCT 3000	61.3	0.203	100,000	0.137	0.1	This study	3	
Mammalia, Hominidae	Homo sapiens	GU S-0013	Tibias	Stratec XCT 3000	61.3	0.203	100,000	0.137	0.1	This study	3	
Mammalia, Hominidae	Homo sapiens	GU S-0013	Fibulap	Stratec XCT 3000	61.3	0.203	100,000	0.137	0.1	This study	3	
Mammalia, Hominidae	Homo sapiens	GU S-0013	Fibulad	Stratec XCT 3000	61.3	0.203	100,000	0.137	0.1	This study	3	
Sauria, Squamata	Varanus komodoensis	AM R.106933	Femur	Siemens Inveon	80	0.45	1,000	0.053	0.053	This study	1	
Sauria, Squamata	Varanus komodoensis	AM R.106933	Tibia	Siemens Inveon	80	0.45	1,000	0.053	0.053	This study	1	
Sauria, Squamata	Varanus komodoensis	AM R.106933	Fibula	Siemens Inveon	80	0.45	1,000	0.053	0.053	This study	1	
Sauria, Squamata	Varanus spenceri	QMJ 84416	Femur	Siemens Inveon	80	0.5	900	0.053	0.053	This study	1	
Sauria, Squamata	Varanus spenceri	QMJ 84416	Tibia	Siemens Inveon	80	0.5	900	0.053	0.053	This study	1	
Sauria, Squamata	Varanus spenceri	QMJ 84416	Fibula	Siemens Inveon	80	0.5	900	0.053	0.053	This study	1	
Sauria, Squamata	Varanus panoptes	QMJ 91981	Femur	Siemens Inveon	80	0.5	900	0.053	0.053	This study	1	
Sauria, Squamata	Varanus panoptes	QMJ 91981	Tibia	Siemens Inveon	80	0.5	900	0.053	0.053	This study	1	
Sauria, Squamata	Varanus panoptes	QMJ 91981	Fibula	Siemens Inveon	80	0.5	900	0.053	0.053	This study	1	
Archosauria, Crocodylia	Crocodylus johnstoni	QMJ 47916	Femur	Siemens Inveon	80	0.5	900	0.053	0.053	This study	1	
Archosauria, Crocodylia	Crocodylus johnstoni	QMJ 47916	Tibia	Siemens Inveon	80	0.5	900	0.053	0.053	This study	1	
Archosauria, Crocodylia	Crocodylus johnstoni	QMJ 47916	Fibula	Siemens Inveon	80	0.5	900	0.053	0.053	This study	1	
Archosauria, Crocodylia	Crocodylus porosus	QMJ 48127	Femur	Siemens Inveon	80	0.45	1,000	0.053	0.053	This study	1	
Archosauria, Crocodylia	Crocodylus porosus	QMJ 48127	Tibia	Siemens Inveon	80	0.45	1,000	0.053	0.053	This study	1	
Archosauria, Crocodylia	Crocodylus porosus	QMJ 48127	Fibula	Siemens Inveon	80	0.45	1,000	0.053	0.053	This study	1	
Non-avian theropod, Ceratosauria	Ceratosaurus nasicornis	UMNH VP 5278	Tibia + astragalus + calcaneum	Siemens Somatom Definition Flash	80, 140	480	500	0.504	0.5	This study	5	
Non-avian theropod, Ceratosauria	Masiakasaurus knopfleri	FMNH PR 2117	Femur	GE Lightspeed 16	100	70	1,297	0.1875	1	Farke & Alicea (2009)	2	
Non-avian theropod, Ceratosauria	Masiakasaurus knopfleri	FMNH PR 2153	Femur	GE Lightspeed 16	100	100	2,101	0.1875	1.338	Farke & Alicea (2009)	2	
Non-avian theropod, Ceratosauria	Masiakasaurus knopfleri	FMNH PR 2208	Femur	GE Lightspeed 16	100	70	1,297	0.1875	1	Farke & Alicea (2009)	2	
Non-avian theropod, Ceratosauria	Masiakasaurus knopfleri	UA 8684	Femur	GE Lightspeed 16	100	70	1,297	0.1875	1	Farke & Alicea (2009)	2	
Non-avian theropod, Allosauroidea	Allosaurus sp.	MOR 693	Femora × 2	Toshiba Aquilion 64	135	250	750	0.625	0.5	This study	5	
Non-avian theropod, Allosauroidea	Allosaurus sp.	MOR 693	Tibiae × 2	Toshiba Aquilion 64	135	300	750	0.468	0.4	This study	5	
Non-avian theropod, Allosauroidea	Allosaurus sp.	MOR 693	Fibulae × 2	Toshiba Aquilion 64	135	250	750	0.625	0.5	This study	5	
Non-avian theropod, Allosauroidea	Allosaurus sp.	MOR 693	Astragalus	Toshiba Aquilion 64	135	250	750	0.625	0.5	This study	5	
Non-avian theropod, Allosauroidea	Allosaurus sp.	MOR 693	Calcaneum	Toshiba Aquilion 64	135	250	750	0.625	0.5	This study	5	
Non-avian theropod, Allosauroidea	Allosaurus fragilis	DNM 2560	Femur	Siemens Somatom Definition Flash	80, 140	315	500	0.549	0.5	This study	5	
Non-avian theropod, Allosauroidea	Allosaurus fragilis	DNM 2560	Tibia + astragalus + calcaneum	Siemens Somatom Definition Flash	80, 140	360	500	0.637	0.5	This study	5	
Non-avian theropod, Allosauroidea	Allosaurus fragilis	UMNH VP 7884	Femur	Siemens Somatom Definition Flash	80, 140	630	500	0.529	0.5	This study	5	
Non-avian theropod, Allosauroidea	Allosaurus fragilis	UMNH VP 7885	Femur	Siemens Somatom Definition Flash	80, 140	225	500	0.387	0.5	This study	5	
Non-avian theropod, Allosauroidea	Allosaurus fragilis	UMNH VP 7889	Femurp	Siemens Somatom Definition Flash	80, 140	630	500	0.523	0.5	This study	5	
Non-avian theropod, Allosauroidea	Allosaurus fragilis	UMNH VP 7928	Tibia	Siemens Somatom Definition Flash	80, 140	630	500	0.355	0.5	This study	5	
Non-avian theropod, Allosauroidea	Allosaurus fragilis	UMNH VP 9480	Femur	Siemens Somatom Definition Flash	80, 140	80	500	0.217	0.5	This study	5	
Non-avian theropod, Allosauroidea	Allosaurus fragilis	UMNH VP 20363	Femur	Siemens Somatom Definition Flash	80, 140	315	500	0.664	0.5	This study	5	
Non-avian theropod, Allosauroidea	Allosaurus fragilis	UMNH VP 24326	Tibiap	Siemens Somatom Definition Flash	80, 140	200	500	0.459	0.5	This study	5	
Non-avian theropod, Tyrannosauridae	Tyrannosaurus rex	MOR 009	Femurp	Toshiba Aquilion 64	135	500	500	0.782	0.4	This study	5	
Non-avian theropod, Tyrannosauridae	Tyrannosaurus rex	MOR 1125	Femur	Toshiba Aquilion 64	135	350	500	1.178	0.5	This study	5	
Non-avian theropod, Tyrannosauridae	Tyrannosaurus rex	MOR 1125	Tibia	Toshiba Aquilion 64	135	350	500	0.976	0.5	This study	5	
Non-avian theropod, Tyrannosauridae	Tyrannosaurus rex	MOR 1125	Fibulae × 2	Toshiba Aquilion 64	135	350	500	0.873	0.8	This study	5	
Non-avian theropod, Tyrannosauridae	Tyrannosaurus rex	MOR 1128	Femur	Toshiba Aquilion 64	135	350	500	0.976	2	This study	5	
Non-avian theropod, Tyrannosauridae	Tyrannosaurus rex	MOR 1128	Tibiad	Toshiba Aquilion 64	135	350	500	0.961	0.5	This study	5	
Non-avian theropod, Tyrannosauridae	Tyrannosauridae indet.	MOR 1192	Fibula	Toshiba Aquilion 64	135	150	1,000	0.976	2	This study	5	
Non-avian theropod, Tyrannosauridae	Gorgosaurus libratus	TMP 1994.012.0602	Femur	GE Lightspeed Ultra	120	160	1,195	0.723	1.25	This study	5	
Non-avian theropod, Tyrannosauridae	Gorgosaurus libratus	TMP 1994.012.0602	Tibia + astragalus + calcaneum	GE Lightspeed Ultra	140	150	1,195	0.703	1.25	This study	5	
Non-avian theropod, Tyrannosauridae	Daspletosaurus torosus	TMP 2001.036.0001	Femur	GE Lightspeed Ultra	140	150	1,195	0.838	1.25	This study	5	
Non-avian theropod, Tyrannosauridae	Daspletosaurus torosus	TMP 2001.036.0001	Tibia	GE Lightspeed Ultra	120	245	1,195	0.832	1.25	This study	5	
Non-avian theropod, Tyrannosauridae	Daspletosaurus torosus	TMP 2001.036.0001	Fibula	GE Lightspeed Ultra	120	245	1,195	0.832	1.25	This study	5	
Non-avian theropod, Tyrannosauridae	Daspletosaurus torosus	TMP 2001.036.0001	Astragalus	GE Lightspeed Ultra	140	155	1,195	0.879	1.25	This study	5	
Non-avian theropod, Ornithomimidae	Ornithomimidae indet.	TMP 1985.036.0276	Femurp	Siemens Inveon	80	500	825	0.05	0.05	This study	4	
Non-avian theropod, Ornithomimidae	Ornithomimidae indet.	TMP 1991.036.0569	Femur	Siemens Inveon	80	250	1,500	0.05	0.05	This study	4	
Non-avian theropod, Ornithomimidae	Ornithomimidae indet.	TMP 1991.036.0854	Femurp	GE Lightspeed Ultra	140	150	1,195	0.943	1.25	This study	5	
Non-avian theropod, Ornithomimidae	Ornithomimidae indet.	TMP 1992.036.0696	Femurp	GE Lightspeed Ultra	140	150	1,195	0.943	1.25	This study	5	
Non-avian theropod, Ornithomimidae	Ornithomimidae indet.	TMP 1993.066.0002	Tibiap	GE Lightspeed Ultra	140	155	1,195	0.879	1.25	This study	5	
Non-avian theropod, Ornithomimidae	Ornithomimidae indet.	TMP 1999.055.0337	Femurd	Siemens Inveon	80	250	1,500	0.05	0.05	This study	4	
Non-avian theropod, Ornithomimidae	Ornithomimidae indet.	TMP 2006.012.0065	Fibula	GE Lightspeed Ultra	120	185	1,195	0.738	1.25	This study	5	
Non-avian theropod, Therizinosauria	Falcarius utahensis	UMNH VP 12360	Femurd	Siemens Inveon	80	250	1,600	0.05	0.05	This study	4	
Non-avian theropod, Therizinosauria	Falcarius utahensis	UMNH VP 12361	Femurp	Siemens Inveon	80	250	1,700	0.05	0.05	This study	4	
Non-avian theropod, Caenagnathidae	Caenagnathidae indet.	TMP 1986.036.0323	Femur	Siemens Inveon	80	250	1,600	0.05	0.05	This study	4	
Non-avian theropod, Dromaeosauridae	Saurornitholestes langstoni	MOR 660	Tibiae × 2	Siemens Inveon	80	250	1,600	0.05	0.05	This study	4	
Non-avian theropod, Troodontidae	Troodontidae sp.	MOR 553s-7.11.91.41	Tibia	Siemens Inveon	80	200	1,900	0.04	0.04	This study	4	
Non-avian theropod, Troodontidae	Troodontidae sp.	MOR 553s-7.28.91.239	Femur	Siemens Inveon	80	200	1,800	0.04	0.04	This study	4	
Non-avian theropod, Troodontidae	Troodontidae sp.	MOR 553s-8.17.92.265	Fibula	Siemens Inveon	80	250	1,600	0.04	0.04	This study	4	
Non-avian theropod, Troodontidae	Troodontidae sp.	MOR 748	Femur	Siemens Inveon	80	200	1,900	0.04	0.04	This study	4	
Non-avian theropod, Troodontidae	Troodontidae sp.	MOR 748	Tibia + astragalus + calcaneum	Siemens Inveon	80	200	1,900	0.04	0.04	This study	4	
Aves, Struthioniformes	Struthio camelus	MV R.2385	Femur	GE BrightSpeed	120	55	1,681	0.488	0.3	This study	2	
Aves, Struthioniformes	Struthio camelus	MV R.2385	Tibiotarsus	Siemens Somatom Definition AS+	120	199	1,000	0.363	0.4	This study	2	
Aves, Struthioniformes	Struthio camelus	MV R.2385	Fibula	GE BrightSpeed	120	55	1,681	0.488	0.3	This study	2	
Aves, Struthioniformes	Struthio camelus	MV R.2711	Femur	GE BrightSpeed	120	55	1,681	0.488	0.3	This study	2	
Aves, Struthioniformes	Struthio camelus	MV R.2711	Tibiotarsus	GE BrightSpeed	120	55	1,681	0.488	0.3	This study	2	
Aves, Struthioniformes	Struthio camelus	MV R.2711	Fibula	GE BrightSpeed	120	55	1,681	0.488	0.3	This study	2	
Aves, Struthioniformes	Struthio camelus	YPM 2124	Femur	GE Lightspeed 16	100	70	1,297	0.311	1.25	Farke & Alicea (2009)	None**	
Aves, Struthioniformes	Struthio camelus	RVC Ostrich 2	Femur	Picker PQ5000	120	200	1,000	0.391	2	Doube et al. (2012)	2	
Aves, Struthioniformes	Struthio camelus	RVC Ostrich 2	Tibiotarsus	Picker PQ5000	120	200	1,000	0.313	4	Doube et al. (2012)	2	
Aves, Struthioniformes	Struthio camelus	RVC Ostrich 2	Fibula	Picker PQ5000	120	200	1,000	0.313	4	Doube et al. (2012)	2	
Aves, Struthioniformes	Struthio camelus	RVC-JRH-OST 1	Femur	GE LightSpeed Pro 16	120	200	800	0.273	0.625	This study	2	
Aves, Struthioniformes	Struthio camelus	RVC-JRH-OST 1	Tibiotarsus	GE LightSpeed Pro 16	120	200	800	0.369	1.25	This study	2	
Aves, Struthioniformes	Struthio camelus	RVC-JRH-OST 1	Fibula	GE LightSpeed Pro 16	120	200	800	0.369	1.25	This study	2	
Aves, Rheiformes	Rhea americana	QMO 23517	Femur	GE BrightSpeed	120	80	1,584	0.488	0.3	This study	2	
Aves, Rheiformes	Rhea americana	QMO 23517	Tibiotarsus	GE BrightSpeed	120	80	1,584	0.488	0.3	This study	2	
Aves, Rheiformes	Rhea americana	QMO 23517	Fibula	GE BrightSpeed	120	80	1,584	0.488	0.3	This study	2	
Aves, Tinamiformes	Crypturellus soui	MVB 23647	Femur	Siemens Inveon	80	0.35	1400	0.035	0.035	This study	1	
Aves, Tinamiformes	Crypturellus soui	MVB 23647	Tibiotarsus	Siemens Inveon	80	0.35	1,400	0.035	0.035	This study	1	
Aves, Tinamiformes	Crypturellus soui	MVB 23647	Fibula	Siemens Inveon	80	0.35	1,400	0.035	0.035	This study	1	
Aves, Tinamiformes	Eudromia elegans	UMZC 404.e	Femur	Nikon HMX ST 225				0.034	0.034	Doube et al. (2012)	1	
Aves, Tinamiformes	Eudromia elegans	UMZC 404.e	Tibiotarsus	Nikon HMX ST 225				0.047	0.047	Doube et al. (2012)	1	
Aves, Tinamiformes	Eudromia elegans	UMZC 404.e	Fibula	Nikon HMX ST 225				0.047	0.047	Doube et al. (2012)	1	
Aves, Apterygiformes	Apteryx owenii	UMZC 378.iii	Femur	Nikon HMX ST 225				0.046	0.046	Doube et al. (2012)	1	
Aves, Apterygiformes	Apteryx owenii	UMZC 378.iii	Tibiotarsus	Nikon HMX ST 225				0.061	0.061	Doube et al. (2012)	1	
Aves, Apterygiformes	Apteryx owenii	UMZC 378.iii	Fibula	Nikon HMX ST 225				0.061	0.061	Doube et al. (2012)	1	
Aves, Apterygiformes	Apteryx haastii	UMZC 378.p	Femur	Nikon HMX ST 225				0.044	0.044	Doube et al. (2012)	1	
Aves, Casuariiformes	Dromaius novaehollandiae	QMO 11685	Femur	GE BrightSpeed	120	55	1,681	0.379	0.3	This study	2	
Aves, Casuariiformes	Dromaius novaehollandiae	QMO 11686	Femur	GE BrightSpeed	120	80	1,584	0.326	0.3	This study	2	
Aves, Casuariiformes	Dromaius novaehollandiae	QMO 11686	Tibiotarsus	GE BrightSpeed	120	55	1,681	0.488	0.3	This study	2	
Aves, Casuariiformes	Dromaius novaehollandiae	QMO 11686	Fibula	GE BrightSpeed	120	55	1,681	0.326	0.3	This study	2	
Aves, Casuariiformes	Dromaius novaehollandiae	QMO 16140	Femur	GE BrightSpeed	120	55	1,681	0.232	0.3	This study	2	
Aves, Casuariiformes	Dromaius novaehollandiae	QMO 16140	Tibiotarsus	GE BrightSpeed	120	55	1,681	0.232	0.3	This study	2	
Aves, Casuariiformes	Dromaius novaehollandiae	QMO 16140	Fibula	GE BrightSpeed	120	55	1,681	0.188	0.3	This study	2	
Aves, Casuariiformes	Casuarius casuarius	QMO 30105	Femur	GE BrightSpeed	120	55	1,681	0.215	0.3	This study	2	
Aves, Casuariiformes	Casuarius casuarius	QMO 30105	Tibiotarsus	GE BrightSpeed	120	55	1,681	0.219	0.3	This study	2	
Aves, Casuariiformes	Casuarius casuarius	QMO 30105	Fibula	GE BrightSpeed	120	55	1,681	0.215	0.3	This study	2	
Aves, Casuariiformes	Casuarius casuarius	QMO 30604	Femur	GE BrightSpeed	120	80	1,584	0.467	0.3	This study	2	
Aves, Casuariiformes	Casuarius casuarius	QMO 30604	Tibiotarsus	GE BrightSpeed	120	80	1,584	0.488	0.3	This study	2	
Aves, Casuariiformes	Casuarius casuarius	QMO 30604	Fibula	GE BrightSpeed	120	80	1,584	0.488	0.3	This study	2	
Aves, Casuariiformes	Casuarius casuarius	QMO 31137	Femur	GE BrightSpeed	120	55	1,681	0.213	0.3	This study	2	
Aves, Casuariiformes	Casuarius casuarius	QMO 31137	Tibiotarsus	GE BrightSpeed	120	80	1,584	0.488	0.3	This study	2	
Aves, Casuariiformes	Casuarius casuarius	QMO 31137	Fibula	GE BrightSpeed	120	80	1,584	0.488	0.3	This study	2	
Aves, Casuariiformes	Dromaius novaehollandiae	YPM 2128	Femur	GE Lightspeed 16	100	70	1,297	0.188	0.5	Farke & Alicea (2009)	None**	
Aves, Galliformes	Alectura lathami	PJB	Femur	Siemens Inveon	80	0.45	1,000	0.053	0.053	This study	1	
Aves, Galliformes	Alectura lathami	PJB	Tibiotarsus	Siemens Inveon	80	0.45	1,000	0.053	0.053	This study	1	
Aves, Galliformes	Alectura lathami	PJB	Fibula	Siemens Inveon	80	0.45	1,000	0.053	0.053	This study	1	
Aves, Galliformes	Leipoa ocellata	MVB 20194	Femur	Siemens Inveon	80	0.45	1,000	0.053	0.053	This study	1	
Aves, Galliformes	Leipoa ocellata	MVB 20194	Tibiotarsus	Siemens Inveon	80	0.45	1,000	0.053	0.053	This study	1	
Aves, Galliformes	Leipoa ocellata	MVB 20194	Fibula	Siemens Inveon	80	0.45	1,000	0.053	0.053	This study	1	
Aves, Galliformes	Numida meleagris	PJB	Femur	Siemens Inveon	80	0.45	1,000	0.053	0.053	This study	1	
Aves, Galliformes	Numida meleagris	PJB	Tibiotarsus	Siemens Inveon	80	0.45	1,000	0.053	0.053	This study	1	
Aves, Galliformes	Numida meleagris	PJB	Fibula	Siemens Inveon	80	0.45	1,000	0.053	0.053	This study	1	
Aves, Galliformes	Colinus virginianus	PJB	Femur	Siemens Inveon	80	0.35	1,400	0.035	0.035	This study	1	
Aves, Galliformes	Colinus virginianus	PJB	Tibiotarsus	Siemens Inveon	80	0.45	1,000	0.053	0.053	This study	1	
Aves, Galliformes	Colinus virginianus	PJB	Fibula	Siemens Inveon	80	0.45	1,000	0.053	0.053	This study	1	
Aves, Galliformes	Coturnix chinensis	PJB	Femur	Siemens Inveon	80	0.35	1,400	0.035	0.035	This study	1	
Aves, Galliformes	Coturnix chinensis	PJB	Tibiotarsus	Siemens Inveon	80	0.35	1,400	0.035	0.035	This study	1	
Aves, Galliformes	Coturnix chinensis	PJB	Fibula	Siemens Inveon	80	0.35	1,400	0.035	0.035	This study	1	
Aves, Galliformes	Coturnix japonica	PJB	Femur	Siemens Inveon	80	0.35	1,400	0.035	0.035	This study	1	
Aves, Galliformes	Coturnix japonica	PJB	Tibiotarsus	Siemens Inveon	80	0.45	1,000	0.053	0.053	This study	1	
Aves, Galliformes	Coturnix japonica	PJB	Fibula	Siemens Inveon	80	0.45	1,000	0.053	0.053	This study	1	
Aves, Galliformes	Gallus gallus	PJB	Femur	Siemens Inveon	80	0.45	1,000	0.053	0.053	This study	1	
Aves, Galliformes	Gallus gallus	PJB	Tibiotarsus	Siemens Inveon	80	0.45	1,000	0.053	0.053	This study	1	
Aves, Galliformes	Gallus gallus	PJB	Fibula	Siemens Inveon	80	0.45	1,000	0.053	0.053	This study	1	
Aves, Galliformes	Gallus gallus	PJB	Femur	Siemens Inveon	80	0.45	1,000	0.053	0.053	This study	1 (for head only)	
Aves, Galliformes	Meleagris gallopavo	PJB	Femur	Siemens Inveon	80	0.45	1,000	0.053	0.053	This study	1	
Aves, Galliformes	Meleagris gallopavo	PJB	Tibiotarsus	Siemens Inveon	80	0.45	1,000	0.053	0.053	This study	1	
Aves, Galliformes	Meleagris gallopavo	PJB	Fibula	Siemens Inveon	80	0.45	1,000	0.053	0.053	This study	1	
Aves, Galliformes	Meleagris gallopavo	RVC turkey 1	Tibiotarsus	Nikon HMX ST 225				0.122	0.122	Doube et al. (2012)	1	
Aves, Galliformes	Meleagris gallopavo	RVC turkey 1	Fibula	Nikon HMX ST 225				0.122	0.122	Doube et al. (2012)	1	
Aves, Galliformes	Meleagris gallopavo	YPM 2113	Femur	GE Lightspeed 16	100	70	1,297	0.188	0.5	Farke & Alicea (2009)	None**	
Aves, Galliformes	Argusianus argus	YPM 2100	Femur	GE Lightspeed 16	100	70	1,297	0.188	0.5	Farke & Alicea (2009)	None**	
Aves, Anseriformes	Anseranus semipalmata	QMO 29529	Femur	Siemens Inveon	80	0.45	1,000	0.053	0.053	This study	1	
Aves, Anseriformes	Anseranus semipalmata	QMO 29529	Tibiotarsus	Siemens Inveon	80	0.45	1,000	0.053	0.053	This study	1	
Aves, Anseriformes	Anseranus semipalmata	QMO 29529	Fibula	Siemens Inveon	80	0.45	1,000	0.053	0.053	This study	1	
Aves, Otidiformes	Ardeotis australis	MVB 20408	Femur	Siemens Inveon	80	0.45	1,000	0.053	0.053	This study	1	
Aves, Otidiformes	Ardeotis australis	MVB 20408	Tibiotarsus	Siemens Inveon	80	0.45	1,000	0.053	0.053	This study	1	
Aves, Otidiformes	Ardeotis australis	MVB 20408	Fibula	Siemens Inveon	80	0.45	1,000	0.053	0.053	This study	1	
Aves, Gruiformes	Porphyrio porphyrio	PJB	Femur	Siemens Inveon	80	0.45	1,000	0.053	0.053	This study	1	
Aves, Gruiformes	Porphyrio porphyrio	PJB	Tibiotarsus	Siemens Inveon	80	0.45	1,000	0.053	0.053	This study	1	
Aves, Gruiformes	Porphyrio porphyrio	PJB	Fibula	Siemens Inveon	80	0.45	1,000	0.053	0.053	This study	1	
Aves, Gruiformes	Gallinula tenebrosa	PJB	Femur	Siemens Inveon	80	0.45	1,000	0.053	0.053	This study	1	
Aves, Gruiformes	Gallinula tenebrosa	PJB	Tibiotarsus	Siemens Inveon	80	0.45	1,000	0.053	0.053	This study	1	
Aves, Gruiformes	Gallinula tenebrosa	PJB	Fibula	Siemens Inveon	80	0.45	1,000	0.053	0.053	This study	1	
Aves, Pelecaniformes	Threskiornis moluccus	PJB	Femur	Siemens Inveon	80	0.45	1,000	0.053	0.053	This study	1	
Aves, Pelecaniformes	Threskiornis moluccus	PJB	Tibiotarsus	Siemens Inveon	80	0.45	1,000	0.053	0.053	This study	1	
Aves, Pelecaniformes	Threskiornis moluccus	PJB	Fibula	Siemens Inveon	80	0.45	1,000	0.053	0.053	This study	1	
Aves, Cuculiformes	Geococcyx californianus	UMZC 429.p	Femur	Nikon HMX ST 225				0.032	0.032	Doube et al. (2012)	1	
Aves, Cuculiformes	Geococcyx californianus	UMZC 429.p	Tibiotarsus	Nikon HMX ST 225				0.049	0.049	Doube et al. (2012)	1	
Aves, Cuculiformes	Geococcyx californianus	UMZC 429.p	Fibula	Nikon HMX ST 225				0.049	0.049	Doube et al. (2012)	1	
Aves, Columbiformes	Pezophaps solitaria	YPM 1154	Femur	GE Lightspeed 16	120	70	1,297	0.188	0.5	Farke & Alicea (2009)	None**	
Aves, Columbiformes	Raphus cucullatus	YPM 2064	Femur	GE Lightspeed 16	120	70	1,297	0.188	0.5	Farke & Alicea (2009)	None**	
Aves, Accipitriformes	Sagittarius serpentarius	YPM 1797	Femur	GE Lightspeed 16	100	70	1,297	0.188	0.5	Farke & Alicea (2009)	None**	
Aves, Accipitriformes	Sagittarius serpentarius	YPM 14150	Femur	GE Lightspeed 16	120	70	1,297	0.188	0.5	Farke & Alicea (2009)	None**	
Notes:

In addition to providing the higher-level taxonomy of the species studied (cf. Fig. 8), this table also lists the settings used in acquiring the CT scan data for each specimen, as well as the protocol used to process the raw CT images for subsequent analysis.

* AM R., Australian Museum; DNM, UMNH VP; Natural History Museum of Utah; FMNH PR, Field Museum of Natural History; GU, Griffith University teaching collection; MOR, Museum of the Rockies; MVB, MV R., Museum Victoria; PJB, P.J.B. personal collection (housed within Queensland Museum collections); QMJ, QMO, Queensland Museum; RVC, Royal Veterinary College; TMP, Royal Tyrrell Museum of Palaeontology; UA, Université d’Antananarivo; UMZC, Cambridge University Museum of Zoology; YPM, Yale Peabody Museum of Natural History.

** The scans of these specimens were qualitatively analysed only, and did not require any further processing.

p = proximal end, d = distal end, s = shaft.

All scripts and data used are held in the Geosciences Collection of the Queensland Museum, and are available upon request to the Collections Manager. Additionally, a complete copy of the fossil CT scan data is accessioned with the respective museums in which the specimens are housed (see Table 1).

Data acquisition

This study focused on the main bones of the hindlimb, the femur, tibia (tibiotarsus5) and fibula, in a variety of extant avian and non-avian theropods, as well as extant reptilian species, crocodilians and lizards (Table 1; institutional abbreviations for museum specimens are also detailed here). In addition, data were collected for a representative human specimen, to provide further comparative context; this was an adult male specimen used for teaching purposes, which showed no apparent pathologies. A schematic illustration of the higher-level phylogenetic relationships of the study species is given in Fig. 8.

Figure 8 Phylogenetic interrelationships of the major groups of animals studied.

Interrelationships of non-avian theropods are based on Zanno et al. (2009), Carrano, Benson & Sampson (2012) and Turner, Makovicky & Norell (2012); interrelationships of birds are based on Hackett et al. (2008), Morgan-Richards et al. (2008), Ksepka (2009), Phillips et al. (2010), Haddrath & Baker (2012), Jetz et al. (2012), Smith, Braun & Kimball (2013), Yuri et al. (2013), Mitchell et al. (2014), Jarvis et al. (2014), Ksepka & Phillips (2015) and Prum et al. (2015). The interrelationships of the neoavian species studied here are currently not well agreed upon, and so Neoaves is shown as an unresolved polytomy. Silhouettes depict exemplar members or each group, and are not to scale. The individual determinate species of non-avian theropod studied are listed in the inset box, and their phylogenetic position signified by superscripts.

For the fossil specimens, careful inspection was undertaken to ensure that the best-preserved and most complete available specimens were studied, avoiding bones, or regions thereof, that had evidence of taphonomic deformation (see also below). Only primarily ground-dwelling species of birds were investigated because, by virtue of spending most or all of their time on the ground, they have well-developed hindlimb locomotor systems. Where possible, only male bird specimens were chosen, so as to preclude the possibility of medullary bone being present and influencing the results (Dacke et al., 1993; but see below). Crocodilians were chosen as they represent the closest extant outgroup of Theropoda. Varanids (monitor lizards) were chosen to represent squamates because most are highly terrestrial species, and their large size better facilitates analysis of cancellous bone architecture compared to smaller species. Generally, only one or two individuals were sampled for a given species. As this study is the first detailed survey of cancellous bone architecture in theropods, and more broadly, saurians, preference was given to maximizing the number of species investigated, rather than achieving a larger number of samples for fewer species, in order to elucidate any general, broad-scale patterns that were present. With sample sizes of usually n = 1 or 2, it was hence not possible (or meaningful) to quantify potential intraspecific variation in bone architecture, nor was it possible to examine finer-scale patterns as may occur in association with more subtle differences in anatomy, behaviour or habitat.

The 3D cancellous bone architectural data was acquired through CT scanning of the limb bones. Scanning parameters varied depending on the size and bulk density of the specimen and the locally available scanning machine (details given in Table 1). In all cases, the highest possible resolution and contrast between bone and non-bone phases was sought. Nevertheless, as a consequence of using various scanning modalities on different-sized bones, the resulting scans had a range of absolute and relative (to bone dimensions) voxel resolutions. This in turn required that multiple techniques for image data processing and analysis were used, as detailed below (sections ‘Image data processing’, ‘Architectural analyses’). Regarding the non-avian theropods, more than 40 additional fossil specimens (including of other species) were scanned throughout the course of this study, but owing to various factors (e.g. high density, low contrast between bone and matrix, mineral-induced scanning artefacts, insufficient scanning power or resolution) their resulting scans were not useful and thus excluded. In addition to the data collected in the present study, data collected by previous studies were also used (Doube et al., 2012; Farke & Alicea, 2009).

Image data processing

The CT scans of each bone were processed using the software ImageJ 1.47 (http://imagej.nih.gov/ij/) and Mimics 17.0 (Materialize NV, Belgium), so as to segment the bone from the non-bone phases. All image processing was undertaken by a single person (PJB). Five different image processing protocols were used, depending on differences in the specimens and how they were scanned (Table 1):

Protocol 1—extant animal micro-CT scans (∼70 bones)

Scans were segmented using the local thresholding algorithm of Bernsen (1986), as implemented in ImageJ (Landini, 2008; Landini et al., 2017). The window radius was set to a low value (typically on the order of five pixels), whilst the contrast threshold was set to a fairly high value (typically in the range of 20–50), so as to reduce the possibility of relatively high density non-bone material (e.g. dried marrow tissue) from becoming segmented along with the bone phase. The specific values for each parameter varied from specimen to specimen, and were chosen based on iterative visual comparison of the segmented vs. unsegmented scans. Subsequent to image segmentation, the scans were ‘cleaned’ in Mimics with the ‘region growing tool’, which removed extraneous matter that was included in the segmentation but was not connected to the cancellous bone network. That is, this process removed isolated (‘floating’) voxels that were not connected to any adjacent voxels in three dimensions; since cancellous bone is always connected to other bone material, this step did not cause any loss of cancellous bone material (but did remove voxels pertaining to marrow tissue, for example).

One extant bird femur (Gallus gallus) was found to possess material that was possibly medullary bone; it was processed according to Protocol 1 but only for the femoral head region, where the unidentified material essentially filled all the intertrabecular space and thus was able to be effectively isolated from the cancellous bone. Only the femoral head region was subsequently analysed in this particular specimen.

Protocol 2—extant animal medical CT scans (∼35 bones)

This protocol differs from protocol 1 only in that an additional step was undertaken prior to segmentation. As these scans were acquired with a medical-grade machine, the original dataset was comprised of anisotropic voxels (slice thickness differed from in-plane pixel resolution), which is not ideal for further architectural analyses. Moreover, the scans had a markedly lower resolution compared to the micro-CT scans. As a result, protocol 1 would not result in accurate segmentation of the cancellous structure. To negotiate this problem, the scans were first resampled to triple the original in-plane pixel resolution, and simultaneously resampled along the axial direction to produce isotropic voxels. The axial resampling factor is given as 3f, where (1) f=Slice thicknessPixel resolution.

This was performed in ImageJ using 3D bicubic interpolation, after which the scans were processed according to protocol 1.

The above process of resampling the CT scans is simply to facilitate a more complete and accurate extraction of cancellous bone, without altering the underlying structure in the scan data (Fig. 9). Despite a relatively low resolution to the scans, each individual trabecula was still visible to the naked eye, owing to the partial volume effect (Ketcham & Carlson, 2001; Ryan & Ketcham, 2002b). Here, pixels with intermediate grey-values reflect a volumetric averaging of the high grey-values of bone and the low grey-values of intertrabecular spaces, and are interpreted by the observer as reflecting intertrabecular spaces. However, as these intermediate-valued pixels are adjacent to high-valued (bone) pixels, they may not be recognized as reflecting non-bone material by a local segmentation algorithm because of insufficient contrast, and so become included in the segmented bone phase, producing an erroneous result. By resampling the CT scans to a higher resolution, the underlying structure in the data is retained, but the number of pixels associated with each phase (bone and non-bone) of each part of the structure is increased (Fig. 9C). This increases the ability of a local thresholding algorithm to distinguish the two phases from each other, as there is higher local contrast, resulting in a more accurate segmentation. Theoretically, the greater the degree to which the scan is resampled, the higher the accuracy of the resulting segmentation. However, there will be a limit to the degree of resampling, beyond which any further information extracted is not genuine, and is artefactual; furthermore, greater resampling produces larger scan datasets, increasing computational requirements for all successive steps in image processing and analysis. Experience has shown that, with the current scan dataset at least, a resampling factor of three was sufficient to achieve accurate segmentation.

Figure 9 Resampling the medical CT scans, as per protocol 2, helps produce a more accurate segmentation of the cancellous bone structure.

This is illustrated here with a CT slice of the proximal femur of a southern cassowary, Casuarius casuarius (QMO 31137). (A) The raw CT slice. (B) A local segmentation algorithm, as per protocol 1, is applied to the slice in (A), with window radius eight pixels and contrast threshold of 25; notice that it performs poorly, with some trabeculae becoming disconnected and some intertrabecular spaces being obliterated. (C) The same CT slice as in (A), but resampled to triple the resolution, that is, voxel dimensions are now ⅓ of original in-plane pixel resolution. (D) A local segmentation is applied to the resampled slice in (C), using the same parameters as in (B); notice that it performs far better in extracting the cancellous structure. Bicubic interpolation results in smoother boundaries to the segmented scans, in contrast to lower-order (e.g. bilinear) interpolation. The location of the insets in each panel is shown in (C).

Protocol 3—human CT scans (three bones)

The human bone scans were collected using peripheral quantitative CT scanning; as with the medical CT scans, the resulting scans often comprised anisotropic voxels. Since the in-plane resolution was always lower than the slice thickness (if the two were not equal), the scans were first resampled in the in-plane directions using bicubic interpolation in ImageJ, resulting in an isotropic image stack. Next, a low-radius 3D median filter of kernel radius two or three pixels was applied, again in ImageJ, to remove high-frequency noise in the data (Ollion et al., 2013). Finally, the steps outlined in protocol 1 were followed.

Protocol 4—non-avian theropod micro-CT scans (15 bones)

Owing to a maximum capable peak tube voltage of 80 kV in the machine used, the resulting scans of the fossil bones suffered from both high-frequency (i.e. short wavelength) and low frequency (i.e. long wavelength) background noise that overprinted the actual structure in the images. The former was manifest as speckled, ‘salt and pepper’ patterns, whilst the latter stemmed from scanning artefacts such as beam hardening (Ketcham & Carlson, 2001) and differences in bulk density throughout a specimen. These two sources of noise were removed in ImageJ in the following manner, and the process is illustrated in Fig. 10. First, a low-radius 3D median filter was applied to the original image stack, removing the high frequency noise; the kernel radius used varied between 2 and 10 pixels, depending on the specimen. Secondly, a large-radius 2D median filter was successively applied to the original image stack in all three directions (x, y and z), to isolate the low-frequency noise; the kernel radius used was large in comparison to the trabecular thickness, and varied between 10 and 40 pixels, depending on the specimen. A single application of a large-radius 3D median filter could have been used instead to obtain the same effect, but owing to the larger number of calculations required and large kernel radius, it was prohibitively slow. The low-frequency filtered image stack was then subtracted from the high-frequency filtered stack. That is, this process removes the high-frequency noise, and then removes the background noise. Whilst the resulting image stack was devoid of noise and captured the structure in the data, it nevertheless was texturally ‘rough.’ Consequently, a low-radius 3D mean filter, of kernel radius two or three pixels, was applied to remove this roughness. As this filter was of a high frequency compared to the structure itself, this step did not alter the structure in the data (e.g. trabeculae were not removed, connections did not disappear). Lastly, a global segmentation, with a high-pass grey-value threshold set to 1 or 2, was applied to extract the cancellous bone structure.

Figure 10 Schematic illustration of the image processing protocol used for the non-avian theropod micro-CT scans.

(A) The original image, affected by both high and low frequency noise; segmentation of this image by global or local thresholding techniques will not work. (B) A low-radius median filter is applied to remove high-frequency noise. (C) A large-radius median filter is applied to isolate the low-frequency (background) noise. (D) The low-frequency filtered image in C is subtracted from the high-frequency filtered image in B. (E) A low-radius mean filter is applied, followed by a global high-pass segmentation to produce the final image.

It is important to note that the above approach did not result in the total removal of matrix from the medullary cavity, because the filtering parameters were chosen based on the results for regions of the specimen entirely occupied by cancellous bone. This did not pose a problem to analysis, however, because the remaining noise was essentially random and isotropic, and thus easily distinguished visually from the cancellous bone structure elsewhere. It was therefore easy to avoid the medullary canal in the architectural analyses.

Protocol 5—non-avian theropod medical CT scans (∼30 bones)

As with the other set of medical CT scans, the scans of the fossil non-avian theropod bones comprised anisotropic voxels. Moreover, the resolution of the scans, and spatial variation in density and preservation quality, prohibited the proper segmentation of the cancellous bone. Consequently, the scans were resampled in the axial direction only, to produce isotropic voxels; they were not resampled to a higher resolution, as per protocol 2. This also meant that the cancellous bone architecture was only able to be analysed qualitatively. One exception to this was the CT scans of the Masiakasaurus femora collected by Farke & Alicea (2009). Owing to the relatively high scan resolution and good preservation of the fossil bones, these scans were able to be processed as per protocol 2, and were subjected to quantitative analyses.

Architectural analyses

As the resolution and quality of the CT scans varied, several analytical techniques were used. In their own way, each helped to identify the dominant architectural direction in a given region of cancellous bone, and how this varied throughout an entire bony element. Quantitative fabric analysis (section ‘Quantitative analyses’) was possible for scans of most (85%) of the extant animal bones, as well as all the fossil specimens imaged using micro-CT scanning, that is, most scans that were processed as per protocols 1–4 above. The remaining scans of extant animals that were unable to be quantitatively analysed (pertaining to small bird bones), were analysed in a qualitative fashion (Section ‘Qualitative analyses’), as were the scans of extant birds obtained by Farke & Alicea (2009). The non-avian theropod scans processed as per protocol 5 above were only able to be analysed in a qualitative fashion; the scans of Masiakasaurus were able to be analysed quantitatively. A subset of the results from the quantitative analyses were also subject to additional analysis to obtain further insight into two key regions of the femur, the femoral head and medial condyle (section ‘Anatomically explicit analyses of femora’). Lastly, the architecture of diaphyseal cancellous bone in birds was analysed in a semi-quantitative fashion, via a categorical scoring method (section ‘Diaphyseal cancellous bone analyses’).

Quantitative analyses

Quantitative analyses of cancellous bone architecture were conducted using the software Quant3D 2.3 (Ketcham & Ryan, 2004; Ryan & Ketcham, 2002a, 2002b). The 3D fabric tensor (Cowin, 1986; see also Fig. 2) for a given volume of cancellous bone was calculated from the segmented CT scans using the star volume distribution method (Cruz-Orive et al., 1992; Odgaard, 1997, 2001). This is a stereological technique that expresses how cancellous bone material is distributed in three dimensions, and the results of this approach have previously been demonstrated to show strong correlation with cancellous bone mechanical properties (Kabel et al., 1999; Odgaard et al., 1997; Turner et al., 1990; Ulrich et al., 1999). To examine how the fabric varied spatially throughout a given bone, analyses were conducted on numerous (sometimes >700) discrete volumes of cancellous bone; each volume of interest (VOI) was spherical in shape, to avoid corner effects. The diameter of the VOIs was typically 5 or more times the mean trabecular spacing for a specimen, which is sufficiently large for the continuum assumption of cancellous bone behaviour to hold (Cowin, 2001; Harrigan et al., 1988), thus permitting the calculation of a fabric tensor.

Volumes of interest were arranged in the geometric pattern of cubic close-packed spheres (Fig. 11). This arrangement maximizes the volume of cancellous bone analysed, and as no overlap occurs between adjacent spheres, each part of cancellous bone was analysed only once. Hence, unlike the results of Saparin et al. (2011), the results for each VOI were independent of one another. The close-packed pattern was generated in the computer-aided design software Rhinoceros 4.0 (McNeel, Seattle, WA, USA) and scaled and fitted to each specimen as appropriate. By altering the VOI positions relative to the whole bone in some test specimens, it was found that the exact location of the VOIs did not influence the overall results. That is, the fabric vector field changed gradually enough throughout a bone such that the exact location of VOIs did not alter the results, and no part of the pattern of change was missed. Great care was taken to ensure that peripheral VOIs included at most only minimal amounts of cortical bone or the medullary cavity. This was achieved by manually removing VOIs that did not fall into cancellous bone regions, using 3D geometric models of each specimen in Rhinoceros (Fig. 11). Post-analysis inspection revealed that inadvertently including small amounts of cortical bone or medullary cavity did not alter the overall results. In the fossil specimens, cracks or other regions of obvious deformation (e.g. twisting, bending, flattening) were avoided (Bishop et al., 2017); they were identified and their geometry subsequently mapped out in the CT scans using Mimics, and the resulting geometry was used to remove VOIs from analysis in Rhinoceros (Fig. 11).

Figure 11 The arrangement of VOIs for quantitative analysis of cancellous bone architecture follows the geometric pattern of cubic close-packed spheres.

This is illustrated here for the proximal right femur of Troodontidae sp. (MOR 748). Each spherical VOI has a diameter of 5 mm. In this arrangement, each VOI is just touching its neighbouring VOIs, and each sphere can be in contact with up to 12 other VOIs. (A, B) Shown in medial view. (C, D) Shown in oblique anteromedial view. Notice in (C) and (D) that VOI placement avoids the medullary cavity (orange) and cracks (green).

For each VOI in a specimen, measurements were made for 2,049 uniformly distributed orientations at 4,000 points within the bone phase, with random rotation and dense vector sampling (Ketcham & Ryan, 2004). These were batch processed in Quant3D, controlled using a custom script in MATLAB 8.0 (MathWorks, Natick, MA, USA). For each VOI, Quant3D calculated the fabric tensor, of which the three principal directions (u1, primary; u2, secondary; u3, tertiary) corresponded to the principal fabric directions of cancellous bone in that particular VOI (i.e. the principal directions of alignment). Additionally, the eigenvalues of this tensor (e1, e2, e3) expressed the degree to which the cancellous bone was aligned to each of these principal directions. These parameters that described the fabric tensor were extracted from the Quant3D outputs for each VOI using a custom MATLAB script, and were plotted and visualized in 3D relative to the whole bone using another custom MATLAB script. The vector results were also further visualized and interrogated with respect to the whole bone in Rhinoceros.

Qualitative analyses

There were a few bird specimens that were too small (especially at their distal ends) for the quantitative architectural analyses described above to be implemented. Specifically, their bones were so small, and the trabeculae spaced far enough apart, that no VOI could be placed in the bone which would span at least five intertrabecular lengths. That is, the continuum assumption of cancellous bone behaviour (Cowin, 2001; Harrigan et al., 1988) would not hold, preventing a quantitative fabric analysis. For these specimens, cancellous bone architecture was qualitatively assessed by 3D visualization of the segmented CT scans using isosurface and volume renderings in Mimics and ImageJ, facilitating visual assessment of the predominant orientation of trabeculae in the different regions of a bone. This approach was also used for the scans of extant bird bones collected by Farke & Alicea (2009), which had relatively low resolution or high voxel anisotropy. Additionally, 3D visualization was performed on other specimens where quantitative architectural analysis was possible, to provide further insight into their architecture. Importantly, this also demonstrated that the physical morphologies observed by 3D visualization correlated with the fabric directions calculated via quantitative analyses: the dominant orientation of trabeculae as assessed visually largely coincided with the primary fabric direction.

For the non-avian theropod specimens imaged using medical CT scans, 3D visualization and assessment of cancellous bone architecture used the Volume Viewer 2.0 plugin for ImageJ (Barthel, 2006). This allowed real-time re-slicing and racking of a CT scan image stack in any orientation. Not only did this permit visualization of the scans from any direction, it also helped identify architectural features that were difficult to see in any single static slice, but which were revealed upon dynamic racking through successive slices. It also allowed for the identification of scanning artifacts (Ketcham & Carlson, 2001), and hence these could be ignored from assessments of architectural patterns. Based on these visualizations, the observed patterns were mapped onto whole-bone models via geometric representations of the dominant architectural patterns and directions, using a combination of spline curves and deformable surfaces in Rhinoceros. The accuracy of these geometric models was assessed through constructing multiple cutting planes through the model in different orientations, and comparing the cut model geometry to the CT slices in the same orientation. The geometric models were iteratively developed and refined as appropriate until good visual agreement between the scan data and the model was achieved. In using these geometric models, it is implicitly assumed that they accurately capture the dominant fabric directions in a given region of cancellous bone. All qualitative assessments were initially undertaken by a single person (PJB), with later conferral provided by the other authors. To help illustrate the above procedure for the reader, an example femur CT scan dataset and the resulting geometric model is freely accessible from a Figshare repository (https://doi.org/10.6084/m9.figshare.c.4224926). The scans and models can be visualized using the open source softwares ImageJ and Meshlab (http://meshlab.sourceforge.net/), respectively.

Anatomically explicit analyses of femora

The architectural directions identified by the quantitative and qualitative analyses above were largely interpreted in a qualitative fashion, in the context of whole-bone general osteology and spatial variation in cancellous bone fabric (i.e. how they varied throughout 3D space). However, strict quantitative focus was also directed towards the primary fabric direction of cancellous bone in the femoral head and medial femoral condyle, referenced to an explicit anatomical coordinates system defined by osteological landmarks. As extant birds employ a subhorizontal femoral posture, while most, if not all, non-avian theropods have been hypothesized to have employed a subvertical femoral posture (Carrano, 1998; Farlow et al., 2000; Gatesy, 1991b; Gatesy, Bäker & Hutchinson, 2009; Hutchinson, 2006; Hutchinson & Allen, 2009), it is logical that this may be reflected in the architecture of cancellous bone in these two regions. This correspondence is possible because flexion of the hip joint determines the degree of femoral crouch, and flexion of the knee joint will vary with femoral crouch such that the feet remain underneath the whole-body COM (Hutchinson & Allen, 2009). To assess this conjecture, the mean orientation of the primary fabric vector in the femoral head and medial femoral condyle was determined for each specimen (except for the crocodylians and varanids, which are sprawling quadrupeds) using a custom MATLAB script, by calculating the vectorial mean of the primary fabric vectors for all VOIs in the relevant region (Allmendinger, Cardozo & Fisher, 2013). The mean fabric direction was then referenced in a bone anatomical coordinate system, which was defined in a consistent manner across all species (Fig. 12). This anatomical coordinate system was based on two spheres fitted to the distal condyles, calculated using 3-Matic 9.0 (Materialize NV, Leuven, Belgium), and the principal axis of inertia of the whole bone, calculated for a surface mesh of the bone using Meshlab 1.3.3. The z-axis was defined by the principal axis of inertia (i.e. the long-axis of the bone; +z is proximal); the y-axis defined as the cross-product of the z-axis and the vector joining the centres of the two condyle spheres (+y is anterior); and the x-axis defined as the cross-product of the y- and z-axes (+x is lateral). The anatomically referenced mean fabric directions were then assessed using a stereographic projection (stereoplot) of the data (Amson et al., 2017; Ryan & Ketcham, 2005).

Figure 12 Definition of the anatomical coordinate system for the femur used in this study.

This is illustrated here with the right femur of a Australian bustard, Ardeotis australis (MVB 20408), in anterior (A) and oblique anteromedial (B) views. The anatomical coordinate system is defined by the principal axis of inertia of the bone (dashed line) and the centres of two spheres fit to the distal condyles (white circles). For left femora, their geometry and calculated mean orientation of the primary fabric vector were mirrored prior to construction of an anatomical coordinate system.

Diaphyseal cancellous bone analyses

In both the bird and other reptile specimens investigated, a significant amount of cancellous bone was frequently observed to encroach from the ends of the bone in to the diaphysis (shaft), in both the femur and tibia. In the regions nearer the ends, the amount was sometimes sufficient enough that quantitative fabric analyses could be undertaken, particularly in the bones of larger species. Usually, however, too little cancellous bone was present to permit such quantitative analysis. Nevertheless, throughout the course of 3D visualization, these bones were observed to show a variety of interesting patterns, and so an attempt was made to characterize the variation in architectures in the bird bones, by categorical scoring.

Categorical scoring was performed by five independent, volunteer observers (non-scientists who had varying levels of experience with vertebrate anatomy), who were blind to the objectives of the present study. They were asked to inspect the same 3D isosurface models of each bone, derived from the segmented CT scans, assess them for three features and score each bone on a pre-defined, categorical scale for each feature. The first feature was the bulk spatial extent of cancellous bone in the diaphysis, with scores assigned on a four-point scale: 0 = essentially the whole diaphysis was void of cancellous bone, 1 = less than half of the diaphysis was occupied by cancellous bone, 2 = half or more than half of the diaphysis was occupied by cancellous bone, 3 = essentially the whole diaphysis was occupied by cancellous bone. The second feature was the average orientation of trabeculae with respect to the long-axis of the shaft, with scores assigned on a nine-point scale of 10° increments (Fig. 13A): 0 = 0–10°, 1 = 10–20°, 2 = 20–30° and so on. The third feature was the tendency of trabeculae to be closely associated with other trabeculae, with scores assigned on a three-point scale (Fig. 13B): 0 = trabeculae mostly or always occur singly, well separated from any other trabeculae; 1 = trabeculae occur in small groups of two or three other, similarly situated trabeculae; 2 = trabeculae mostly or always occur in close association with many other similarly situated trabeculae. If a given bone was scored as ‘0’ for the first feature, then it was not scored for the second or third (‘n/a’).

Figure 13 Schematic illustration of two of the features of diaphyseal cancellous bone that were subject to categorical scoring.

(A) The orientation of trabeculae with respect to the long-axis of the shaft. (B) The tendency of trabeculae to be closely associated with other trabeculae. Upper row is oblique view, lower row is section-on view. These illustrations were given to the scorers to provide them with a point of reference.

Following scoring, the mean score across the five observers was taken for each bone and for each morphological feature. For the second and third features, a given bone was sometimes scored as ‘n/a’ (if the first feature was scored as a ‘0’), and other times it was given an actual numeric score. If there were more ‘n/a’ scores than numeric scores assigned for a given specimen, then the mean score was taken as ‘n/a’, and that particular specimen did not contribute towards further analyses (detailed below). However, if the majority of scores assigned were numeric, then the mean score was taken as the mean of those numeric scores.

Statistics

Three datasets were examined to test for a relationship between measured parameters and body size in birds: the anterior inclination of the mean fabric direction in the femoral head, the posterior inclination of the mean fabric direction in the medial femoral condyle, and the results of categorical scoring of diaphyseal cancellous bone. The length of the relevant bone was taken here as a proxy for body size, and was measured as the total proximal distance between articular condyles (interarticular length), excluding crests and trochanters. Bone length was used here as it has direct mechanical relevance to the loads a bone experiences (e.g. it determines lever arms for bending moments); additionally, using bone length precludes any margin for error that is attendant with the use of other body size measures, such as estimated body mass. A relationship with body size would normally be assessed using a standard parametric test of the slope of a major axis regression (Warton et al., 2006). However, the data often exhibited non-normal distribution of errors (non-normal probablility plot) and heteroscedasticity (Breusch–Pagan test), as determined in PAST 3.09 (Hammer, Harper & Ryan, 2001). As such, a parametric test could not be implemented. Instead, a permutation test of the slope with 100,000 replicates was used, implemented in a custom MATLAB script (Legendre & Legendre, 2012); the significance level was set at p = 0.05.

In addition, the reliability of the scorers in the analysis of diaphyseal cancellous bone was assessed using the intraclass correlation coefficient (ICC). This was calculated in PAST 3.09, using the ICC(2, k) model of (Shrout & Fleiss, 1979), where k in this instance is 5. The higher the value of the ICC, the more reliable the scorers. The ICC was only able to be calculated for bones that had received numeric scores by all five scorers. Furthermore, for features 2 and 3 of the tibiotarsus, an ICC could not be calculated, because there was only one specimen in each set that had received a numeric score by all five scorers. That is, for every other specimen, at least one scorer had scored feature 1 as ‘0’, and then assigned the ‘n/a’ score to the remaining two features.

Results

Since the architecture of cancellous bone in all but one of the species investigated here (humans) has never been studied before, the results of this study are first and foremost descriptive, qualitatively characterizing the whole-bone architectural patterns. Where quantitative data was produced, it is presented where appropriate, including graphs, stereographic plots and vector field plots as well as reporting of numerical results (including statistics when possible); elsewhere, qualitative observations are presented using geometric models and isosurface renderings of bones in cross-section, in concert with verbal anatomical description. The presentation of data and observations is organized primarily by each bone investigated: the femur in section ‘Femur’, the tibia (tibiotarsus) in section ‘Tibia or tibiotarsus’ and the fibula in section ‘Fibula’. For each bone, results are presented in sequential order, first with humans and birds (extant bipeds), then other extant reptiles (theropod outgroups) and finally the non-avian theropods. Overall, birds were quite consistent in their observed architectural patterns, across all species studied, and it is therefore convenient to treat all species together, with specific differences noted where appropriate. The observations for non-avian theropods are presented in approximate phylogenetic order (i.e. clades progressively closer to crown-group birds). Furthemore, the observations reported for a given group pertain to all specimens studied for that group, unless otherwise indicated. It should also be noted that since this study was undertaken, a recent re-evaluation of North American troodontid taxonomy has cast doubt on the validity of the name Troodon formosus (Van der Reest & Currie, 2017). As such, the troodontid material investigated here will be referred to as ‘Troodontidae sp.,’ pending further study.

Femur

General remarks

Observations for the single human femur studied are consistent with previously published reports (Elke et al., 1995; Garden, 1961; Koch, 1917; Singh, Nagrath & Maini, 1970; Takechi, 1977; Tobin, 1955; Townsley, 1944; Von Meyer, 1867), supporting the use of this specimen as a general reference for humans. Cancellous bone occurs throughout both proximal and distal ends, but the majority of the diaphysis is devoid of it. However, cancellous bone is present deep to and in the immediate vicinity of the lesser trochanter on the proximal shaft, which is distinct from and non-homologous with the lesser trochanter of theropods.

Cancellous bone is more extensive in the birds and other reptiles than humans. It usually encroaches into the diaphysis to a varied, and often large, extent, although moving towards mid-shaft it tends to become progressively more sparse and restricted to the endosteal margin of the cortex. A more detailed treatment of diaphyseal cancellous bone is given below (section ‘Diaphyses’). Pneumatized bird femora—easily distinguished by the presence of pneumatopores—are distinctly more ‘loose-packed’ compared to marrow-filled bird femora; that is, pneumatized bones have greater trabecular spacing, a distinction that exists in both small and large species (Fig. 14). Despite the modulating effect that pneumatization has on trabecular spacing, no effect on the patterns of fabric direction was evident.

Figure 14 Pneumatization modulates trabecular spacing in the femur of both large and small birds.

This is illustrated here with processed CT scan slices located approximately midway through the femoral head in the axial plane. (A) A marrow-filled femur of a southern cassowary, Casuarius casuarius (QMO 30105), with mean trabecular spacing = 0.638 mm. (B) A pneumatized femur of an emu, Dromaius novaehollandiae (QMO 16140) with mean trabecular spacing = 1.128 mm. (C) A marrow-filled femur of a chicken, Gallus gallus (PJB coll.), with mean trabecular spacing = 0.320 mm. (D) A pneumatized femur of an Australian brush turkey, Alectura lathami (PJB coll.), with mean trabecular spacing = 0.999 mm. Reported trabecular spacing values were calculated (for illustrative purposes) for the femoral head using the BoneJ 1.3.11 plugin for ImageJ (Doube et al., 2010). (A) and (B) are shown at the same scale, as are (C) and (D). Scale bars are 10 mm, and yellow asterisks denote pneumatopores.

Similar to humans, in non-avian theropod femora the proximal and distal ends are fully occupied by cancellous bone, but the majority of the diaphysis remains empty. As with the human lesser trochanter, cancellous bone also occurs locally deep to the fourth trochanter, although this was only observed in the larger species, where the fourth trochanter is prominently developed (both on account of bone size and phylogenetic position: Gatesy, 1990; Hutchinson, 2001a). In larger femora, cancellous bone can encroach further into the diaphysis, with a limited number of thick trabeculae extending from the metaphysis along the endosteal surface. This encroachment is most extensive in Tyrannosaurus femora, although the middle third to half of the bone still remains free of trabeculae. It is likely that this size-dependency of diaphyseal encroachment by cancellous bone is purely a function of allometry: in longer femora, the epiphyses are larger and the diaphysis is relatively shorter in length.

Proximal femur

In the human proximal femur, the primary fabric direction (u1) in the head is essentially proximodistally oriented, with a gentle anteromedial inclination of about 5° from the proximodistal axis, which grades into a stronger medial inclination in the distal (inferior) part of the femoral neck (Fig. 15A). This corresponds to the widely recognized ‘primary compressive group’ noted in previous studies of humans (see also Fig. 4A), as well as other primates (Rafferty, 1998; Scherf, 2008). In the region of the greater trochanter (which is distinct from and non-homologous with the greater trochanter of theropods), u1 largely parallels the lateral margin of the trochanter, corresponding to the ‘greater trochanter group’ of previous studies (Fig. 15A). Within the main part of the metaphysis, a double-arcuate pattern of u1 occurs in the coronal plane (or more accurately, the plane containing the shaft and femoral neck), where it arcs from opposite sides of the metaphysis to intersect in the middle. Moreover, the secondary fabric direction (u2) in this region is also largely contained within (parallel to) the coronal plane. This double-arcuate pattern corresponds to the ‘secondary compressive’ and ‘secondary tensile’ groups of previous studies (Fig. 15B). The orientation of u1 in the distal metaphysis, underneath the lesser trochanter and in the transition to the diaphysis is subparallel to the long-axis of the bone (Fig. 15C).

Figure 15 The main architectural features of cancellous bone in the human proximal femur.

(A) Vector field of u1 in the head, inferior neck and greater trochanter regions, plotted on a translucent rendering of the external bone geometry; view is in the coronal plane. Schematic inset illustrates close correspondence with the principal compressive (PC) and greater trochanter (GT) groups of previous studies. (B) Vector field of u1 (red) and u2 (blue) in the middle of the metaphysis, viewed in the coronal plane. Schematic inset illustrates close correspondence with the secondary compressive (SC) and secondary tensile (ST) groups of previous studies. Note that both u1 and u2 are largely parallel to the coronal plane. (C) Vector field of u1 in the distal metaphysis and lesser trochanter (in oblique proximomedial view), which is largely parallel to the bone’s long-axis. In this and all subsequent illustrations of fabric vector fields, the length of each fabric vector is proportional to its corresponding fabric eigenvalue. Additionally, all images are of bones from the right side of the body.

The proximal femur of birds shows the same general pattern irrespective of size. In the femoral head, u1 is mainly proximodistally oriented, but there tends to be a variable degree of anteromedial inclination superimposed upon this (Figs. 16A–16D). The anterior component of this inclination is often quite pronounced, ranging up to 60° from the proximodistal axis. This anteromedial inclination continues down into the distal femoral neck as well. Under the facies antitrochanterica, u1 is largely proximodistally oriented, but often there is also a gentle posteromedial inclination (Figs. 16E–16I). This orientation continues towards the trochanteric crest, where the medial and posterior inclination often becomes more pronounced (Figs. 16J–16N). However, in the anteriormost part of the trochanter, there can sometimes be no posterior inclination at all. Progressing distally down the metaphysis, beyond the distal level of the femoral head and into the diaphysis, the orientations of u1 become more ‘disorganised’: they take on a more oblique orientation relative to the long-axis of the bone, and the change in direction across the bone is no longer a gradual transition (Figs. 16O and 16P). No double-arcuate pattern of u1 and u2, as described above for the human femur, was observed in any bird specimen. Moreover, u2 is generally subparallel to the axial plane, and in large bones they are also parallel to the bone periphery, forming a concentric pattern. Progressing distally through the metaphysis, however, the orientations of u2 become more obliquely oriented and disorganized, as with the orientations of u1. Although fabric orientation could not be extensively quantified in the smaller bird femora, visualization of isosurface and volume renderings of the bones themselves shows that for a given region, the trabeculae are oriented in the same general direction as u1 for larger bird femora (Figs. 16I and 16N).

Figure 16 The main architectural features of cancellous bone in the proximal femur of birds.

(A–D) Vector field of u1 in the femoral head and inferior neck of an emu, Dromaius novaehollandiae (QMO 16140; A, B), and ostrich, Struthio camelus (MV R.2385; C, D). (E–H) Vector field of u1 under the facies antitrochanterica of a greater rhea, Rhea americana (QMO 23517; E, F), and chicken, Gallus gallus (PJB coll.; G, H). (I) Isosurface rendering of cancellous bone under the facies antitrochanterica of a dusky moorhen, Gallinula tenebrosa (PJB coll., between arrows), sectioned in the sagittal plane. (J–M) Vector field of u1 in the trochanteric crest of a southern cassowary, Casuarius casuarius (QMO 30105; J, K), and Struthio camelus (MV R.2385, L, M). (N) Isosurface rendering of cancellous bone in the trochanteric crest of a magpie goose, Anseranus semipalmata (QMO 29529, between arrows), sectioned in the sagittal plane. (O, P) Vector field of u1 throughout the entire proximal femur of Dromaius novaehollandiae (QMO 16140, O) and Casuarius casuarius (QMO 30105, P), which illustrates the increasing obliquity and disorganization of vectors in the distal metaphysis and transition to the diaphysis, shown in regions with braces. (A, C, J and L) are anterior views; (B and D) are medial views; (E and G) are posterior views; (F, H, K and M) are lateral views; (O) is an oblique anterolateral view; (P) is an oblique anteromedial view. For reference, silhouettes of the animals depicted are provided in this figure and those that follow.

In the extant sprawling reptile femora examined, the orientation of u1 is subparallel to the long-axis of the bone throughout the metaphysis (Figs. 17A–17D), except in the region of the fourth trochanter, where it is largely parallel to the long-axis of the trochanter itself (Figs. 17B and 17D). Leading up to the head region, u1 fans out, away from the metaphysis and towards the articular surface (Figs. 17A–17D). In the distal metaphysis and transitioning to the diaphysis, the orientation of u1 progressively becomes more obliquely oriented and disorganized, in the same fashion as the bird femora (Fig. 17E).

Figure 17 The main architectural features of cancellous bone in the proximal femur of extant sprawling reptiles.

(A, B) Vector field of u1 in the proximal femur of a freshwater crocodile, Crocodylus johnstoni (QMJ 47916). (C, D) Vector field of u1 in the proximal femur of a Spencer’s goanna, Varanus spenceri (QMJ 84416). (E) Vector field of u1 throughout the proximal femur of a Komodo dragon, Varanus komodoensis (AM R.106933), which illustrates the increasing obliquity and disorganization of vectors in the distal metaphysis and transition to the diaphysis, shown in region with braces. (A and C) are anterior views (‘dorsal view’ of herpetologists); (B and D) are lateral views (‘posterior view’ of herpetologists); (E) is an oblique anterolateral view. For clarity, the vectors of u1 in the fourth trochanter are not visible in (A, C and E).

Only limited information could be gleaned for the proximal femur of the small non-avian theropod Masiakasaurus, owing to the small size of the specimens. The orientation of u1 throughout the proximal end is more or less proximodistally directed, leading from the base of the femoral neck up to the apex of the head (Fig. 18). A gentle medial inclination from the proximodistal axis is present in most specimens.

Figure 18 The orientation of u1 throughout the femoral head of Masiakasaurus knopfleri, here exemplified by FMNH PR 2117.

(A) Anterior view. (B) Lateral view.

The proximal femur of both Allosaurus and the tyrannosaurids show a strikingly similar pattern to that of humans (Figs. 19A–19E). There is a well-developed tract of dense, cancellous bone extending from the base of the femoral neck up towards the apex of the head of the femur, much like the ‘primary compressive group’ of humans (Figs. 19A–19E, maroon). As in humans, too, this tract has a gentle anterior inclination relative to the proximodistal axis (Fig. 19F). Additionally, a double-arcuate pattern is evident in the tyrannosaurids, also similar to the human pattern. The 3D visualization of CT scans suggests that whilst this pattern is most developed in the coronal plane, it does extend out of that plane somewhat, with the ‘sheets’ of trabeculae being partially concentric with the bone’s periphery (Figs. 19A–19E, maroon and green). A second double-arcuate pattern is also present in the lesser trochanter, again subparallel to the coronal plane (Figs. 19A–19E, turqoise and purple). A modest quantity of cancellous bone is present in the fourth trochanter, and the dominant direction is directed posteroproximally, parallel to the distal margin of the trochanter (Figs. 19A–19E, blue). In the distal metaphysis all observed cancellous bone (and sometimes individual trabeculae) is oriented subparallel to the bone’s long-axis; there is no indication of marked obliquity or disorganization as seen in the bird and other reptile femora.

Figure 19 The main architectural features of cancellous bone in the proximal femur of both Allosaurus and the tyrannosaurids.

These are illustrated here with a 3D geometric model of the observed architecture, mapped to the femur of Daspletosaurus torosus (TMP 2001.036.0001). (A–E) Five progressive rotations of the bone, in 30° increments, from anteromedial to anterolateral views (C is a purely anterior view). (F) The observed orientation of the dominant tract of cancellous bone in the femoral head (blue) has a gentle anterior inclination; bone shown in medial view. For explanation of the features and colour coding, refer to the main text. Inset below C is a CT slice through the proximal femur of Tyrannosaurus rex (MOR 1128), parallel to the coronal plane and through the middle of the femoral head. This illustrates the very characteristic tract of cancellous bone that extends from the base of the femoral neck up towards the apex of the head, highly comparable to the tract present in humans (cf. Fig. 4A).

In the femoral head of ornithomimids, the most conspicuous feature is that u1 is oriented predominantly in an anteroposterior direction, yet u2 is oriented proximodistally with a gentle medial inclination (directed towards the apex of the head), much like the orientation of u1 in the femoral head of humans, birds and (presumably) Allosaurus and the tyrannosaurids (Figs. 20A–20D). In fact, a generally proximodistal orientation of u2 and a generally anteroposterior orientation of u1 also occurs throughout much of the proximal end of the femur. Only around the anterior and posterior peripheries of the bone does u1 assume the more typical proximodistal orientation, with a medial inclination in the region of the femoral head and neck. It also assumes a more proximodistal orientation progressing towards the greater trochanter and distal end of the metaphysis, but in the latter region is still shows a marked level of ‘disorganization’ (Fig. 20E). Along the anterior part of the lesser trochanter, u1 is aligned more or less parallel with the anterior margin in a generally proximodistal fashion (Fig. 20F). Further posteriorly into the main body of the trochanter, however, it takes on a more mediolateral orientation, which gradually becomes more anteroposteriorly directed as the trochanter merges with the main body of the proximal femur.

Figure 20 The main architectural features of cancellous bone in the proximal femur of ornithomimids and caenagnathids.

(A–D) Vector field of u1 (A, C) and u2 (B, D) in the femoral head and proximal metaphysis of an inteterminate ornithomimid (TMP 85.36.276), in oblique anteromedial (A, B) and oblique anterolateral (C, D) views. Note that the vector field along the anterior and posterior peripheries of the femoral head are not shown here (for clarity), where they are more typically oriented as in birds and humans. (E) Vector field of u1 in the greater trochanter region and distal metaphysis of an indeterminate ornithomimid (TMP 85.36.276), in oblique anterolateral view; note the increased obliquity and disorganization of vectors in the distal metaphysis (region with braces). (F) Vector field of u1 in the lesser trochanter of an indeterminate ornithomimid (TMP 91.36.569), in oblique anteromedial view. (G) Vector field of u1 in the proximal femur of an indeterminate caenagnathid (TMP 86.36.323), in a 3D slice parallel to the axial plane and through the femoral head and lesser trochanter. Main image is shown in axial view (anterior is toward bottom of page), with inset illustrating the region illustrated in context of the whole bone. The medialmost part of the femoral head is missing (dotted line).

The single caenagnathid femur examined exhibits a pattern very much like that in the ornithomimids. The only notable difference occurs in relation to the lesser trochanter, which has decreased in size and become more unified with the greater trochanter. Here, the fabric pattern is essentially a continuation of what occurs elsewhere in the head and metaphysis, with u1 becoming anteromedially oriented, but still subparallel to the axial plane (Fig. 20G). The orientation of u2 remains predominantly proximodistal, although a small posterior component is also present.

In the therizinosauroid Falcarius, the spatial pattern of the orientation of u1 is largely comparable to that observed in humans, throughout the whole proximal end of the femur, save the lesser trochanter. In the head, u1 has a slight anteromedial component superimposed over an otherwise predominantly proximodistal orientation, which is directed toward the apex (Fig. 21A). In the middle of the metaphysis, u1 exhibits a weakly developed double-arcuate pattern that is parallel to the coronal plane (Fig. 21A). Additionally, in the distal metaphysis u1 is largely subparallel to the long-axis of the bone, showing little ‘disorganization’ as seen in the birds. In the lesser trochanter, both u1 and u2 are largely contained within the coronal plane. Towards the base of the trochanter, u1 is generally oriented proximodistally, and u2 is generally oriented mediolaterally; however, nearer the apex, u1 is generally oriented mediolaterally, and u2 is generally oriented proximodistally (Figs. 21B and 21C). One distinct difference to the pattern observed in humans is the orientation of u2 in much of the proximal end; here, it is anteroposteriorly aligned, much like u1 in the ornithomimid femora (Fig. 21D).

Figure 21 The main architectural features of cancellous bone in the proximal femur of Falcarius utahensis and Troodontidae sp.

(A) Vector field of u1 in the proximal femur of Falcarius (UMNH VP 12361), viewed as a 3D slice parallel to the coronal plane and through the middle of the bone. Schematic inset illustrates three main trajectories in this image, which are not too dissimilar from the patterns observed in humans and large non-avian theropods (cf. Figs 15, 19). (B, C) Vector field of u1 (B) and u2 (C) in the lesser trochanter of Falcarius, in oblique anterolateral view. (D) Vector field of u2 in the proximal femur of Falcarius, in a 3D slice parallel to the axial plane and through the femoral head. Main image is shown in axial view (anterior is toward bottom of page), with inset illustrating the region illustrated in context of the whole bone. (E, F) Vector field of u1 in the femoral head and inferior neck of Troodontidae sp. (MOR 748) in anterior (E) and medial (F) views. (G, H) Vector field of u1 in the region of the greater trochanter of Troodontidae sp. (MOR 553s-7.28.91.239) in posterior (G) and lateral (H) views. (I) Orientation of u1 in the lesser trochanter, or immediate region thereof, of Troodontidae sp., in oblique anterolateral view (main image illustrates MOR 748; inset illustrates MOR 553s-7.28.91.239). (J) Vector field of u1 throughout the metaphysis of Troodontidae sp. (MOR 553s-7.28.91.239), illustrating increasing obliquity and disorganization of vectors in the distal metaphysis and transition to the diaphysis (region with braces).

In the proximal femur of Troodontidae sp., the orientation of u1 is predominantly proximodistal. Relative to the proximodistal axis, it assumes a gentle medial inclination as it courses from the base of the femoral neck up towards the apex of the head; within the head, it also takes on an anterior inclination (Figs. 21E and 21F). Under the region homologous with the facies antitrochanterica of birds, and in the region of the greater trochanter, u1 has a posteromedial component to its orientation (Figs. 21G and 21H). In the anterior part of the proximal femur, u1 is largely contained within the coronal plane, but shows little preferred orientation within that plane; u2 is not contained within the coronal plane, unlike in the homologous region of the femur of Falcarius (base of lesser trochanter). In the region of the lesser trochanter, which is small and proximally located in Troodontidae sp., u1 tends to take on a more mediolateral orientation (Fig. 21I). Progressing more distally through the metaphysis, the orientation of u1 becomes more disorganized and oblique to the bone’s long-axis, as seen in the proximal femur of birds and sprawling reptiles (Fig. 21J). No double-arcuate pattern of any form, as observed in humans, Allosaurus, tyrannosaurids or Falcarius, was observed in Troodontidae sp.

The mean orientation of the primary fabric direction u1 in the femoral head for each specimen analysed is presented in Fig. 22. The ornithomimid and caenagnathid specimens were excluded, owing to their distinct fabric architecture (mean fabric directions of u1 are oriented at almost 90° to the general orientation of what is observed in other species; see also above); additionally, the femoral head of the caenagnathid specimen was incomplete. A general orientation was taken to represent Allosaurus and the tyrannosaurids, based on the geometric model developed to represent their architectural patterns (Fig. 22A, purple; cf. Figs. 22C and 22D). In almost all specimens, the mean fabric direction is oriented anteromedially. The human and Masiakasaurus specimens plot close to the pole of the stereoplot, indicating that their mean directions are oriented almost purely proximodistally. The plot for Falcarius and the general Allosaurus–tyrannosaurid pattern is a little further from the pole (more medially directed), but still has little anterior inclination. The bird results demonstrate a sizeable degree of spread in the northwest quadrant of the plot, and indeed variability occurs within the species or genera for which multiple individuals were studied (Fig. 22B). However, on the whole, birds exhibit a substantial anterior inclination to the mean fabric direction; the mean direction across all birds is inclined 21.6° anterior of the proximodistal axis in the sagittal plane. Major axis regression of the bird data set revealed that the sagittal inclination of the mean fabric direction did not vary significantly with femur length (slope = 0.059, intercept = 58.4, r2 = 0.100, p = 0.108). The mean primary fabric direction in Troodontidae sp. is of an intermediate orientation between that of birds and the other non-avian theropods, having an anterior inclination in the sagittal plane of 15.8°.

Figure 22 The mean orientation of u1 in the femoral head, referenced in the femur anatomical coordinate system.

This is plotted on an equal-angle stereoplot, with northern hemisphere projection (using StereoNet 9.5; Allmendinger, Cardozo & Fisher, 2013; Cardozo & Allmendinger, 2013). (A) Results for all specimens analysed. Note that for the fossil specimens, only those that were complete and well-preserved, and enabled an anatomical coordinate system to be defined, were analysed. Colour codes: black = birds, pink = human, blue = Masiakasaurus (FMNH PR 2153, UA 8384), orange = Falcarius, green = Troodontidae sp. (MOR 748), purple = general orientation for Allosaurus and the tyrannosaurids, red = mean orientation across birds. (B) Results for species or genera that were multiply sampled often displayed significant instraspecific or intrageneric variation, ranging up to 30.6°. Colour codes: black = ostrich, Struthio camelus, blue = southern cassowary, Casuarius casuarius, orange = emu, Dromaius novaehollandiae, red = chicken, Gallus gallus, green = kiwi, Apteryx spp. (C, D) The orientation of u1 superimposed on the CT scan of a representative tyrannosaur femur (Daspletosaurus torosus, TMP 2001.036.0001) in coronal (C) and sagittal (D) views, illustrating how the orientation data extracted and illustrated in the geometric models (e.g. Fig. 19) qualitatively reflects the observed cancellous bone architecture in the scans.

Distal femur

Throughout the distal femur of the human specimen, u1 is predominantly oriented subparallel to the long-axis of the bone. In the central part of the metaphysis, u1 exhibits a double-arcuate pattern in both the sagittal and coronal planes, with the individual fabric vectors tending to arc from the periphery of the bone in towards the centre (Figs. 23A and 23B). However, this double-arcuate pattern is not as strongly developed as in the proximal femur. In both the medial and lateral condyles, u1 has a largely proximodistal orientation (Fig. 23C). Additionally, u2 is largely parallel to the axial plane, and two prominent tracts or trajectories are evident, one in each condyle (Fig. 23D). These tracts, noted previously (Takechi, 1977) arc from the anterior aspect of their respective condyle back towards the posterior aspect, and together they form a distinctive ‘butterfly pattern’.

Figure 23 The main architectural features of cancellous bone in the human distal femur.

(A) Vector field of u1 in a 3D slice parallel to the coronal plane, made through the middle of the condyles. Schematic inset illustrates weakly developed double-arcuate pattern. (B) Vector field of u1 in a 3D slice parallel to the sagittal plane, made between the condyles. Schematic inset illustrates weakly developed double-arcuate pattern. (C) Vector field of u1 in the medial condyle, shown for a 3D slice parallel to the sagittal plane, made through the middle of the condyle. (D) Vector field of u2 at the level of the condyles, shown for a 3D slice parallel to the axial plane, made through the middle of the condyles; anterior is toward top of page. Schematic inset illustrates the two distinctive tracts that comprise a ‘butterfly pattern.’

In birds, the distal femur exhibits a fairly consistent set of patterns across species. The orientation of u1 in the metaphysis is largely proximodistally oriented. In the central metaphysis of larger birds, for which substantial quantification of fabric direction was possible, u1 exhibits a moderately developed double-arcuate pattern in the sagittal plane between the medial and lateral condyles, much as in humans (Fig. 24A). Also as observed in humans, the orientation of u2 in the condyles forms a butterfly pattern in the axial plane, or more correctly, the plane that passes through the centres of the condyles (Figs. 24B–24D). Unlike humans, however, the orientation of u1 in the condyles often has a marked posterior inclination relative to the proximodistal axis, although it generally remains subparallel to the sagittal plane (Figs. 24E–24K). Moreover, u1 sweeps a distinctly wide arc in the sagittal plane (often in excess of 100°), such that in the posterior and posterodistal extremities of the condyles, u1 can be perpendicular to the proximodistal axis. In large birds, this sweeping can also extend into the anterior parts of the condyles, where u1 is anterodistally directed (Fig. 24F). In the condyles of small bird femora for which only limited quantitative analysis was possible, both u1 and u2 are subparallel to the sagittal plane, and u1 is inclined posteriorly. In small bird femora for which quantitative analysis was not possible, much of the distal end is typically occupied by a small number of large but sparsely dispersed trabeculae. They vary from rod- to plate-shaped, but generally are parallel to the sagittal plane (Figs. 24D and 24K). As with the proximal femur, progressing towards the proximal metaphysis and into the diaphysis reveals a more oblique and disorganized nature to the individual vectors (Fig. 24L).

Figure 24 The main architectural features of cancellous bone in the distal femur of birds.

(A) Vector field of u1 in the central metaphysis of a southern cassowary, Casuarius casuarius (QMO 30105), in a 3D slice, parallel to the sagittal plane and between the condyles, shown in lateral view. Note the weakly developed double arcuate pattern. (B, C) Vector field of u2 in a 3D slice through the middle of the condyles in an ostrich, Struthio camelus (MV R.2385, B), and a emu, Dromaius novaehollandiae (QMO 11685, C), shown in distal view. Note the ‘butterfly pattern’ in both examples. (D) Isosurface rendering of cancellous bone in the distal condyles of a Japanese quail, Coturnix japonica (PJB coll.), sectioned in the axial plane; notice the ‘butterfly pattern’ between the arrows. (E–H) Vector field of u1 in the medial condyle of Dromaius novaehollandiae (QMO 16140, E, F) and Casuarius casuarius (QMO 30604, G, H), shown in anterior (E, G) and medial (F, H) views. (I, J) Vector field of u1 in the lateral condyle of a chicken, Gallus gallus (PJB coll.), shown in anterior (I) and lateral (J) views. (K) Isosurface rendering of cancellous bone in the medial condyle of a purple swamphen, Porphyrio porphyrio (PJB coll.), sectioned in the sagittal plane. (L) Vector field of u1 throughout the entire distal femur of Casuarius casuarius (QMO 31137), illustrating increasing obliquity and disorganization of vectors in the proximal metaphysis and transition to the diaphysis (region with braces).

In the distal femur of extant sprawling reptiles, the orientation of u1 in the metaphysis is largely subparallel to the long-axis of the bone. In some specimens u1 becomes more disorganized and obliquely oriented relative to the bone’s long-axis progressing towards the diaphysis, but this is not as pronounced as compared to the proximal femur, or as compared to birds. The orientation of u2 in the condyles is largely parallel to the axial plane, and exhibits the butterfly pattern seen in birds and humans (Figs. 25A and 25B). The main point of difference from birds arises in the orientation of u1 in the condyles: whilst u1 is posteriorly inclined in the sagittal plane, it does not display the large anteroposterior sweeping that is often present in birds, sweeping at most about 40° (Figs. 25C–25F).

Figure 25 The main architectural features of cancellous bone in the distal femur of extant sprawling reptiles.

(A, B) Vector field of u2 in a 3D slice through the middle of the condyles in a freshwater crocodile, Crocodylus porosus (QMJ 48127, A), and a Komodo dragon, Varanus komodoensis (AM R.106933, B), shown in proximal view. (C, D) Vector field of u1 in the medial condyle of Crocodylus porosus (QMJ 48127), shown in anterior (C) and medial (D) views. (E, F) Vector field of u1 in the lateral condyle of a Spencer’s goanna, Varanus spenceri (QMJ 484416), shown in anterior (E) and lateral (F) views.

As with the proximal femur, only limited information could be gleaned for the distal femur of Masiakasaurus. The orientation of u1 throughout the distal end is more or less proximodistally oriented, generally with a slight posterior inclination in the sagittal plane.

In the distal femur of the tyrannosaurids, cancellous bone around the periphery of the metaphysis is oriented subparallel to the long-axis of the bone. Additionally, there are two sets of paired, arcuate, sheet-like tracts of cancellous bone, which arc largely in the coronal plane (Fig. 26). The obliquity of these sheet-like tracts appears to change across the bone, such that one set of tracts radiates from the ‘patellar’ or intercondylar groove at the anterior margin of the bone (Fig. 26, maroon and turquoise), and the other set radiates from the popliteal area at the posterior margin of the bone (Fig. 26, green and purple). This feature was not observed in the Allosaurus specimens studied, for insufficient CT scan contrast or resolution did not reveal any information about the metaphysis. In the medial and lateral condyles of both Allosaurus and the tyrannosaurids, the dominant direction of cancellous bone is largely parallel to the sagittal plane, and parallel to the long axes of the condyles, thus producing a butterfly pattern in axial cross-section. Within the sagittal plane, the dominant direction has a marked posterior inclination relative to the proximodistal axis; it also exhibits some amount of anteroposterior sweeping, about 30° or so (Fig. 26, red).

Figure 26 The main architectural features of cancellous bone in the distal femur of both Allosaurus and the tyrannosaurids.

These are illustrated here with a 3D geometric model of the observed architecture, mapped to the femur of Daspletosaurus torosus (TMP 2001.036.0001). (A–G) Seven progressive rotations of the bone, in 30° increments, from medial to lateral views (D is a purely anterior view). Note that the architecture of the metaphysis was not observed in the Allosaurus specimens studied, owing to insufficient contrast or resolution in the CT scans failing to reveal any information about the metaphysis. For explanation of the features and colour coding, refer to the main text.

In both the ornithomimid and caenagnathid specimens, the orientation of u1 is predominantly oriented subparallel to the long-axis of the bone throughout much of the distal femur, particularly anteriorly. In much of the medial and lateral condyles, u1 is largely parallel to the sagittal plane and gently inclined posteriorly (Figs. 27A–27D); anteroposterior sweeping in the sagittal plane is limited to about 20°. However, in the posterior parts of both condyles, the posterodistal orientation gradually changes to become nearly perpendicular to the long-axis of the bone, and almost perpendicular to the local bone surface around the intercondylar sulcus (Figs. 27A–27D, yellow). That is, in the posterior parts of the condyles, u1 appears to radiate away from the intercondylar sulcus, largely within the axial plane (Fig. 27E). In these regions, u2 is oriented largely proximodistally in the sagittal plane, but elsewhere in the condyles u2 forms the butterfly pattern observed in all other groups (Fig. 27F).

Figure 27 The main architectural features of cancellous bone in the distal femur of ornithomimids and caenagnathids.

(A, B) Vector field of u1 in the lateral condyle of an indeterminate ornithomimid (TMP 99.55.337) in posterior (A) and lateral (B) views. (C, D) Vector field of u1 in the medial condyle of an indeterminate caenagnathid (TMP 86.36.323) in posterior (C) and medial (D) views. (E, F) Vector field of u1 (E) and u2 (F) in the distal femur of an indeterminate ornithomimid (TMP 91.36.569) at the level of the distal condyles, shown in proximal view for a 3D slice parallel to the axial plane (inset shows location of slice). In (A–D), the highlighted yellow vectors in the posterior extremities of the condyles have a much more mediolateral orientation compared to elsewhere in the condyle. This is also seen in (E), where vectors that appear longer are more parallel to the axial plane, and vectors that appear shorter are more proximodistally oriented.

The central part of the Falcarius distal femur studied (UMNH VP 12360) is fractured, so little can be said concerning the metaphysis, except that along the medial and lateral peripheries u1 is oriented largely parallel to the long-axis of the bone. In both medial and lateral condyles, the orientation of u1 is gently inclined posteriorly and subparallel to the sagittal plane, with little anteroposterior sweeping evident (Figs. 28A and 28B). The orientation of u2 in the condyles is subparallel to the axial plane, and forms a typical butterfly pattern.

Figure 28 The main architectural features of cancellous bone in the distal femur of Falcarius utahensis and Troodontidae sp.

(A, B) Vector field of u1 in the medial condyle of Falcarius (UMNH VP 12360) in anterior (A) and medial (B) views. (C) Vector field of u1 throughout the distal femur of Troodontidae sp. (MOR 553s-7.28.91.239), illustrating increasing obliquity and disorganization of vectors in the proximal metaphysis and transition to the diaphysis (region with braces). (D, E) Vector field of u1 in the lateral condyle of Troodontidae sp. (MOR 748) in anterior (D) and lateral (E) views. (F) Vector field of u2 in the condyles of Troodontidae sp. (MOR 748), shown as a 3D slice through the middle of the condyles in axial view; anterior is toward top of page.

Throughout much of the distal femur of Troodontidae sp., the orientation of u1 is subparallel to the long-axis of the bone, although in the proximal metaphysis it tends to become more obliquely oriented and disorganized, as seen in birds (Fig. 28C). No indication of any arcing patterns in u1, as observed in humans and birds, was observed. Moreover, the radiating patterns (originating from the intercondylar sulcus) that were observed in the tyrannosaurids, ornithomimids and caenagnathid are not evident either. The orientation of u1 in the condyles is subparallel to the sagittal plane and gently inclined posteriorly (Figs. 28D and 28E); as in Falcarius, little anteroposterior sweeping is apparent. As with all other groups, u2 in the condyles is subparallel to the axial plane and forms a butterfly pattern (Fig. 28F).

The mean orientation of the primary fabric direction in the medial femoral condyle for each specimen analysed is presented in Fig. 29A. Note that some of the smallest bird femora could not be analysed here, because they possessed too little cancellous bone to facilitate a quantitative analysis. As for the femoral head, a general orientation was taken to represent Allosaurus and the tyrannosaurids. In most specimens, the mean fabric direction is oriented posteriorly, with a small medial inclination. The human and Masiakasaurus specimens again plot close to the pole of the stereoplot, indicating an almost proximodistal mean direction. The plots for the ornithomimid, caenagnathid, Falcarius and Troodontidae sp. are slightly further from the pole, and the general Allosaurus–tyrannosaurid pattern is a little further away again. As in the femoral head, the bird results demonstrate marked variation, although on the whole a substantial posterior inclination is present; the mean direction across all birds is inclined 24.7° posterior of the proximodistal axis in the sagittal plane. Furthermore, the mean primary fabric direction in smaller birds tends to be more posteriorly inclined in the sagittal plane compared to that in larger birds, as indicated by major axis regression (Fig. 29B; slope = −0.12248, r2 = 0.3858, p = 0.00228).

Figure 29 The mean orientation of u1 in the medial femoral condyle, referenced in the femur anatomical coordinate system.

This is plotted on an equal-angle stereoplot, with southern hemisphere projection (using StereoNet 9.5). (A) The results for all specimens analysed; for clarity, only the posteromedial quadrant of the plot is shown. Note that for the fossil specimens, only those that were complete and well-preserved, and enabled an anatomical coordinate system to be defined, were analysed. Colour codes: black = birds, pink = human, blue = Masiakasaurus (FMNH PR 2153, UA 8384), orange = Falcarius, green = Troodontidae sp., purple = general orientation for Allosaurus and the tyrannosaurids, yellow = indeterminate ornithomimid (TMP 91.36.569), brown = indeterminate caenagnathid, red = mean orientation across birds. (B) Comparison of posterior inclination of u1 in sagittal plane versus femur length in birds, with major axis regression line (and associated statistics) plotted.

Tibia or tibiotarsus

General remarks

As with the femur, observations for the single human tibia studied are comparable with previously published reports (Takechi, 1977; Von Meyer, 1867), again supporting the use of this specimen as a general reference for humans. Cancellous bone is present throughout the entirety of both proximal and distal ends; a small amount also extends well into the diaphysis along the endocortical margin, but is mostly only one or two trabeculae thick (see section ‘Diaphyses’).

Cancellous bone is again generally more extensive in the birds and other extant reptiles compared to humans, at least in the larger species. It usually encroaches into the diaphysis to a varied extent, although moving towards mid-shaft it becomes more sparse and restricted to the endosteal margin of the cortex. Compared to the femur, the tibiotarsus of birds is generally less invaded by cancellous bone; this is especially true of the smaller species. Indeed, in some of the smaller species of birds, the medullary cavity extends well into the proximal and distal ends, to the point that only a handful of large, well-spaced trabeculae remain, with many smaller trabeculae distributed around the periphery. There is virtually no cancellous bone under the tibiofibular crest in birds, regardless of body size. Rather, the crest mostly comprises thickened (but higher porosity) cortical bone. In those birds with a prominent tibiofibular crest (e.g. Porphyrio, Gallinula, Ardeotis, Threskiornis), some cancellous bone does exist, but trabeculae are still few in number.

Owing to the logistical constraints of specimen availability and limited available scan time, the tibia and proximal tarsals of non-avian theropods were not investigated as thoroughly as they were for the femur. In those specimens that were studied, the proximal and distal ends of the tibia are fully occupied by cancellous bone (as is the entirety of the astragalus and calcaneum), but the majority of the diaphysis remains empty. Proximally, cancellous bone encroaches as far distally as the beginning of the tibiofibular crest; distally, cancellous bone usually does not occur any further proximally than the point at which the tibia begins to flare out mediolaterally. In the large tyrannosaurids, however, the distal diaphysis contains a significant amount of cancellous bone proximal to the point of mediolateral widening, which is again probably an effect of allometry. Cancellous bone is also present under the tibiofibular crest for its entire length, with more being present in the larger species.

Proximal tibia

In the human proximal tibia, the primary fabric direction u1 is more or less proximodistally oriented throughout the entire end. Under the medial and lateral tibial condyles it has a slight (approximately 10°) posterior inclination relative to the proximodistal axis, and under the medial condyle there is also a slight medial component (Fig. 30). As with the femur of humans, in the distal metaphysis and transition into the diaphysis u1 remains proximodistally oriented, with little disorganization or obliquity.

Figure 30 The main architectural feature of cancellous bone in the human proximal tibia is the gentle posterior inclination of u1 underneath the medial and lateral condyles.

(A) Vector field of u1 under the lateral condyle, in lateral view. (B) Vector field of u1 under both condyles, in anterior view. (C) Vector field of u1 under the medial condyle, in medial view.

In birds, the orientation of u1 throughout the proximal tibia is predominantly proximodistal, but there are marked departures from this locally throughout the bone. In the anterior cnemial crest, u1 has an anteroproximal inclination (Figs. 31A and 31B), which is much the same in the lateral cnemial crest, although a variable lateral component may also be present (Figs. 31C and 31D). In the thinnest parts of the crests where fabric was unable to be quantified, as well as in the cnemial crests of smaller bird tibiae, 3D visualization of the CT scan data demonstrates that the individual trabeculae tend to maintain this general orientation, essentially following the anterior margins of the crests (Figs. 31E and 31F). Under the medial articular condyle, u1 projects proximally, up and away from the metaphyseal cortex and arcs posteriorly towards the articular surface, generally remaining subparallel to the sagittal plane (Figs. 31G–31K). Immediately under the articular surface, u1 has a posterior inclination of about 20–30° to the proximodistal of the bone. A similar pattern occurs for u1 under the lateral articular condyle, although there is also a strong lateral component to the inclination; sometimes the amount of lateral inclination exceeds the amount of posterior inclination (Figs. 31L–31P). In the central part of the metaphysis, there is sometimes a double-arcuate pattern in u1 parallel to the sagittal plane; one ‘tract’ arcs from the posterior metaphysis to the cnemial crests, the other arcs from the anterior aspect towards the articular condyles (Fig. 31Q). Notably, this pattern was not observed in all specimens examined, not even in all specimens of the same species. Furthermore, the orientation of the secondary fabric direction u2 is not constrained to being subparallel to the plane of the arcing, as is the case in the human proximal femur. Progressing distally through the tibial metaphysis and into the diaphysis brings about increased obliquity and disorganization to the orientation of u1 (Fig. 31R). Fabric orientation could not be extensively quantified in the tibiotarsus of many of the smaller bird bones (or even at all in the smallest ones). Nevertheless, 3D visualization of the trabeculae themselves reveals that, in the regions of the cnemial crests and articular condyles, they tend to be oriented in the same general direction as u1 in the larger bird specimens (Figs. 31E, 31F, 31K and 31P). In the smallest species (e.g. Coturnix chinensis), there are hardly any trabeculae at all in the entire proximal end, with the medullary cavity extending almost to the proximal cortical surface. Additionally, little room for trabeculae exists in the cnemial crests between the two opposing cortices: the crests are either devoid of trabeculae, or there are very small trabeculae acting as spacers in a ‘sandwich structure’ (Currey, 2002).

Figure 31 The main architectural features of cancellous bone in the proximal tibia of birds.

(A, B) Vector field of u1 in the anterior (cranial) cnemial crest of an emu, Dromaius novaehollandiae (QMO 11686, A), and turkey, Meleagris gallopavo (RVC turkey 1, B), shown in medial view. (C, D) Vector field of u1 in the lateral cnemial crest of a southern cassowary, Casuarius casuarius (QMO 30105), shown in anterior (C) and lateral (D) views. (E) Isosurface rendering of cancellous bone in the anterior cnemial crest of an Australian white ibis, Threskiornis moluccus (PJB coll., between arrows), sectioned in the sagittal plane. (F) Isosurface rendering of cancellous bone in the lateral cnemial crest of a guineafowl, Numida meleagris (PJB coll., between arrows), sectioned in the plane of the crest. (G–J) Vector field of u1 under the medial condyle of an ostrich, Struthio camelus (MV R.2385, G, H), and chicken, Gallus gallus (PJB coll., I, J), shown in posterior (G, I) and medial (H, J) views. (K) Isosurface rendering of cancellous bone under the medial condyle of an elegant-crested tinamou, Eudromia elegans (UMZC 404.e, between arrows), sectioned in the sagittal plane. (L–O) Vector field of u1 under the lateral condyle of Struthio camelus (MV R.2711, L, M) and Dromaius novaehollandiae (QMO 11686, N, O), shown in posterior (L, N) and lateral (M, O) views. (P) Isosurface rendering of cancellous bone under the lateral condyle of a little spotted kiwi, Apteryx owenii (UMZC 378.iii, between arrows), sectioned in the coronal plane. (Q) Vector field of u1 in a 3D slice through the middle of the proximal metaphysis, cnemial crests and condyles of Dromaius novaehollandiae (QMO 11686), parallel to the sagittal plane. Schematic inset illustrates the moderately developed double-arcuate pattern present. (R) Vector field of u1 throughout the entire proximal tibia of Dromaius novaehollandiae (QMO 11686), illustrating increasing obliquity and disorganization of vectors in the distal metaphysis and transition to the diaphysis (region with braces).

In the extant sprawling reptiles examined, u1 is generally oriented proximodistally throughout the entire proximal tibia, although it fans out proximally, away from the middle of the bone towards the articular surfaces (Fig. 32). There does not appear to be any appreciable increase in the degree of obliquity and disorganization of u1 in the more distal parts of the metaphysis, as observed in the femur.

Figure 32 The main architectural features of cancellous bone in the proximal tibia of extant sprawling reptiles, as exemplified by a saltwater crocodile, Crocodylus porosus (QMJ 48127).

(A) Vector field of u1 in anterior view. (B) Vector field of u1 in medial view.

The proximal tibiae of both Allosaurus and the tyrannosaurids again show similar patterns to each other, as with the femur. Under the medial condyle, strongly developed tracts of cancellous bone are parallel to the sagittal plane and gently inclined posteriorly relative to the proximodistal axis (by 5–10°); they also have a gentle lateral inclination as well (Figs. 33A and33B). This continues into the region of the lateral condyle, but here the tracts assume a more marked lateral inclination, similar to the pattern described above for u1 in the birds (Figs. 33B and 33C). In the cnemial crest, the dominant direction of cancellous bone is largely parallel to the anterior margin of the crest, and is oriented proximoanterolaterally, again similar to the pattern described above for u1 in the birds. In axial cross-section, the cancellous bone actually forms concentric, proximodistally oriented ‘sheets’ that are parallel to the external surface of the cnemial crest (Figs. 33D–33J). Further posteriorly, towards the base of the cnemial crest, these sheet-like tracts become progressively more posteriorly inclined, directed toward the metaphysis. In the central metaphyseal region, a double-arcuate pattern is present, which roughly parallels the sagittal plane, where one set of tracts arcs up from the posterior periphery towards the cnemial crest (Figs. 33D–33J, purple and turqoise), and the other set of curved sheets arcs up from the anterior periphery towards the articular condyles (Figs. 33D–33J, green and maroon). Furthermore, 3D visualization indicates that these tracts are curved, more or less concentric with the bone margins. Progressing proximally, the anterior and posterior sets of tracts gradually change inclination to merge with the tracts in the regions of the cnemial crest and articular condyles, respectively. The tyrannosaurids provide a few further details on cancellous bone in the proximal tibia, owing to better contrast in their CT scans compared to those of Allosaurus. Firstly, in the (incipient) lateral cnemial crest, cancellous bone is oriented parallel to the margin of the adjacent part of the main (anterior) cnemial crest (Figs. 33D–33J, yellow). Secondly, in the fibular crest the cancellous bone forms a double-arcuate pattern parallel to the axis of the crest (Figs. 33D–33J, red and orange); these arcs intersect proximally, and proximally they also curve inwards towards the diaphysis. Thirdly, there is no indication in the distal metaphysis that cancellous bone is anything but oriented parallel to the long-axis of the bone.

Figure 33 The main architectural features of cancellous bone in the proximal tibia of both Allosaurus and the tyrannosaurids.

These are illustrated here with a 3D geometric model of the observed architecture, mapped to the tibia of Daspletosaurus torosus (TMP 2001.036.0001). (A) The dominant orientation of cancellous bone in the medial condyle, in medial view. (B) The dominant orientation of cancellous bone in the medial and lateral condyles, in posterior view. (C) The dominant orientation of cancellous bone in the lateral condyle, in lateral view. (D–J) Seven progressive rotations of the bone, in 30° increments, from proximally oblique medial to lateral views. For explanation of the features and colour coding, refer to the main text.

In the single proximal ornithomimid tibia that was studied (TMP 93.066.0002), the region of the lateral condyle was characterized by a tract of cancellous bone with a slight posterolateral inclination, superimposed on an otherwise proximodistal orientation. This is comparable to the pattern in Allosaurus and the tyrannosaurs.

The proximal tibia of Troodontidae sp. largely shows the same general patterns for u1 as described in the birds. Under the medial and lateral condyles, u1 exhibits a gentle posterior inclination superimposed on an otherwise proximodistal alignment (Figs. 34A–34D). However, only a slight lateral inclination occurs under the lateral condyle, in contrast to the often marked lateral inclination observed in birds, as well as Allosaurus and the tyrannosaurids. In the cnemial crest, u1 is again oriented largely parallel to the anterior margin of the crest, inclined proximoanteriorly (Figs. 34E and 33F). This pattern is also present in the cnemial crest of Saurornitholestes (Fig. 34G). Throughout the metaphysis of Troodontidae sp., the orientation of u1 is largely proximodistal; a weakly developed double-arcuate pattern, parallel to the sagittal plane, is present in one specimen examined, but not the other (Fig. 34H). This is similar to the large birds studied, where only some specimens exhibited a comparable double-arcuate pattern.

Figure 34 The main architectural features of cancellous bone in the proximal tibia of Troodontidae sp. and Saurornitholestes langstoni.

(A, B) Vector field of u1 under the medial condyle of Troodontidae sp. (MOR 553s-7.11.91.41) in posterior (A) and medial (B) views. (C, D) Vector field of u1 under the lateral condyle of Troodontidae sp. (MOR 748) in posterior (C) and lateral (D) views. (E, F) Vector field of u1 in the cnemial crest of Troodontidae sp. (MOR 748) in lateral (E) and anterior (F) views. (G) Vector field of u1 in the cnemial crest of Saurornitholestes (MOR 660) in lateral view. (H) Vector field of u1 in a 3D slice, parallel to the sagittal plane, through the central metaphysis of Troodontidae sp. (MOR 553s-7.28.91.239), shown in medial view. Schematic inset illustrates the moderately developed double-arcuate pattern present.

Distal tibia or tibiotarsus

In the human distal tibia, u1 is largely proximodistally oriented throughout the entire bone. However, throughout most of the metaphysis, u1 has a slight (10° or so) inclination relative to the proximodistal axis towards the centreline of the bone, forming a conical pattern with the cone’s apex pointing distally (Figs. 35A and 35B). The orientation of u2 is generally parallel to the axial plane; it has no orientation preference except in the periphery where it is generally subparallel to the local bone surface, forming a roughly concentric pattern (Fig. 35C).

Figure 35 The main architectural features of cancellous bone in the human distal tibia.

(A) Vector field of u1 in a 3D slice, parallel to the coronal plane, through the middle of the bone, shown in anterior view. (B) Vector field of u1 in a 3D slice, parallel to the sagittal plane, through the middle of the bone, shown in lateral view. (C) Vector field of u2 in a 3D slice, parallel to the axial plane, through the distal end of the bone, shown in proximal view (anterior is toward top of page). Inset shows location of slice.

The distal tibiotarsus of birds shows a characteristic pattern regardless of size (Fig. 36). By and large, u1 is oriented proximodistally and parallel to the sagittal plane throughout the whole distal end of the bone. In large birds for which substantial quantification of fabric was possible, u2 is oriented more or less anteroposteriorly throughout most of the tibiotarsus, in stark contrast to the human pattern, although it can become more parallel to the bone margin towards the periphery (Figs. 36A–36E). Within the condyles, u1 and u2 can also become ‘rotated’ within the sagittal plane to a variable degree. This distinctive pattern reflects the highly anisotropic and plate-like nature of the trabeculae (parallel to the sagittal plane) in this region of the bone, as evident in 3D visualizations (Figs. 36F–36H). In many of the smaller bird species, there was too little cancellous bone (too few trabeculae spaced too far apart) to permit an extensive quantitative fabric analysis, or any analysis at all in the smallest specimens. Nonetheless, 3D visualization clearly shows that most of the distal tibiotarsus in these species is dominated by relatively large, usually plate-like trabeculae that are oriented more or less parallel to the sagittal plane, qualitatively similar to the architecture observed in larger specimens (Figs. 36F–36H). For those species in which fabric analysis was possible to some degree, the general pattern in u1 and u2 observed for larger birds was also observed here.

Figure 36 The main architectural features of cancellous bone in the distal tibiotarsus of birds.

(A–D) Vector field of u1 (A, C) and u2 (B, D) in the distal tibiotarsus of an emu, Dromaius novaehollandiae (QMO 16140), in oblique anterolateral (A, B) and oblique anteromedial (C, D) views. (E) Vector field of u1 (red) and u2 (blue) in the condyles of Dromaius novaehollandiae (QMO 16140) in proximal view (anterior is toward top of page). Note how both u1 and u2 are strongly aligned parallel to the sagittal plane. This particular specimen exemplifies a very stereotypical pattern that is observed in all large birds; the general pattern illustrated here was also observed in smaller species for which only limited fabric analysis was possible. (F) Isosurface rendering of cancellous bone in the distal tibiotarsus of a southern cassowary, Casuarius casuarius (QMO 30105), shown in oblique anteromedial view, with multiple cuts through the bone to illustrate 3D architecture. (G) Isosurface rendering of cancellous bone in the distal tibiotarsus of a bustard, Ardeotis australis (MVB 20408), shown in oblique anterolateral view, with multiple cuts through the bone to illustrate 3D architecture. (H) Isosurface rendering of cancellous bone in the distal tibiotarsus of a painted quail, Coturnix chinensis (PJB coll.), sectioned in the axial plane through the middle of the condyles and shown in proximal view (anterior is toward top of page). In (F and G), cut surfaces are coloured red to better show the nature of the cancellous bone architecture, in particular, the plate-like nature of many of the trabeculae, largely aligned parallel to the sagittal plane.

In the distal tibia of extant sprawling reptiles, u1 is predominantly oriented proximodistally, although in the varanids it also takes on a posterior inclination throughout much of the metaphysis (Figs. 37A–37C). Distally, u1 tends to fan out towards the articular surface, much in the fashion as described for the proximal tibia. Similar to the human pattern, but unlike the birds, u2 is subparallel to the axial plane and largely concentric with the margins of the bone (Fig. 37D).

Figure 37 The main architectural features of cancellous bone in the distal tibia of extant sprawling reptiles.

(A, B) Vector field of u1 in a Komodo dragon, Varanus komodoensis (AM R.106933) in anterior (A) and medial (B) views. (C) Vector field of u1 in a saltwater crocodile, Crocodylus porosus (QMJ 48127) in anteromedial view. (D) Vector field of u2 in a 3D slice through the distal end of the tibia of Crocodylus porosus (QMJ 48127), shown in proximal view (anterior is toward top of page). Inset shows location of slice.

The distal tibia of both Allosaurus and the tyrannosaurids examined presents an intriguing cancellous bone architecture (Figs. 38A–38G). In the axial plane, it is manifest as a two sets of double-arcuate patterns that are largely parallel to the margins of the bone (Fig. 38D, inset), yet in both the sagittal and coronal planes it is manifest as a more typical double-arcuate pattern, with the arcs from opposing sides of the bone intersecting distally. In three dimensions, this produces a set of ‘Gothic arches’ (Garden, 1961). More distally, these arches progressively open up and become more distally directed, and the sheet-like tracts of cancellous bone become somewhat more anteroposteriorly oriented. They also start to fan away from the centreline of the bone, such that in the distal extremity of the tibia they are oriented medially on the medial side of the bone and are oriented laterally on the lateral side of the bone. Despite this opening up of the arches, a concentric pattern in the axial plane is somewhat retained. Within the astragalus and calcaneum, cancellous bone is only present in considerable quantities in the regions of the articular condyles. Here, the architecture is relatively simple, being strongly aligned in both the anteroposterior and proximodistal directions, paralleling the pattern for u1 and u2 in birds (Fig. 38H–38J). This pattern also occurs in the fused astragalocalcaneum of Ceratosaurus nasicornis (UMNH VP 5278). Progressing distally, it also fans out and away slightly from the centreline. Unlike in the distalmost tibia, there is no indication of a concentric pattern of cancellous bone in the axial plane.

Figure 38 The main architectural features of cancellous bone in the distal tibiotarsus of basal theropods.

These are illustrated here with a 3D geometric model of the observed architecture, mapped to the tibiotarsus of Daspletosaurus torosus (TMP 2001.036.0001; note that calcaneum was digitally sculpted based on other tyrannosaurid calcanei). (A–G) Seven progressive rotations of the tibia, in 30° increments, from proximally oblique medial to lateral views. Schematic inset in D illustrates generic cross-sectional pattern at the level indicated. (H–J) Three views of the astragalus and calcaneum, corresponding to those in (C–E), respectively. The various colours are used to help visualize the various tracts of cancellous bone more clearly.

In contrast to the complex architectural patterns observed in more stemward theropods, the architecture in the distal tibia of Troodontidae sp. and Saurornitholestes is relatively simple (Fig. 39). The orientation of u1 throughout the distal tibia is largely subparallel to the proximodistal axis, and u2 is largely contained in the axial plane and aligned anteroposteriorly (except around the periphery), much like birds. Furthermore, 3D visualization reveals that the cancellous bone architecture is dominated by plate-like trabeculae that are oriented more or less parallel to the sagittal plane, also like birds (Fig. 39C). Within the astragalus and calcaneum of Troodontidae sp., u1 and u2 become somewhat more disorganized, but importantly they largely remain parallel to sagittal plane. Thus, whilst the architecture of the distal tibia is decidedly different to that in the more plesiomorphic theropods examined, the architecture of the proximal tarsals is quite similar, as it is in birds.

Figure 39 The main architectural features of cancellous bone in the distal tibiotarsus of Troodontidae sp. and Saurornitholestes langstoni.

(A, B) Vector field of u1 (red) and u2 (blue) in the distal tibiotarsus of Troodontidae sp. (MOR 748) in anterior (A) and proximal (B) views; in B, anterior is toward top of page. Note how both u1 and u2 are generally aligned parallel to the sagittal plane. (C) Isosurface rendering of cancellous bone in the medial distal tibia of Saurornitholestes (MOR 660), shown in oblique anteromedial view, with multiple cuts through the bone (cut surfaces are coloured red) to illustrate 3D architecture.

Fibula

In humans, the primary fabric direction u1 is oriented proximodistally throughout the proximal and distal ends of the fibula (Figs. 40A and 40B). The same general pattern is also present in the extant sprawling reptiles examined, although u1 often fans out and away from the centerline leading towards the articular surfaces (Figs. 40C–40F).

Figure 40 The main architectural features of cancellous bone in the fibula.

(A, B) Vector field of u1 in the human fibula, in anterior (A) and lateral (B) views. (C, D) Vector field of u1 in a freshwater crocodile, Crocodylus johnstoni (QMJ 47916), in anterior (C) and lateral (D) views. (E, F) Vector field of u1 in an Argus monitor, Varanus panoptes (QMJ 91981), in anterior (E) and lateral (F) views. (G–I) Vector field of u1 in the fibular head of an emu, Dromaius novaehollandiae (QMO 11686, G), greater rhea, Rhea americana (QMO 23517, H), and chicken, Gallus gallus (PJB coll., I), in lateral view. (J, K) Isosurface rendering of cancellous bone in the proximal fibula of a malleefowl, Leipoa ocellata (MVB 20194, J), and painted quail, Coturnix chinensis (PJB coll., K), sectioned in the plane of the head and shown in lateral view. (L, M) The dominant architectural direction of cancellous bone in the fibula of Allosaurus and tyrannosaurids, shown in anterior (L) and lateral (M) views. This is illustrated here with a 3D geometric model of the observed architecture, mapped to the fibula of Daspletosaurus torosus (TMP 2001.036.0001). (N) Vector field of u1 in the proximal fibula of Troodontidae sp. (MOR 553s-8.17.92.265), in lateral view.

In all bird fibulae for which quantitative architectural analysis was possible, u1 is consistently oriented throughout the whole head, directed posteroproximally from the long-axis of the bone and subparallel to the local bone margin (Figs. 40G–40I). In those specimens for which quantitative analysis was not possible, 3D visualization revealed that the trabeculae themselves followed a similar orientation (Figs. 40J and 40K). There are typically very few trabeculae under the iliofibularis tubercle, even in the large birds, and none in the distal end, which is reduced to a splint of thin cortical bone.

In the fibular head of Allosaurus and the tyrannosaurids, as well as an indeterminate ornithomimid (TMP 2006.012.0065), cancellous bone is largely proximodistally orientated. It gently fans out away from the centreline of the bone in the sagittal plane, paralleling the anterior and posterior margins of the head and leading up towards the articular surface (Figs. 40L and 40M). In the distal fibula, the dominant architectural direction is largely parallel to the local centerline of the bone, and progressing distally this acquires a gentle lateral inclination (Figs. 40L and 40M). The proximal fibula of Troodontidae sp. exhibited a pattern in u1 that was comparable to that observed in the other theropods examined (Fig. 40N); the distal end of the Troodontidae sp. fibula that was studied was not preserved.

Diaphyses

As noted above, the diaphysis of birds and other extant reptiles often contains a significant volume of cancellous bone, in both the femur and tibia. In bird femora, and to a lesser extent the tibiae, the most conspicuous feature of this diaphyseal cancellous bone is the abundant trabeculae that are obliquely oriented relative to the long-axis of the bone, typically by 45° or more (Fig. 41). These trabeculae vary in individual form, displaying a range of rod-like to plate-like morphologies. They also vary in the degree to which they are connected to the adjacent cortex, ranging from being tightly appressed to the cortex (appearing little more than large ‘wrinkles’ in the endosteal surface), to being well separated from the cortex except at their ends, and arcing across the medullary cavity. When considered across the diaphysis as a whole, these oblique trabeculae tend to form conjugate helices that spiral along the endosteal margin of the cortex, especially in the bones of larger species (Figs. 41A, 41D and 41I). Markedly oblique trabeculae are also often present in the diaphysis of the sprawling reptile femora and tibiae examined, although they are not usually as abundant compared to the birds.

Figure 41 Oblique trabeculae in the diaphyses of birds and reptiles.

These are illustrated here with a number of examples demonstrating the variety of forms the individual trabeculae can assume. (A) Femoral mid-shaft of an emu, Dromaius novaehollandiae (QMO 16140). (B) Proximal femoral shaft of a turkey, Meleagris gallopavo (PJB coll.). (C) Distal femoral shaft of a malleefowl, Leipoa ocellata (MVB 20194). (D) Proximal femoral shaft of Dromaius novaehollandiae (QMO 11686). (E) Proximal femoral shaft of an Australian brush turkey, Alectura lathami (PJB coll.). (F) Proximal femoral shaft of an elegant-crested tinamou, Eudromia elegans (UMZC 404.e). (G) Femoral mid-shaft of an Argus monitor, Varanus panoptes (QMJ 91981). (H) Distal femoral shaft of a little tinamou, Crypturellus soui (MVB 23647). (I) Tibial mid-shaft of an ostrich, Struthio camelus (MV R.2385). In all figures, proximal is towards the top of the page.

The results of the categorical scoring analyses of cancellous bone architecture in bird femora and tibiotarsi are presented in Table 2 and Fig. 42. The ICC was moderate to good (Koo & Li, 2016) for all three femoral features (0.69–0.85), but was poor (0.46) for the one feature able to be scored for the tibiotarsus. For n = 5 scorers, this suggests that the results of the reliability of the scoring should be viewed tentatively, especially as concerns the tibiotarsus. In both the femur and tibiotarsus, the extent to which the diaphysis is occupied by cancellous bone (feature 1) tends to increase in larger bones, although in the tibiotarsus the increase is only really noticeable for the largest birds (Figs. 42A and 42B). The average orientation of the trabeculae relative to the long-axis (feature 2) also changes with size in the femur (Fig. 42C), but not the tibiotarsus (Fig. 42D). In the femur, it increases from approximately 45–50° in the largest birds to around 70° or more in the smallest birds. The average degree of association of trabeculae with other trabeculae (feature 3) appears to increase in larger femora and tibiotarsi (Figs. 42E and 42F); however, the results were not statistically significant. Thus, generally speaking, as bird femora get smaller, they become occupied by progressively less cancellous bone, the individual trabeculae of which are fewer in number and more widely spaced, and which have a more perpendicular orientation to the long-axis of the bone. Additionally, as bird tibiotarsi get smaller, they also become occupied by progressively less cancellous bone, but the arrangement of individual trabeculae does not appear to change significantly.

Table 2 Statistical results of categorical scoring analyses of cancellous bone architecture in bird femora and tibiotarsi vs. bone length.

Element	Feature	Slope	Intercept	r2	p-value	ICC	
Femur	1 (extent)	0.004123	0.3668	0.3846	0.0002	0.8533	
2 (orientation)	−0.0061	6.5355	0.2196	0.0245	0.8064	
3 (association)	0.00198	0.92015	0.1577	0.0609	0.6909	
Tibiotarsus	1 (extent)	0.001268	0.14686	0.2263	0.0168	0.4642	
2 (orientation)	−0.006	6.6016	0.2807	0.1438	***	
3 (association)	0.001055	1.0996	0.2239	0.2110	***	
Notes:

*** For features 2 and 3 of the tibiotarsus, an ICC was unable to be calculated, because there was only one specimen in each set that had actually received a numeric score by all five scorers.

Figure 42 Size-dependent variation in diaphyseal cancellous bone architecture in the femora and tibiae of birds.

These plots compare the results of the categorical scoring of each bone against its length. (A, B) Feature 1 (extent of cancellous bone) in the femur (A) and tibia (B); a higher score indicates greater extent. (C, D) Feature 2 (average orientation of trabeculae) in the femur (C) and tibia (D); a higher score indicates that trabeculae are more perpendicular to the bone’s long-axis. (E, F) Feature 3 (degree of association of trabeculae) in the femur (E) and tibia (F); a higher score indicates that trabeculae tend to be more closely associated with other similar trabeculae. Major axis regression lines are also plotted when correlations were statistically significant. N signifies number of species represented, and n signifies number of individuals. Other statistical metrics for each comparison are reported in Table 2.

Investigation of the human bones in this study revealed that a small amount of cancellous bone occurs along the endosteal surface of much of the tibial diaphysis. Whilst of insufficient quantity for quantitative fabric analysis, it is noteworthy that near the middle of the diaphysis, the cancellous bone architecture is dominated by trabeculae (or endosteal ‘wrinkles’) that are obliquely oriented, by about 10–20° to the long-axis of the bone (Fig. 43). Thus, some degree of similarity is present between the diaphyseal cancellous bone of humans and birds.

Figure 43 Gently oblique trabeculae in the tibial diaphysis of a human.

Inset shows location of section relative to the whole bone. Although the trabeculae (or endosteal ‘wrinkles’) are less obliquely oriented compared to the birds or reptiles, they are nonetheless consistently oriented in an oblique fashion.

Discussion

This study had two primary objectives, first of which was the broad-scale comparative assessment of cancellous bone architecture in the main hindlimb bones of avian and non-avian theropods, as well as in humans and some large extant sprawling reptiles. This comparative assessment focused on the gross architectural features across the whole bone, and used both quantitative and qualitative observations. The second main objective of the study was to draw upon the comparative assessment to identify patterns of similarity and contrast between the different groups examined, which may be used to provide insight into bone loading and locomotor biomechanics.

Overarching patterns across taxa

Despite great differences in size, and to a lesser degree, phylogenetic heritage, all birds investigated showed largely consistent cancellous bone architecture for a given region of a given bone. The patterns illustrated by birds were often in stark contrast to that exhibited by humans. The extant sprawling reptiles examined (varanid lizards and crocodilians) were also largely consistent in their observed architectural patterns across taxa, and typically showed greater similarity to the architectural patterns of birds than humans.

Among non-avian theropods, there are a number of different patterns of cancellous bone architecture in the femur. In the non-maniraptoriform theropods examined (Masiakasaurus, Allosaurus and tyrannosaurids), the femur showed marked similarity to the architecture observed in humans. In the proximal end, a pronounced double-arcuate pattern occurs in the coronal plane (visible in the tyrannosaurid specimens), and the primary fabric direction of cancellous bone (u1) in the femoral head had little anterior inclination, much like humans but unlike birds. In the distal femur, the primary architectural direction in the condyles did not have the often large posterior inclination as observed in birds, but it was often still greater than in humans. Additionally, the primary architectural direction in the condyles did not show the large amount of anteroposterior sweeping as observed in birds. The derived non-avian theropod Troodontidae sp. had a femoral architecture more closely resembling that of birds than most other non-avian theropods examined: no coronal plane double-arcuate pattern was present, u1 in the femoral head had a significant anterior inclination, and in the diaphysis-ward parts of the metaphysis the primary fabric vectors were disorganized and often oblique to the long-axis of the bone. This latter feature, ubiquitous in the femora and tibiae of birds and extant sprawling reptiles, is interpreted to reflect the onset of markedly oblique trabecular spirals in the diaphysis (see section ‘Oblique trabeculae in the diaphyses’).

The ornithomimid and caenagnathid femora examined illustrated a distinct and intriguing pattern in the proximal end that was not observed in any extant groups studied. Most conspicuously, u1 was oriented predominantly in an anteroposterior direction, with the secondary fabric direction (u2) oriented more or less proximodistally. Additionally, u1 tended to exhibit some significant amount of obliquity and disorganization in the diaphysis-ward parts of the proximal and distal metaphyses, and in this respect was comparable to the pattern observed in birds and sprawling reptiles. The femur of the basal therizinosauroid Falcarius shows some similarity to the ornithomimid and caenagnathid pattern in that the orientation of u2 in the central metaphysis and head was anteroposteriorly aligned, much like u1 in the ornithomimid femora. This is unlike the pattern observed in humans, or expected in Allosaurus and the tyrannosaurids based on the observed architectural patterns. However, the orientation of u1 in the diaphysis-ward parts of the proximal and distal metaphyses of Falcarius tended to be organized and subparallel to the long-axis of the bone, more comparable to humans and more plesiomorphic theropods. Distally, the femoral condyles of the ornithomimids, caenagnathid and Falcarius all had a gentle posterior inclination to the direction of u1, with little anteroposterior sweeping of the fabric, more like that in humans than birds.

The proximal tibia and fibula of the non-avian theropods showed a largely consistent architectural pattern, although it is acknowledged that these bones were not as extensively sampled as the femur. The architecture of the proximal tibia was quite comparable to that of birds, in both the cnemial crests and underneath the articular regions. A double-arcuate pattern, parallel to the sagittal plane, is present in the proximal end of the tibia of the large non-avian species, but is only occasionally present in Troodontidae sp. and large birds.

Cancellous bone architecture in the proximal tarsals (astragalus and calcaneum) was broadly comparable across all theropods examined, both avian and non-avian, being dominated by a strong anteroposterior-proximodistal alignment. In contrast, two distinctly different architectures were observed in the distal tibia (in non-avian theropods) or tibial component of the tibiotarsus (in birds). In the large non-maniraptoriform theropods (Allosaurus and tyrannosaurids), the distal tibia exhibited a complex double set of double-arcuate patterns, parallel to both the sagittal and coronal planes. In Troodontidae sp., Saurornitholestes and birds, however, the architecture was very much a continuation of that observed in the proximal tarsals, with a relatively simple pattern of anteroposterior-proximodistal alignment of u1 and u2.

Taken together, the various architectural patterns observed in the various bones of the non-avian theropods show a general correspondence with their phylogenetic relationships (Fig. 44). The more stemward theropods investigated tended to possess architectures that were more broadly comparable, and sometimes strikingly similar, to those in humans. In contrast, the paravians Troodontidae sp. and Saurornitholestes (as far as can be observed) possessed architectures more comparable to those of extant birds. Intriguingly, of all the non-avian theropods examined, Allosaurus and the tyrannosaurids had the greatest degree of posterior inclination in the fabric of the femoral condyles, most like birds (Fig. 44, character B). A further point of interest concerns the ornithomimid and caenagnathid specimens. Their femoral architectural patterns were similar to each other but distinctly different to those in other theropods studied, yet Ornithomimosauria and Oviraptorosauria are not thought to be each other’s closest relatives, nor do their lineages branch from successive nodes in theropod phylogeny (Zanno et al., 2009; Fig. 44, characters A, C and F). Thus, if taken at face value, the present set of observations suggest some amount of convergence in cancellous bone architectures, and by inference mechanical loading, between the two groups, or alternatively a reversion to a more plesiomorphic (tyrannosaurid-like) architecture in early therizinosaurs. Neither scenario can be tested until further specimens, of more species and more individuals of each species, are examined.

Figure 44 Summary of the main cancellous bone architectural patterns observed in the theropods studied, placed in context of their phylogenetic relationships.

Architectural patterns were discretized into six multistate characters, and optimized to phylogeny using equal-weighted, parsimony-based ancestral state reconstruction in Mesquite 3.40 (https://www.mesquiteproject.org/). Ancestral states are shown for each node, and character state changes along branches leading to terminal taxa are given in boxes. Characters (as letters) and states (as numbers) were defined as follows: A0, u1 in the femoral head has little anterior inclination; A1, u1 in femoral head has significant anterior inclination; A2, u1 in femoral head is oriented anteroposteriorly; B0, u1 in femoral condyles has small posterior inclination, with minimal anteroposterior sweeping; B1, u1 in femoral condyles has markedly increased posterior inclination; B2, u1 in femoral condyles often has strong posterior inclination, with substantial anteroposterior sweeping; C0, coronal-plane double arcuate pattern present in proximal femur; C1, proximal femur lacks coronal-plane double arcuate pattern; D0, sagittal-plane double arcuate pattern present in proximal tibia between condyles and cnemial crest; D1, sagittal-plane double-arcuate pattern in proximal tibia not as widely observed (present only in larger bones); E0, double set of double-arcuate patterns in distal tibia, juxtaposed with highly anisotropic, sagittally aligned fabric in proximal tarsals; E1, highly anisotropic, sagittally aligned fabric throughout both distal tibia and proximal tarsals; F0, fabric near diaphyses is subparallel to bone’s long-axis; F1, increased obliquity of fabric nearer the femoral diaphysis; F2, fabric strongly oblique near femoral diaphysis, and oblique trabeculae can extensively invade femoral diaphysis.

The importance of holistic analyses

The results of this study re-affirm the benefit of investigating cancellous bone in a holistic fashion, by considering patterns of architectural variation across whole bones and across multiple bones (Georgiou et al., 2018; Gross et al., 2014; Kivell, 2016; Ryan & Test, 2007; Saers et al., 2016; Saparin et al., 2011; Scherf, 2008; Skinner et al., 2015; Stephens et al., 2016; Su, Wallace & Nakatsukasa, 2013; Tsegai et al., 2013, 2017). Here, the integration of both qualitative and quantitative observations, across several bones, has helped identify many patterns of similarity and contrast between various theropod and non-theropod groups. Few of these patterns would be evident on the basis of a single observation alone. Indeed, considering only one or two regions in isolation of the others may well mislead the investigator into making incorrect interpretations. For example, the human specimen studied falls within or very near the cloud of data points for birds on stereographic plots of the mean orientation of u1, in both the femoral head (Fig. 22A) and medial femoral condyle (Fig. 29A). In isolation of all other observations, this would lead to the false conclusion that humans have a posture and locomotor biomechanics not too different from that of extant birds. Other observations, such as the double-arcuate pattern of u1 and u2 in the coronal plane of the proximal femur (present in humans but absent in birds), marked anteroposterior sweeping of u1 in the femoral condyles (absent in humans but present in birds) and often abundant, markedly oblique diaphyseal trabeculae (absent in humans but present in birds) point to a very distinct difference in locomotor biomechanics between humans and birds, a fact borne out in many experimental studies (Bishop et al., 2018, and references cited therein; Gatesy & Biewener, 1991). Only by considering all available cancellous bone architecture, across whole bones and across multiple bones, can the similarities and differences between species be truly appreciated, and only though this may a robust set of biomechanical inferences be developed. Moreover, in the case of extinct, non-avian theropods, future study of more species and more bones can add to the dataset produced here, further improving biomechanical inferences (see also below).

Biomechanical implications

As noted above, birds and humans demonstrate many distinct differences in cancellous bone architecture throughout their hindlimb bones. This was to be expected, since they exhibit very different bipedal locomotor biomechanics. Here, an attempt is made to correlate the salient differences in architectural patterns and biomechanics, and use these inferences to provide insight into the hindlimb locomotor biomechanics of the non-avian theropods studied.

Femoral head

The mean orientation of u1 in the femoral head of the human specimen had only a very small anterior inclination, whilst in birds the inclination was on the whole far more pronounced (Fig. 22A). This is interpreted to reflect the stark difference in femoral posture between humans and birds, with a large degree of flexion at the hip joint in birds leading to the hip joint reaction force being more anteriorly directed relative to the femur (Fig. 45). It cannot be discounted at present that the differences in architecture may also reflect differences in trunk posture between humans (orthograde) and birds (pronograde), although how trunk posture may influence hip joint loading remains unexplored. Most non-avian theropods examined had a minor anterior component to the mean orientation of u1, as in the human, suggesting that these species also held their femur in a similar, subvertical orientation. However, the mean orientation of u1 in Troodontidae sp. lay between that of birds and the other non-avian theropods, suggesting that its femoral orientation was intermediate between the subvertical posture of humans and the subhorizontal posture of birds.

Figure 45 Schematic demonstrating the effect of differences in the degree of hip and knee flexion on the joint forces experienced by the femur.

This is illustrated with right lateral views of a human (A) and a typical bird (B) in approximate mid-stance postures. In the more flexed posture of birds, the hip joint force is more anteriorly oriented relative to the long-axis of the femur (dotted line) compared to humans. Further, the knee joint force is more posteriorly oriented relative to the long-axis of the femur compared to humans.

Medial femoral condyle

Much as in the femoral head, the mean orientation of u1 in the medial femoral condyle is telling of postural differences between humans and birds, paralleling the results of previous experimental studies (Polk, Blumenfeld & Ahluwalia, 2008; Pontzer et al., 2006). In the human, it had only a small posterior inclination, whereas in birds the posterior inclination was generally substantial (Fig. 29A), which is inferred to reflect the greater degree of habitual knee flexion in birds (Fig. 45). This is supported by the results for the extant sprawling reptiles studied, which also exhibited a marked posterior inclination to u1 (although this was not strictly quantified), reflecting a marked level of habitual knee flexion during locomotion (Blob & Biewener, 2001; Clemente et al., 2013; Gatesy, 1991a). It was also demonstrated that the mean orientation of u1 in smaller birds tends to be more posteriorly inclined compared to larger birds (Fig. 29B). This reflects the fact that smaller birds tend to have a larger degree of postural crouch (Bishop et al., 2018; Gatesy & Biewener, 1991), which can be brought about by greater flexion of the knee joint. The non-avian theropods examined showed a variable amount of posterior inclination in the mean orientation of u1, although on the whole it was usually less compared to birds (less than 20°; Fig. 29A), suggesting a level of habitual knee flexion intermediate between that of humans and birds. No phylogenetic pattern was apparent, and curiously the Allosaurus–tyrannosaurid architecture had the greatest posterior inclination of all.

A further point of difference between birds and the other groups investigated here is the degree to which the orientation of u1 swept throughout the condyles in the anteroposterior plane. In birds, this sweeping often exceeded 100°, yet it was less than 40° in all other groups (although this was not strictly quantified). This may reflect the greater degree of habitual knee flexion birds, but it may also correlate to the greater range of knee flexion-extension employed by birds during the stride cycle compared to other groups (Andrada et al., 2013; Blob & Biewener, 2001; Clemente et al., 2013; Cracraft, 1971; Gatesy, 1991a, 1999; Kambic, Roberts & Gatesy, 2014, 2015; Reilly, 2000; Rubenson et al., 2007; Stoessel & Fischer, 2012; Winter, 2009). It should be noted that, owing to the fact that u1 swept throughout the condyles, the mean direction results reported for the medial condyle above need to be viewed with some caution. This is because significant sweeping in the anterior part of the condyles, particularly in larger species, could influence the calculated mean direction.

Proximal and distal femur

The pronounced double-arcuate architecture of cancellous bone in the coronal plane of the human proximal femur has been widely recognized for nearly two centuries (Ward, 1838) and much discussion has focused upon the mechanical significance of this for almost as long (see reviews by Cowin, 2001; Skedros & Baucom, 2007). Despite the various interpretations that have been proposed over time, the inescapable fact remains that this bears strong resemblance to the principal stress trajectories that would be engendered under mediolateral bending, via a load applied to the head of the femur (the ‘trajectorial theory’; see Introduction and Fig. 4). Such a loading regime, with compression dominating medially and tension laterally, is supported by in vivo strain gauge data (Aamodt et al., 1997). Moreover, the ‘primary compressive group’ that runs from the base of the femoral neck up to the head has been widely considered as reflecting transmission of the hip joint reaction force, away from the hip and down toward the rest of the femur (Skedros & Baucom, 2007). The presence of a strikingly similar double-arcuate pattern, also parallel to the coronal plane, in the proximal femur of the tyrannosaurids suggests that very much the same loading environment as occurs in humans also occurred in these species. Mediolateral bending of the femur is also suggested by another double-arcuate pattern, parallel to the coronal plane, in the distal femoral metaphysis.

An additional point of note is that the ‘primary compressive group’ in Allosaurus and the tyrannosaurids is directed towards the apex of the femoral head. By analogy with the proximal femur of humans, this suggests that the hip joint reaction force was principally applied there, implying that the articulation with the acetabulum was centred about the apex of the femoral head. This interpretation differs from previous suggestions, that the articulation was more lateral and involved at least part of the trochanteric region (Hotton, 1980; Hutchinson & Allen, 2009). The exact manner in which non-avian theropod hips articulated is therefore worthy of further study. Indeed, how the femur and acetabulum contacted each other may have varied both within and between various behaviours (e.g. differing degrees of hip abduction). These dynamic articulations could have possibly varied with different osteological morphologies (e.g. different degrees of inclination of the femoral neck) or soft tissue arrangements (Tsai & Holliday, 2015; Tsai et al., 2018), and probably changed appreciably on the line from basal theropods to extant birds.

Lesser trochanter

A second double-arcuate architecture in the lesser trochanter of the femur of the tyrannosaurids, also parallel to the coronal plane, further suggests that the trochanter also was loaded predominantly in mediolateral bending. This could conceivably occur via the medial pull (or medial component thereof) of the muscle(s) that inserted there, such as the iliotrochantericus caudalis (Hutchinson, 2001a). In such a situation it would be predicted that the medial arcade would be loaded in compression, and the lateral arcade in tension.

Proximal tibia

The orientation of u1 in the proximal tibia of the human is largely proximodistally oriented throughout the entire end, whereas it shows considerable variation throughout the bone in birds. Under the articular condyles, birds exhibit a more marked posterior inclination compared to that in the human (up to 30°, vs. about 10°), as well as a strong lateral component under the lateral condyles, which does not occur in humans. Anteriorly, u1 can take on a distinct anterior inclination as it parallels the leading margin of the cnemial crests, which are absent in humans. Within the metaphysis, a double-arcuate pattern in u1, parallel to the sagittal plane, may also occur in the proximal tibia of birds, which is also absent in humans. In these respects, the proximal tibia of all the non-avian theropods studied is more similar to that of birds. This similarity in cancellous bone architecture is undoubtedly due in part to the greater similarity in morphologies (e.g. prominent cnemial crest) and nature of the knee articulation (with the fibula being involved laterally) between the two groups. However, it does suggest that anteroposterior bending may be a more significant loading regime in the theropod tibia than the human tibia.

Distal tibia or tibiotarsus

The distal tibia of Allosaurus and the tyrannosaurids has two well-developed sets of double-arcuate patterns, one parallel to the sagittal plane, the other parallel to the coronal plane. By analogy with the proximal femur, this suggests that both anteroposterior bending and mediolateral bending were important loading regimes in this part of the bone. These two different loading regimes may possibly have been engendered during different behaviours, or at different instances during the one behaviour, such as different points throughout the stride cycle. Conspicuously, these complex patterns do not continue into the astragalus and calcaneum. Equally conspicuous is the different cancellous bone architecture in the distal tibia of Troodontidae sp., Saurornitholestes and birds, which is continuous with the architecture in the astragalus and calcaneum (in Troodontidae sp. and birds at least). Not only does this suggest tighter mechanical unity between the three bones in life in the case of Troodontidae sp. (indeed, the astragalus and calcaneum fuse in adults), but it also suggests that the distal tibiotarsus of Troodontidae sp. and Saurornitholestes experienced a different set of loading regimes compared to Allosaurus and the tyrannosaurids, but similar to that of extant birds.

Oblique trabeculae in the diaphyses

One of the more interesting results of this study was the observation of markedly oblique trabeculae in the diaphysis of the femur and tibia of birds and extant sprawling reptiles. Aside from some pterosaur wing bones (Wellnhofer, 1991), the authors are not aware of this feature being reported previously for any other tetrapod group. Interestingly, however, it also appears to be present in the proximal humeral diaphysis of orangutans (Pongo pygmaeus), judging from a figure published by Scherf, Harvati & Hublin (2013, figure 1a). In the present study, the observed oblique trabeculae tended to form helices that spiralled along the endosteal margin, especially in the bones of larger bird species. This feature is interpreted to be responsible for the progressive increase in obliquity and disorganization of the orientation of u1 in the diaphysis-ward part of the metaphysis in birds and sprawling reptiles. Essentially, the more ordered architecture of the main part of the metaphysis gradually breaks down and transitions to a sparser architecture of oblique trabeculae in the diaphysis.

Application of the trajectorial theory to the oblique trabeculae of the diaphysis of bird and reptile femora and tibia would suggest that these bones are loaded predominantly in torsion, or at the very least experience a significant amount of torsion during daily use. This is because for a cylinder under pure torsion, both maximum (tensile) and minimum (compressive) principal stresses are parallel to the margin and oriented at 45° to it, forming conjugate spirals (Beer et al., 2012; Carrano, 1998). In vivo strain gauge studies fully support this interpretation: the femora and tibiae of both birds and reptiles are loaded predominantly by torsion during locomotion (Biewener, Swartz & Bertram, 1986; Blob & Biewener, 1999; Butcher et al., 2008; Carrano, 1998; Carrano & Biewener, 1999; Main & Biewener, 2007; Verner et al., 2016). Oblique trabeculae were also observed to occur in the diaphysis of the human tibia, although not as strongly oblique to the bone’s long-axis compared to birds and reptiles (about 10–20°). This also concurs with in vivo data showing that a considerable torsional component to bone loading occurs during part of the stance phase of locomotion (Lanyon et al., 1975; Yang et al., 2014). The increase in obliquity and disorganization of u1 observed in the diaphysis-ward parts of the femoral metaphyses of Troodontidae sp., and to a lesser extent the ornithomimids and caenagnathid, therefore suggests that torsion was a more important (but not necessarily predominant) loading regime in the femur of these species. By contrast, the lack of any noticeable obliquity in u1 in the femora of the other non-avian theropods studied implies that torsion was minimally important. This too concurs with observations of the human femur, whereby u1 is subparallel to the long-axis of the bone in the diaphysis-ward parts of the metaphyses.

Whilst spiralling trabeculae in the femoral and tibial diaphyses of large birds is consistent with predictions of the trajectorial theory, the agreement breaks down in the bones of smaller birds. Specifically, in smaller birds the trabeculae tended to acquire an increasingly oblique orientation relative to the long-axis, approaching 70° or more; indeed, in some specimens, there were individual trabeculae that were almost orthogonal to the long-axis. Presumably, these smaller bird bones are also loaded predominantly in torsion, as are the bones of their larger relatives, on account of there being no evidence to the contrary, by way of anatomical, kinematic or kinetic observations. It would therefore be expected that principal stresses would still be approaching 45° to the long-axis of the bones. The lack of congruence between trabecular orientation and predictions of the trajectorial theory warrants explanation.

One possible explanation for the observed architectural patterns in smaller bird bones is that these bones are probably more liable to undergo failure through torsion-induced buckling, compared to the bones of larger species. In torsional loading, the critical shear stress needed to initiate buckling in a thin-walled cylindrical tube is given by (2) τcrit=kπ2Dl2t,

where k is a constant depending on the comprising material, l is the length and t is the thickness of the cylinder wall (Batdorf, 1947; Batdorf, Stein & Schildcrout, 1947; Donnell, 1933; Weingarten, Seide & Petersen, 1968). D is the flexural stiffness per unit length along the circumference, given as (3) D=Et312(1−ν2),

where E is Young’s modulus and ν is Poisson’s ratio for the material concerned. Moreover, the stress in a thin-walled tube loaded in torsion is related to the applied torque T as (4) τ=T2tA,

where A is the area of the cross-section (Beer et al., 2012); for a circular geometry, this means that (5) τ=2Tπd2t,

where d is the diameter. Therefore, the critical torque required to initiate buckling may be expressed as (6) Tcrit=Kπ3d2t32l2,

where K is a constant reflecting the material comprising the tube. A tube with a higher value of Tcrit requires a higher applied torque for buckling to initiate, and hence a higher value of Tcrit implies a lower propensity to buckle at a given load. Thus, the propensity for the tube to undergo torsion-induced buckling is proportional to the square of its length and inversely proportional to the square of its diameter. Previous studies of bird allometry have demonstrated that at smaller size, their hindlimb bones become progressively more slender (Alexander, 1983; Brassey et al., 2013; Carrano, 1998; Doube et al., 2012; Gatesy, 1991b; Olmos, Casinos & Cubo, 1996). Conversely, for a given size-normalized cross-sectional geometry, bones that are smaller in absolute terms will be longer in relative terms. The femora and tibiotarsi of smaller bird species may therefore be more prone to torsion-induced buckling. One way by which to mitigate buckling in a thin-walled tube is to support the tube walls against excessive transverse deflection, through the addition of structural stiffeners inside the tube (Chitale & Gupta, 2011). It is therefore hypothesized that the high-angle trabeculae in the femora (and less frequently, the tibiotarsus) of small birds are present mainly to provide cross-bracing support. By stiffening the diaphysis, they help prevent the dimensions of the bone changing too much to the point that buckling is initiated, which could lead to catastrophic failure at the whole-bone level.

Future work

It is worth reiterating the main limitations of the present study, and noting that these may be addressed in future investigations. The foremost limitation concerns that of sampling. In investigating phenomena not studied previously, the work presented here was very much exploratory. Logistical constraints restricted the number of species that were studied, as well as the number of replicates for each species and each bone. It was therefore not possible in many instances to provide a more precise, quantitative assessment of architectural variation, in terms of inter- or intraspecific variability, or how architectural patterns may relate to finer-scale differences in anatomy, behaviour or habitat. However, the data reported in Fig. 22B does suggest, at least qualitatively, that the potential for intraspecific variation in extant birds may be significant, and that intraspecific variation is therefore worthy of future scrutiny. In addition, although large specimens were sought for the extinct, non-avian theropods, this nevertheless could not fully control for possible ontogenetic influences, as recognizing skeletally mature adults in the fossil record can be problematic (Hone, Farke & Wedel, 2016). Furthermore, a number of major non-avian theropod groups were not investigated, such as basalmost tetanurans (e.g. megalosaurs), coelophysoids or alvarezsaurids. Increased sampling of non-avian species, as well as the number of individuals or bones for each species, is therefore an important objective for future studies, and will likely lead to greater refinement of the general patterns identified here, as well as subsequent biomechanical interpretations. Likewise, it will also be important to increase sampling of avian species, including those outside of the crown group, although as many fossils of stem-group birds are taphonomically flattened, this will admittedly be difficult. In addition, it may be worthwhile expanding the scope of study to include other bones as well; for instance, investigating cancellous bone architectural patterns in the ilium may provide further insight on hip joint loading mechanics and posture (cf. sections ‘Femoral head’ and ‘Proximal and distal femur’ above).

It would also be very worthwhile investigating potential effects of body size in non-avian theropods, by conducting denser sampling within lineages that display marked variation in body size (e.g. dromaeosaurids, tyrannosauroids). This can not only provide further illumination on the range of postures and locomotor biomechanics used by non-avian theropods, but may also help disentangle the relative influence of body size from other factors, such as phylogeny, musculoskeletal anatomy or the location of the whole-body centre of mass. For instance, the greatest amount of posterior inclination of the mean orientation of u1 in the medial femoral condyle was observed in Allosaurus and the tyrannosaurids, which were also the largest species studied. This observation is counter to what would be predicted from extant birds (Fig. 29B), and so may reflect other factors. Alternatively, this observation may indeed reflect their very large size, where at such large body size other factors become relevant, factors that are not important at smaller sizes.

A second limitation of this study concerns the fact that focus was directed primarily towards the directionality of cancellous bone fabric. Whilst architectural directionality is an important indicator of loading mechanics (see section ‘The fabric of cancellous bone (and why cancellous bone shows directionality)’ above), studies of extant species have demonstrated that other architectural parameters can also be useful, such as bone volume fraction, trabecular thickness and trabecular spacing. Future investigation of these parameters may therefore yield further insights. A serious obstacle to this line of enquiry, however, is being able to achieve sufficient CT scan contrast and resolution with fossil specimens, which currently is only possible for smaller specimens. Additionally, such study also requires excellent preservation of an entire fossil specimen.

One final methodological limitation worth noting here is that the quantitative architectural analyses performed were not able to be used in as extensive a fashion—if at all—in very small bird bones. This was because these analyses rested on the continuum assumption, which breaks down at small spatial scales. Essentially, there were too few trabeculae present to permit rigorous quantitative analysis. A goal for future studies may hence be to explore alternative ways of quantitatively characterising cancellous bone architecture in very small bones. One possible avenue is to use micro-finite element modelling (Ryan & Van Rietbergen, 2005; Van Rietbergen et al., 1999, 2003) to examine the principal material directions of the cancellous bone structure directly, although this can be computationally very expensive.

A further avenue for future work concerns the oblique or spiralling patterns of trabeculae observed in the femoral and tibial diaphyses of many birds, as well as other species. These patterns have hitherto never been reported outside of pterosaurs (Wellnhofer, 1991), and so it would be interesting to investigate how widely distributed they actually are among various tetrapod groups. It would also be worthwhile exploring more quantitative and objective methods of characterizing diaphyseal cancellous bone architectures; a simple categorical scoring approach was used in the present study, which if employed in further studies should employ greater numbers of scorers than that used here (five). Given the mechanical significance hypothesized above for the spiralling patterns observed in many bones examined in the present study, it would also be interesting to investigate whether their presence correlates with certain aspects of locomotor biomechanics.

Conclusion

This study has used new approaches for analysing and quantifying how the 3D architecture of cancellous bone varies throughout a limb bone, as well as new ways of comparing this architecture between species. In doing so, it has produced a broad survey of the major architectural features present in the main hindlimb bones of a variety of extinct, non-avian theropod species, as well as a variety of extant, ground-dwelling birds.

Qualitative and quantitative comparisons between non-avian theropods, birds, sprawling reptiles and humans have identified several patterns of similarity and contrast between these groups. Many of the observed patterns can be mechanistically linked to various aspects of locomotor biomechanics in the extant species, such as the degree of hip or knee flexion. This has in turn provided insight into locomotor biomechanics in non-avian theropods. Although explicit quantitative comparisons were conducted only for two regions of the femur in the present study, the approach used here can be expanded to the analysis of other regions of this and other bones in the future. Not only will this enable a more rigorous characterization of cancellous bone architectural variation in the various species, but it may also provide further bearing on interpretations of locomotor biomechanics, especially with increased sample sizes.

Cancellous bone architecture in the hindlimb bones of birds is quite consistent across the species studied. When variations were apparent, they could be related to differences in size or the presence or absence of pneumatization. Although variation due to phylogeny was not explicitly tested for in this study, no evidence for this was apparent. For instance, comparably-sized kiwis and chickens exhibited similar architectural patterns, as did comparably-sized tinamous and quail.

Broadly speaking, the cancellous bone architecture in more plesiomorphic theropods (ceratosaurs, Allosaurus and tyrannosaurids) is comparable to that in humans in many respects, but is often distinctly different from that observed in birds. The architectural patterns observed in Troodontidae sp. (and Saurornitholestes, where it was possible to assess) are typically intermediate between those of humans and birds. However, some features, such as the architecture of the distal tibiotarsus, are essentially identical to that of birds. Ornithomimid and caenagnathid femora both show a fairly distinct architectural pattern, different from all other groups studied. In particular, the primary fabric direction in the femoral head is largely anteroposteriorly aligned, and the fabric exhbits an axially radiating pattern in the ditstal femur.

Cancellous bone architecture in the hindlimb bones of non-avian theropods clearly varies between the different species studied, implying differences in locomotor biomechanics. Observed architectural features in the more plesiomorphic theropods studied suggest a manner of locomotion not too dissimilar from humans, with a subvertical femoral posture and mediolateral bending being the dominant loading regime in the femur. In contrast, Troodontidae sp. is inferred to have had locomotor biomechanics intermediate between those of the more plesiomorphic theropods and extant birds, befitting its phylogenetic position.

A particularly interesting architectural feature observed in the present study is the abundance of markedly oblique trabeculae in the diaphyses of the femur and tibia of birds, and to a lesser extent, extant crocodilians and lizards. In the bones of large species, this produces spiralling patterns along the endosteal surface of the diaphysis. It is hypothesized that this feature reflects a prominence of torsional loading in these bones during normal use. If this is correct, the presence of oblique or spiralling trabeculae can be used as an indicator of high-torsion limb bone loading in future studies of other extinct vertebrate species.

The staff of the Geosciences Program of the Queensland Museum is thanked for the provision of workspace and access to literature: A. Rozefelds, K. Spring, R. Lawrence, P. Tierney, J. Wilkinson and D. Lewis. A very special thanks is extended to the staff and associated colleagues of the institutions that provided access to the material studied here: D. Henderson, B. Strilisky, G. Housego, R. Russel, T. Courtenay, B. Sanchez and F. Therrien (Royal Tyrrell Museum of Palaeontology, Drumheller); R. Irmis, C. Levitt-Bussian, C. Webb and P. Policelli (Natural History Museum of Utah, Salt Lake City); J. Horner, J. Scannella, D. Varricchio, D. Strosnider, C. Woodruff, D. Fowler and T. Carr (Museum of the Rockies, Bozeman); K. Spring, H. Janetzki, A. Amey, P. Couper and S. Van Dyck (Queensland Museum, Brisbane); K. Roberts (Museum Victoria, Melbourne); R. Sadlier (Australian Museum, Sydney); and M. Forwood (Griffith University, Gold Coast). Many of the above people also provided helpful discussion on various aspects of theropod biology, and also helped transport specimens for CT scanning. The efforts of those who facilitated or performed the scanning itself are also much appreciated: S. Purdy and D. Wetter (Canada Diagnostic Centres, Calgary); K. Ugrin and D. Van Why (Bozeman Deaconess Hospital, Bozeman); M. Bauman, A. Price, S. Matinkhah and K. Sanders (University of Utah Hospital, Salt Lake City); S. Merchant, E. Hsu and J. Morgan (HSC Cores Research Facility, University of Utah, Salt Lake City); I. Mitchell and N. Newman (Queensland X-ray, Brisbane); K. Mardon (Centre for Advanced Imaging, The University of Queensland, Brisbane); and R. Lawrence (Queensland Museum, Brisbane). Great thanks are also due to the volunteers who undertook scoring of diaphyseal cancellous bone architecture: R. Lawrence, J. Rasmussen, J. Macmillan, D. O’Boyle and N. Bishop. SCANCO Medical AG kindly provided the scan image presented in Figure 4A, and Figures 1 and 6A were prepared with the assistance of N. Bishop and P. Tierney. The thorough and constructive comments on earlier versions of the manuscript, provided by S. Gatesy, T. Ryan, D. Henderson, E. Snively and an anonymous reviewer, are all greatly appreciated, and substantially improved the clarity and content of the research presented here.

Additional Information and Declarations

Competing Interests

Author Contributions

Data Availability

1 In an anisotropic continuous material, there exist a number of directions in which a given mechanical property (e.g., stiffness) is at its greatest or is lowest magnitude; these are its principal directions. In an orthotropic material such as bone (Cowin, 1986; Keaveny et al., 2001; Pidaparti & Turner, 1997), there are two such directions (one maximum, one minimum), which are orthogonal; there is also a third principal direction which is mutually orthogonal to the first two and is a minimax (intermediate). Perfectly isotropic materials have no principal directions, for each mechanical property is the same in every direction.

2 The compliance matrix C of a volume of material is a square matrix of order six (generalized Hooke’s Law for anisotropic materials) that describes its mechanical properties in terms of values of Young’s modulus, shear modulus and Poisson’s ratio. The fabric tensor H is a positive definite second-rank tensor that quantitatively describes the 3D microstructural arrangement of trabeculae in a volume of cancellous bone (Cowin, 1986). The principal axes of C and H are given by their eigenvectors (see next footnote).

3 For a given matrix A, there are one or more vectors v which maintain their original direction when multiplied by the matrix, although they are dilated by some scaling factor: Av = λv. These vectors v are the matrix’s eigenvectors, and the scaling factors λ are the matrix’s eigenvalues. The relative magnitudes of the eigenvalues describe the relative extent to which the matrix A is oriented in each direction given by the eigenvectors.

4 When a volume of material is under stress due to applied load, there will be three directions in which the shear component is zero; that is, only normal stresses (compressive or tensile) occur in these directions. The normal stresses in these directions are termed the principal stresses, and tangent lines to these directions form a network of principal stress trajectories. These trajectories essentially show how compressive and tensile forces are distributed throughout a body under loading.

5 The terms tibia and tibiotarsus are used in a specific fashion throughout this study. ‘Tibia’ refers to the bony element per se, whereas ‘tibiotarsus’ refers to the functional unit of the tibia and proximal tarsals (astragalus and calcaneum). Thus, ‘tibiotarsus’ is only meaningful when used in reference to theropods and other dinosaurs, in which the three comprising bones are tightly integrated, and in adult birds they become fused. In the other groups of animals studied, the presence of a tibiotarsal joint precludes the use of this term. As such, cancellous bone architecture in the proximal tarsals of theropods was also investigated.

John Hutchinson and Andrew Farke are Academic Editors for PeerJ.

Peter J. Bishop conceived and designed the experiments, analyzed the data, contributed reagents/materials/analysis tools, prepared figures and/or tables, authored or reviewed drafts of the paper, approved the final draft.

Scott A. Hocknull conceived and designed the experiments, analyzed the data, contributed reagents/materials/analysis tools, authored or reviewed drafts of the paper, approved the final draft.

Christofer J. Clemente conceived and designed the experiments, analyzed the data, contributed reagents/materials/analysis tools, authored or reviewed drafts of the paper, approved the final draft.

John R. Hutchinson analyzed the data, contributed reagents/materials/analysis tools, authored or reviewed drafts of the paper, approved the final draft.

Andrew A. Farke analyzed the data, contributed reagents/materials/analysis tools, authored or reviewed drafts of the paper, approved the final draft.

Belinda R. Beck analyzed the data, contributed reagents/materials/analysis tools, authored or reviewed drafts of the paper, approved the final draft.

Rod S. Barrett conceived and designed the experiments, analyzed the data, authored or reviewed drafts of the paper, approved the final draft.

David G. Lloyd conceived and designed the experiments, analyzed the data, authored or reviewed drafts of the paper, approved the final draft.

The following information was supplied regarding data availability:

All data and code used are held in the Geosciences Collection of the Queensland Museum, and will be made available upon a request being made to the Collections Manager (geoscience.inquiry@qm.qld.gov.au). Additionally, a complete copy of the fossil CT scan data obtained in the present study is accessioned with the respective institutions in which the specimens are housed.

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
