# Peer review of "Cancellous bone and theropod dinosaur locomotion. Part I—an examination of cancellous bone architecture in the hindlimb bones of theropods"

_PeerJ, doi:10.7717/peerj.5778_

## Round 0.1 · original submission · Minor Revisions

I'm happy to report that all three reviewers found this to be a valuable contribution, and all three recommended acceptance pending minor revisions. I've read and considered all of their recommendations, which I think are sound. Please pay special attention to the comments by Reviewer 1 regarding the experimental design and validity of findings - these are legitimate concerns that should be explicitly addressed. I will look forward to seeing an improved version of this work soon.

Reviewer 1 ·

Basic reporting

I would like to commend the authors on this substantive contribution to our understanding of avian locomotion biomechanics. The basic reporting of the article is fine. The manuscript is self-contained and rational behind the study is well justified.

In terms of the structure, particularly results and discussion, I would like to see a clearer distinction between data and interpretation pertaining to the 'quantitative' fabric analysis vs. the 'qualitative' geometric modelling or scoring system. Ideally I'd like to see the quantitative data first, including the stereoplots and regressions, then followed by a qualitative section with the more verbose description. Simply, I should be able to eyeball the first few tables and graphs, and quickly get an idea of what the broad patterns are. At the moment, the interesting findings are somewhat buried in the lengthy text.

In relation to the raw data, there is no mention of making the CT data available as far as I can tell. I leave that to the Editor to decide if that is acceptable. It would be nice to also include a data table of the orientation of primary fabric orientations for specimens when collected.

The literature review is extensive and relatively comprehensive. The authors do, however, fail to mention the many studies that have found distinctly equivocal results when attempting to correlate trabecular architecture with locomotor behaviour interspecifically. Farardo et al. (2007), Ryan and Walker (2010), Shaw and Ryan (2012) have all failed to distinguish between locomotor regimes on the basis on long bone trabecular architecture (although admittedly the authors don’t tend to calculate fabric direction).

Experimental design

Whilst the authors have been ambitious in scope, this has resulted in the inclusion of several different CT protocols and data analysis techniques that are not quite fit for purpose.

I am somewhat concerned that the authors have not quantified intraspecific variation in trabecular fabric orientation. The authors recognise this limitation, but suggest this was necessary in order to achieve their broad intraspecific sample. However, in light of the subsequent Part II and Part III, it is not clear to me how they have put this big interspecific sample to good use. Only 3 species (chicken and 2 dinosaurs) are analysed in any great detail. Furthermore, the authors have actually collected some intraspecific data on modern birds, as they recognise in the legend of Figure 22 by stating that 'significant intraspecific variation' exists.

In the primate literature, several studies have found quite a high degree of intraspecific variability in trabecular architecture (Tsegai et al. 2013; Ryan and Shaw, 2012). This isn’t necessarily a problem per se, as long as the interspecific variation relating to the differences in locomotor behaviour you’re interested in is greater. But given that the authors seem to have the modern replicates, it would be nice if this were given some consideration.

Linked to this, I would like to see more consideration given the potential taphonomic effects on fossil trabecular structure. The authors suggest fabric direction is more reliably assessed in fossils than trabecular thickness or spacing. However given that fabric orientation is obviously only useful relative to the anatomical axes, I wonder how one controls for potential deformation of the fossil, twisting, curvature etc. Again, having an idea of modern intraspecific variability is essential, in order that we can then understand the degree to which fossil taxa fall with this range. Larger intraspecific variation in fossils might be suggestive of some taphonomic artefacts.

There are also issues with the scanning methodology. Specimens were digitized in a range of scanners depending on their size and bulk density. Ultimately, larger specimens were scanned at a coarser voxel size. This might not be a problem if we can assume that larger dinosaur have absolutely larger trabeculae. But to what extent is this the case? Were the specimens at least scanned at a comparable 'relative' voxel size? How many voxels, say, comprised the average trabecular thickness, and is this consistent across the sample?

Obviously the authors have also been constrained by the scanning of 'whole' bones. I agree that it is a step forward compared to studying small cubes of excised bone, but it does limit scanning options and specimen availability. This would all be worthwhile IF the authors used a data analysis technique that could cope with the whole bone data. But they don't. Hence why lots of the small bird bones cannot be quantitatively analysed, and the authors rely on scoring metrics for midshaft trabeculae. It feels like the authors have embarked on a data collection exercise without having a sensible way of analyzing the resulting data. Ultimately, that means that only medium-sized birds and non-avian theropods are actually analyzed using the preferred methodology. The trabecular architecture of small birds and large dinosaurs cannot be quantified with the present methodology. I think this is a shame, and potentially obscures interesting scaling patterns and limits meaningful comparisons.

Validity of the findings

I think the endeavour of documenting and describing trabecular architecture across birds and non-avian theropods is worthwhile. However, I find the attempt to link fabric orientation and limb posture in the discussion to be weak.

Specifically, in the broader context of Parts II and III, it feels like the authors have missed an important stage:

Part I - Measure trabecular fabric in birds and dinos
Part II - Assuming fabric direction correlates to midstance posture, what is the mechanistic link?
Part III - Apply mechanistic model to dino fabric direction to predict midstance posture

But between Part I and Part II, I was expecting an analysis similar to those in the primate literature, in which trabecular variables are actually shown to be correlated to locomotor behaviour. In the Supplementary Materials, the authors have attached an accepted manuscript on modern bird kinematics, in which they have midstance posture for 12 species of bird, for which they also have fabric orientation. Why is there no attempt to correlate the two, since Part III relies upon there being a discernable postural signal in dinosaur trabecular architecture?

Additional comments

Specific comments:

Ln 754, 756, 758 - "Most", "A small proportion". Be specific about the number of specimens that could be quantitatively analyzed.

Ln 843-844 - The construction of 'geometric representations'. This proceedure is obviously not ideal, but is seemingly the only way to visualise the trabecular patterns in the larger theropods. If this were purely an exercise in visualization, this would be fine. But the authors then use this 'data' from the model in the stereoplot of Fig 22. We therefore need to check how well the orientation data extracted from the models agrees with those calculated automatically. The modelling proceedure should be repeated on 1-2 modern birds for which fabric orientation was also quantified, just to check the two metrics agree.

Ln 988 - How was pneumatisation determined?

Ln 998 - It would be nice if the authors included some orthoslice images of the original CT data for the larger theropod dinosaurs. So that readers can judge the quality of the underlying scan data

Ln 1128-1129 - There needs to be a better justification than, they were different so we excluded them

Ln 1250 - Mention regressions statistics here?

Ln 1276 - Logistical constraints such as....?

Ln 1487 - Of the 4 ICC scores calculated, two were 0.46 and 0.69. These cannot be considered 'high'. They are poor-average. I would go so far as to say that an ICC of 0.46 for 5 people is so bad that we shouldn't be interpretting that data at all.

·

Basic reporting

This is one of the best-written manuscripts I have reviewed. Two adjustments will slightly raise clarity and phylogenetic compliance. Be more specific with "this" and "the nature of". The latter could mean anything, and therefore conveys little. When you say "reptiles" clarify that you mean extant, non-avian and/or sprawling reptiles, varanids and crocodilians, or "other reptiles" (when also referring to birds and other theropods). There are a few suggestions for conciseness ("Because.." rather than "owing to" clauses), but the original wording is clear.

Also, be careful about "plesiomorphic" taxa; it's too blanket a characterization. Make sure the reader knows you're referring to the pelvic locomotor system relative to the condition in birds. Birds are just Allosaurus-line theropods in other ways.

Cite Tsai's work a little more for hip morphology, including his PhD thesis. You're correct that it's "worthy of .. study", and his soft tissue reconstructions are relevant to the results here (and vice versa).

The study is self-contained, and sets the stage for further installments of the research. The paper could start on line 493, but the (dissertation-derived?) background is made for online journals.

Experimental design

"Check" for all of the PeerJ criteria.

Fabric tensors are the best! It's wonderful to see them introduced to paleo so lucidly. Maybe (occasionally) refer to u1-3 as principal orientations in the results, to remind the reader.

The reconstruction protocols are actually replicable as described! The methods here and elsewhere are often a model of clarity and completeness.

Validity of the findings

Intriguing results from comprehensive methods.

Additional comments

Comments expanding on those above (including praise for the methods) are embedded in the reviewed manuscript. I found the trabecular architecture figures more informative than the volume-of-interest figures, although I recognize the latter's value. Splendid work, and I look forward to the rest of the opus.

·

Basic reporting

This article presents the preliminary results of CT scanning and 3D model construction from the CT data that surveys the cancellous bone structure in the hind limb bones of a range of living and extinct "reptiles" (crocodilians, lizards, birds, dinosaurs) and one human. It interprets the variety of forms of cancellous bone structure observed in the specimens in terms of the types of loading that the animals experienced when they were alive.

The paper is clearly and well written, if a bit verbose.

There is a MOUNTAIN of references that more than demonstrate the authors' familiarity with pre-existing work and ideas in the field.

The large number of figures support the findings reported in the text, but some more work is needed with the figures (see General Comments below).

The truly epic Table 1 thoroughly documents what specimens were examined and how they were scanned. The details provided should enable anyone to repeat the observations presented in the text.

The results are self-contained. Occasionally the authors do refer to the other two articles that are associated with this theropod CT scanning, but this is not a hindrance to understanding the current work.

Experimental design

The current paper satisfies all four of the stated criteria.

Validity of the findings

Again the current paper satisfies the four criteria. The authors make it very clear that very small bones caused them problems with both scanning and processing to build up a 3D model.

Additional comments

For the most part I enjoyed reading this paper. My only real complaint is that it reads like a thesis. I think a lot of the text could be reduced. This is especially so with the exhaustive reporting of the orientations of the trabeculae for ALL the bones and ALL their regions (proximal, mesial, distal) and sub-regions. My brain quickly overflowed with the blizzard of observations. I just couldn't take it in. I was relieved to find paragraphs summarizing the main trends and patterns in the Discussion.

Some specific comments follow:

Section I.3.4 Statistics

Starting at Line 925 the authors refer to ‘body size’. What do they mean exactly - linear body dimensions or mass? There are several studies that make use of femoral dimensions to estimate body mass. The recent, statistically rigorous work of Campione et al. would be a good place to get regression and scaling relationships to estimate the masses of the various animals that gave up their bones for analysis.

The first of only two word usage/typo mistakes seen: Line 1149 needs a ‘the’ after 'Throughout'.

Section I.5.1

This summary (big picture) section REALLY needs some graphics to help the reader visualize the trends and patterns reported for each of the regions of the bones (proximal, mesial, distal). The messages from the long, verbose interpretations could be much more effectively done by replacing them with some schematic figures. Figure 44 is a nice example of a summary figure. I think several more like it would greatly help in the presenting their findings and interpretations to readers.

Lines 1650-1661 is where the authors report the ranges of ‘sweep’, and bemoan the fact that the local variability of orientations makes the reporting of mean orientations somewhat “mean-ingless”. What if they compute and report the standard deviations about these means?

Section I.5.3
The authors go to the trouble of deriving an expression (equation 6) for the critical torque to initiate buckling, but then don’t do much with it except give a one sentence interpretation of the equation. They have all the geometric data, and with body mass estimates from regression analyses, they could actually compare ciritical torques for the specimens they scanned.

The second sentence structure complaint: Line 1849 doesn’t need the parentheses.

Figure comments and suggestions:

For all the figure captions the scientific names of animals ought to be italicized. Also, I think the common names for extant forms ought to be included as well.

Figure 12 – a 3D, oblique view of a femur with the three, mutually orthogonal axes displayed in proper perspective would be better than the current effort. I think the directions of +x and +y in the current illustration could be interpreted in more than one way.

Figure 15 and onwards – I would like to see at least one cut-away view for each bone that reveals the original trabecular orientations that were used to derive the u1, u2 and u3 vector fields. This has been done in selected figures, but it should be a consistent feature in all the figures.

Figure 19 – for the Allosaurus and Tyrannosaurus results we don’t get the usual red, blue and green vector fields. Instead we are presented with a new style of intersecting surfaces. These surfaces certainly show the trabecular trends, but for consistency and to allow comparison with the other scans, I think the vector fields should be shown as well. I know that the identification of what the colours of the surface represent is mentioned in the main text, but repeating it in the caption would be a great help. Placing a graphical colour legend right in the figure would be even better.

Figures 25 and 26 – the jumping up and down of the figure labels – ‘A’ on top, then dropping down to ‘B’, then back up to ‘C’ is disconcerting, and not seen in previous figures. This needs to be fixed.

Figure 33 sequence ‘D’ through ‘J’. I found it difficult to see the separate surfaces and perceive their orientations, no matter what view was shown. Also, the colours are rather muted as if they are being partially masked by the grey, semi-transparent surface of the bone.

Figure 36, 39 – the red overlays of parts ‘F’ and ‘G’ , and ‘C’ of figure 39, thoroughly obscure the trabecular details of the CT images. I think it would be best to shift the red overlays to lie immediately below the original images.

---

## Round 0.2 · Minor Revisions

Thank you for your diligence in addressing the reviewers' suggestions. Reviewers 1 and 3 still have some concerns. Although I don't think that the remaining issues are serious enough to block publication, you may wish to deal with them to improve the reception of the published work. I look forward to seeing a final revised version soon, and I anticipate publication soon after.

Reviewer 1 ·

Basic reporting

No comment

Experimental design

No comment

Validity of the findings

No comment

Additional comments

Author Response to Reviewer 1: As outlined in the Methods, the qualitative models produced (e.g., Figs 19, 26) were derived directly from the CT scans for those fossil specimens that were imaged using medical-grade machines. They simply convey the observed 3-D patterns in an alternative graphical format (since the fabric vector analysis was not possible), with no ‘higher level’ interpretations added on. Hence they portray just as much data as the fabric vector field plots.

Similarly, the verbal description of fabric vector fields (in an anatomical context) is a necessary item of data, because the cancellous bone architecture of theropod limb bones has never been described before; so the first thing to be done is to lay down the anatomical framework, which is a key objective of this exploratory study. Given the comparative theme to this study, it is important to present the data in as much a structured order as possible, in terms of both bony element and phylogeny.

Now that we have established a general anatomical and comparative background to work from, it is envisaged that future, more focused investigations will be able to report their findings in a manner more like what the Reviewer describes.

Reviewer 1 Response: Please try to provide a more concise summary of the broad trends. Both Reviewer 1 and Reviewer 3 have requested this change, but the authors have chosen not to do so. I appreciate this paper is derived from a thesis, and that summarising the main trends will be more work, but we are trying to help. I think your research is more likely to be cited if you do the reader a favour and spell out the main messages for them. I do not find the argument that a summary might only further confuse the reader convincing. If the authors have chosen to use such a mix of methodologies, across a diverse interspecific sample, and in such a verbose manner, then the burden of responsibility is with the authors to take on summarising the results for the reader. The authors themselves refer to the manuscript as a ‘comparative study’. How is this the case if it is not possible to make some broad comparisons and summarise trends across the sample?

Author Response to Reviewer 1:The data used in this set of studies collectively occupy about 3 TB (and some of the individual files are massive, too), which makes uploading the data to an online repository both impractical and prohibitively expensive. Nevertheless, all the data and code used are permanently accessioned into the Geosciences Collection of the Queensland Museum; the data and code are stored in a databank within a fire-proof safe on-site at the Museum. Anybody is able to freely access this permanent repository upon contacting the Collections Manager. (In future, this data will be made available online, but there are not available avenues for this now.) We originally had stated this solely in the Acknowledgements section, but now have made a statement in the Methods section as well (page 20, lines 613-616). Similar statements concerning data availability have now also been made in the Methods sections of Parts II and III.

Reviewer 1 Response: Again, this is shame for the authors. The work is more likely to be cited if you make your scans available for download from an online repository. Could you consider at least making the isosurface models you generate available?

Author Response to Reviewer 1: See overview comments above (re intraspecific scaling)

Reviewer 1 Response: I don’t agree with the authors’ assertions in-text that ‘It was…not possible to quantify potential intraspecific variation in bone architecture’. The data in Figure 22B clearly give an idea of fabric variation in the larger modern bird species. And more importantly, the variation does appear considerable. When discussing the difficulties associated with intraspecific variation in-text, please explicitly draw the reader’s attention to these figures and acknowledge that within-species variation does appear to be considerable in modern species.

·

Basic reporting

The writing is cogent, and needs no further modification. The authors thoughtfully incorporate broader and specific suggestions (e.g. line 405), and additional literature (such as in line 1804).

Reviewer 1 asks about availability of the raw and reconstructed CT data. I applaud the authors' literal safe-keeping of the data at the Queensland Museum (lines 613-616 and response-to-comments file).

Experimental design

As with the first submission, the research question and experimental design are insightfully and thoroughly described. The exhaustive methods are replicable, and Reviewer 3 is correct about their useful summary in the discussion.

Validity of the findings

Valid results and applicable findings, as the subsequent manuscripts show.

Additional comments

The revised manuscript is acceptable as-is.

·

Basic reporting

The manuscript meets all four criteria.

Experimental design

The manuscript meets all four criteria.

Validity of the findings

The data are derived from a limited set of specimens, hence the true nature of the variation between the sampled bones is not that well known. However, the authors recognize this and comment on it.

The authors do draw attention to their finding of the oblique trabeculae in the diaphyses of bird femora and tibiae. I don't know if this conflicts with the first point in points for reviewers at right.

Additional comments

This paper is an improvement on the previous version. It was a much less onerous read, although I still find the long series of descriptions of the u1, u2, u3 trajectories a bit heavy going.

My only quibble is with the interpretation of equation (6), with diameter 'd' in the numerator, and length 'l' in the denominator. Is it not the case that propensity for buckling is proportional to the square of 'd' and inversely proportional to the square of 'l'? This is not what is stated in the text.

---

## Round 0.3 · accepted · Accept

Thank you for your diligence in addressing the concerns of the reviewers. I am satisfied with the revised manuscript, and I am happy to accept it for publication in PeerJ.

The decision of whether or not to publish the peer reviews alongside the paper is entirely yours, and will not affect how your paper is handled going forward. However, I encourage you to do so. In this case in particular, the reviewers invested considerable time and effort in providing constructive criticism, and I think the exchange of ideas between you and the reviewers is a valuable part of the scientific process--it would be a shame to lose it. Making the reviews public allows the reviewers to receive credit for their efforts, and also contributes to the emerging culture of fairness and transparency in editing and peer review.


#